# Deep muscle-proteomic analysis of freeze-dried human muscle biopsies reveals fiber type-specific adaptations to exercise training

A. S. Deshmukh [1,2,8✉], D. E. Steenberg [3,8], M. Hostrup [4], J. B. Birk [3], J. K. Larsen[2], A. Santos[1], R. Kjøbsted [3], J. R. Hingst[3], C. C. Schéele [2,5], M. Murgia [6,7], B. Kiens [3], E. A. Richter [3], M. Mann [1,6] & J. F. P. Wojtaszewski[3✉]

Skeletal muscle conveys several of the health-promoting effects of exercise; yet the underlying mechanisms are not fully elucidated. Studying skeletal muscle is challenging due to its different fiber types and the presence of non-muscle cells. This can be circumvented by isolation of single muscle fibers. Here, we develop a workflow enabling proteomics analysis of pools of isolated muscle fibers from freeze-dried human muscle biopsies. We identify more than 4000 proteins in slow- and fast-twitch muscle fibers. Exercise training alters expression of 237 and 172 proteins in slow- and fast-twitch muscle fibers, respectively. Interestingly, expression levels of secreted proteins and proteins involved in transcription, mitochondrial metabolism, $Ca^{2+}$ signaling, and fat and glucose metabolism adapts to training in a fiber type-specific manner. Our data provide a resource to elucidate molecular mechanisms underlying muscle function and health, and our workflow allows fiber type-specific proteomic analyses of snap-frozen non-embedded human muscle biopsies.

[1] The Novo Nordisk Foundation Center for Protein Research, Clinical Proteomics, Faculty of Health and Medical Sciences, University of Copenhagen, Copenhagen, Denmark. [2] The Novo Nordisk Foundation Center for Basic Metablic Research, Faculty of Health and Medical Sciences, University of Copenhagen, Copenhagen, Denmark. [3] Section of Molecular Physiology, Department of Nutrition, Exercise and Sports, University of Copenhagen, Copenhagen, Denmark. [4] Section of Integrative Physiology, Department of Nutrition, Exercise and Sports, University of Copenhagen, Copenhagen, Denmark. [5] The Centre of Inflammation and Metabolism and Centre for Physical Activity Research Rigshospitalet, University Hospital of Copenhagen, Copenhagen, Denmark. [6] Department of Proteomics and Signal Transduction, Max-Planck-Institute of Biochemistry, Martinsried, Germany. [7] Department of Biomedical Sciences, University of Padua, Padua, Italy. [8] These authors contributed equally: A. S. Deshmukh, D. E. Steenberg. ✉email: atul.deshmukh@sund.ku.dk; jwojtaszewski@nexs.ku.dk

Regular exercise training ameliorates metabolic dysfunction in a variety of chronic diseases; however, our understanding of the underlying molecular mechanisms remains limited. The beneficial effects of exercise training involve improved glycemic control, partially due to enhanced insulin-stimulated glucose uptake in skeletal muscle of both healthy and insulin-resistant individuals[1,2]. Skeletal muscle is heterogeneous by nature, consisting of large multinucleated muscle fibers interspersed with various other cell types like adipocytes, satellite, and endothelial cells. By mass, muscle fibers contribute to the majority of muscle tissue, mainly due to their substantial content of contractile proteins[3]. Based on the expression of different myosin heavy chain (MHC) isoforms, human skeletal muscle fibers are classified into "slow-twitch fibers" (type I) and two types of "fast-twitch fibers" (type IIa and IIx). These fibers possess different functional and metabolic properties. Slow-twitch fibers contract more slowly, are more resistant to fatigue, and display a higher expression of oxidative enzymes, whereas fast-twitch fibers rely more on glycolytic enzymes to rapidly generate energy[4]. Importantly, this fiber-type-dependent composition is associated with muscle performance, locomotion, sarcopenia, muscle wasting, and muscle-associated metabolic diseases[5]. Therefore, illuminating molecular mechanisms on a fiber-type-basis offers potential for discovering novel therapeutic strategies and may help to implement optimal physical activity in the prevention and treatment of diseases.

Mass spectrometry (MS)-based proteomics is a powerful technology in biological research[6]. However, skeletal muscle proteomics is challenging due to the unfavorable dynamic range caused by highly abundant sarcomeric proteins[3]. Building on the latest development in MS-based proteomics, we and others cataloged the muscle tissue proteome to an unprecedented depth[3,7]. However, these deep muscle tissue proteomes do not distinguish between fiber types. Moreover, ~50% of all nuclei within rat skeletal muscle originate from other cell types than muscle fibers[8], hence confounding investigations of myofiber-specific proteins. Therefore, we devised a sensitive workflow to obtain the proteome of single muscle fibers[9] and subsequently showed that fast fibers age more rapidly than slow fibers as judged by their proteome dynamics[10]. In the present study, we investigated fiber-type-specific adaptations to exercise training in human skeletal muscle. Contrary to previous proteomic studies investigating the effects of training on a whole-muscle level[11–14], our comprehensive analysis of the proteome in both slow and fast fibers, as well as their adaptation to exercise training provides a valuable resource for exploring important proteins for muscle function and health. We performed fiber-type-specific proteomic analysis on snap-frozen, non-embedded freeze-dried human muscle biopsies. This provides advantages to the already established methodologies to obtain fiber-type-specific material, i.e. isolation of fibers from freshly (non-frozen) biopsy material[9] and laser capture microdissection (LCM) of muscle cryosections[15–17]. Isolating fibers from freshly obtained biopsies is challenging due to workflow and manpower in particular with protocols involving multiple biopsies. Hence, the time required for isolation is likely a confounding factor if proteomic analyses should include post translational modifications (PTMs). Our methodology can easily be adapted without advanced technology (as LCM) and does not require embedded muscle specimen for cryosection. This advancement is not trivial as many research studies have not included tissue embedding. We also imagine that with our technology, material sufficient to obtain deep global PTM proteomic analyses is easily obtainable.

## Results and discussion

**Study overview and proteomics workflow.** We analyzed vastus lateralis muscle biopsies from five young, healthy, untrained men obtained before (PRE) and after (POST) 12 weeks of endurance exercise training (1 h at 75–90% of maximum heart rate, 4× weekly). The subjects and training regime have been described elsewhere[18] (subject characteristics; Supplementary Table 1). From freeze-dried muscle biopsies, single fibers (Fig. 1a) were typified using MHC antibodies specific for "slow-twitch fibers" (MHC I) and "fast-twitch fibers" (MHC IIa/IIx) (Fig. 1b). Pools of 31–35 typified slow- and fast-twitch fibers were prepared for proteomic analysis and glycogen measurements.

All subjects completed >90% of the training sessions and remained weight stable (Supplementary Table 1). Training-induced adaptations were evident for several markers of exercise training; for instance, peak rate of oxygen consumption (VO$_{2peak}$) and insulin-stimulated leg glucose uptake increased by 16% and 41%, respectively (Fig. 1c). In addition, exercise training increased glycogen content in whole-muscle samples (+44%, $p < 0.01$) as well as in slow- (+134%, $p = 0.06$) and fast-twitch fibers (+97%, $p = 0.1$) (Fig. 1c).

Analytical challenges associated with highly abundant contractile proteins in skeletal muscle[3] can hinder detection of less abundant proteins. Thus, we maximized peptide detections by measuring the proteome of human primary muscle cells (myoblast and myotubes) together with the slow- and fast-twitch fiber pools under identical chromatographic conditions[19]. We used the "match between runs" capability in the MaxQuant computational proteomics platform. This feature uses liquid chromatography (LC) retention time alignments and peptide mass measurements to transfer identifications from a less complex system (myoblast and myotube) to the experimental system of interest making successful identification of low-abundant muscle peptides and proteins more likely[3,20]. We used a sequential multi-enzyme digestion filter-aided sample preparation (MED-FASP) strategy (with LysC and trypsin) to improve identification depth of peptides and proteins, as the sample amount was limited (~30 μg protein)[21]. All measurements were performed in a linear quadrupole Orbitrap mass analyzer, characterized by high sensitivity, sequencing speed, and mass accuracy, allowing us to expand proteome coverage of low-abundant proteins[22] (Fig. 1d). We used a false discovery rate (FDR) of 1% (peptide and protein level) and identified ~6000 protein groups with a median sequence coverage of 39% (Supplementary Data 1). A total of 4158 protein groups with 18,523 unique peptides and 5636 protein groups with 34,432 unique peptides were identified in the fiber pools and primary human muscle cells, respectively (Fig. 1e). With stringent identification criteria (50% valid value on total dataset), the number of identified protein groups in muscle fibers was 3360. When the raw data were processed without the "match between runs" option, we detected only 2830 proteins in muscle fibers (2299 and 1616 with 50% and 90% valid values, respectively), displaying the significant gain in protein identification by matching (Supplementary Data 1). Detailed analysis of the peptides detected in human muscle fibers revealed that the majority of the peptides were uniquely identified after LysC (32%) or trypsin (48%) digestion, whereas only 20% of the peptides were common between LysC and Trypsin digestion (Fig. 1e). Analyzing peptides from LysC and trypsin digestions individually (MED-FASP) has an advantage compared to conventional protocols where peptides from LysC followed by trypsin digestion are analyzed together (single shot). In support, comparing the single shot and MED-FASP protocol on whole-muscle lysate, we found an ~34% higher protein identification and an ~37% higher identification of unique peptides using the MED-FASP protocol (Supplementary Fig. 1a, b and Supplementary Data 2). Approaches involving two- or three-step digestions with various combinations of trypsin, LysC, GluC, ArgC, and AspN yielded

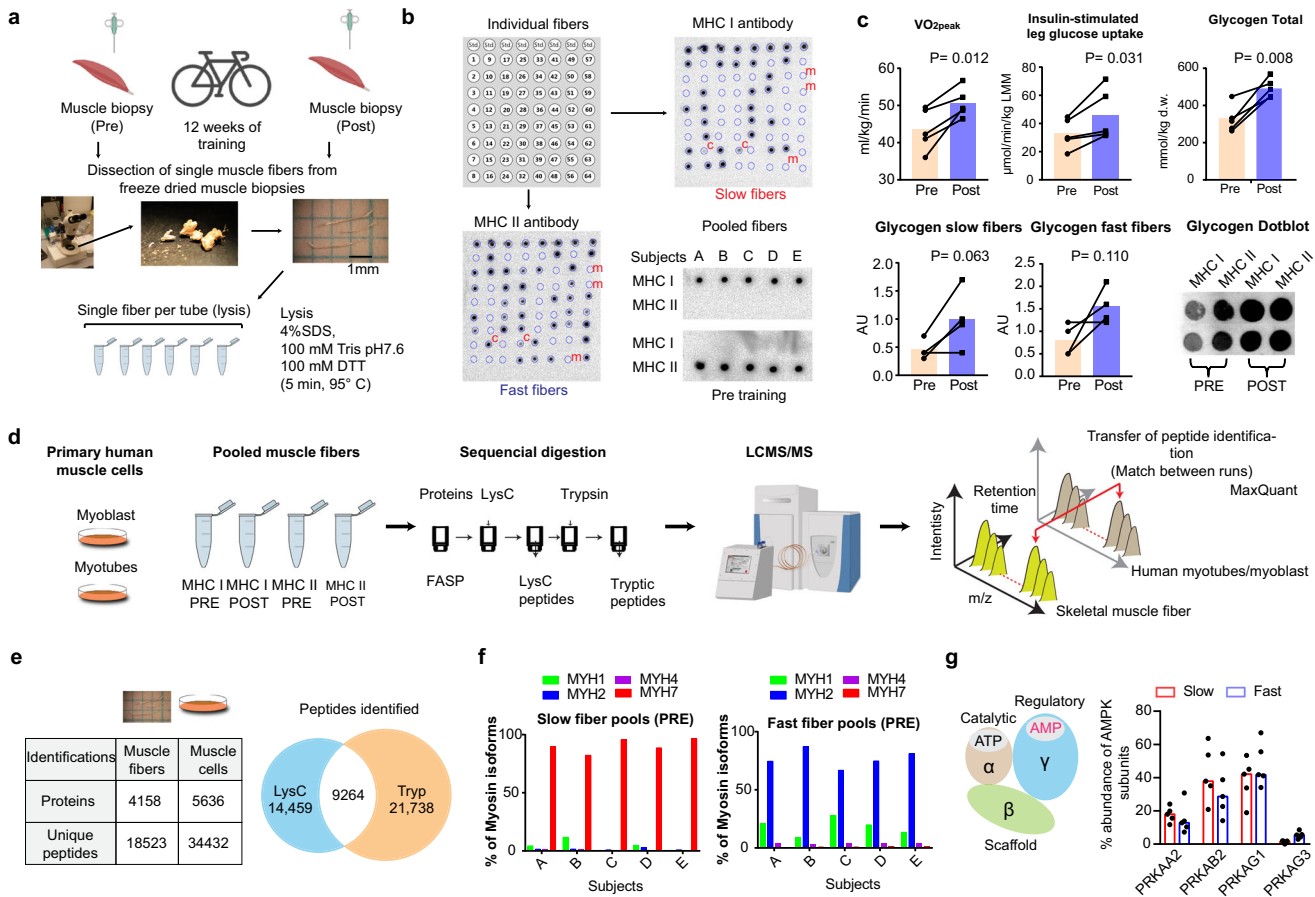

**Fig. 1 Workflow for isolation of muscle fibers and proteomic analysis. a** Single fibers were dissected from freeze-dried muscle biopsies before and after exercise training and fibers were dissolved in lysis buffer. **b** Fiber-type determination by dot-blotting. Top left: loading scheme of two identical membranes incubated in either MHC I (top right) or MHC II antibody (bottom left). m = missing fibers, not recognized by MHC I or II antibody; c = contaminated fibers (or hybrid fibers) were discarded. Fibers were pooled according to fiber type and the pools were validated by dot-blotting (bottom right). **c** Peak rate of oxygen consumption ($VO_{2peak}$) and insulin-stimulated leg glucose uptake (mean last 40 min of a 120 min insulin clamp) before (PRE) and after (POST) training. LLM lean leg mass. Glycogen content in whole-muscle homogenate (total) ($n = 5$) and in slow- and fast-twitch fiber pools ($n = 4$). AU arbitrary units, d.w. dry weight. **d** Proteomic analysis of purified muscle fibers and primary human muscle cells. Protein digestion by LysC and trypsin followed by optimized LC-MS/MS and computational analysis using MaxQuant with the "match between runs" feature. **e** Number of proteins and unique peptides identified in muscle fibers and cells and Venn diagram displaying number of common and unique peptides to LysC and trypsin fraction. **f** Unique peptide-based abundance of MHC isoforms in fiber pools PRE training ($n = 5$). **g** Unique peptide-based abundance of AMPK subunits ($n = 5$). Data in Fig. 1c are shown as means. *P* values by two-tailed paired *t*-test are indicated in Fig. 1c.

superior results while measuring proteome of simpler biological systems, HeLa cells and *Saccharomyces cerevisiae*[21,23,24]. Due to the complexity of skeletal muscle tissue, multi-step digestions with various combinations of enzymes may not ensure exactly similar results to simpler systems. While higher identifications are apparent with multiple digestion steps compared to single-shot proteomes, this comes at the cost of larger starting material and longer measurement times. In summary, our workflow allowed separation and identification of orthogonal populations of peptides, resulting in increased sequence coverage and depth of the proteome.

To investigate the purity of typified slow- and fast-twitch fiber pools, we quantified various myosin isoforms (MYH) by MS (Fig. 1f). Because MYHs have a sequence identity of more than 80%, we only used intensities of peptides unique for each isoform for protein quantification. The slow-twitch fiber pools displayed similarly high abundancies of MYH7 (~90%) (Fig. 1f), consistent with previous observations[10]. The fiber pools had excellent purity judged by the lack of other MYHs that would indicate mixed fiber types. The fast-twitch fiber pools (type II) were more heterogeneous in their MYH isoforms. MYH2 (type IIa) constituted

~80% of the total myosin pools, while MYH1 (type IIx) accounted for the remainder. Contribution of other MYHs (e.g. MYH4 and 7) was negligible as expected[10,25].

The list of identified proteins covered not only contractile proteins, enzymes of metabolic pathways, and mitochondrial proteins, but also transcription factors and known secretory proteins (Supplementary Data 1). For instance, mitochondrial transcription factor A (TFAM) and secreted protein Decorin (DCN) were identified with an average of 5 and 11 unique peptides, respectively. In addition, our analysis provided relative quantification of subunits of AMP-activated protein kinase (AMPK), a master regulator of energy metabolism (Fig. 1g)[26]. This confirms fiber-type-specific expression of the AMPKγ3 subunit by the use of immunoblotting[27].

Thus, our fiber isolation procedure and proteomic workflow yielded detailed proteome coverage, enabling us to explore the effects of exercise training in a fiber-type-specific manner.

**Proteomics analysis reveals new markers specific to slow- and fast-twitch muscle fibers.** We investigated the composition and

complexity of the slow- and fast-twitch muscle fiber proteomes at baseline (PRE training). The signal intensity of all quantified proteins in both fiber types was dominated by a few highly abundant sarcomeric proteins (Supplementary Data 3). For both slow- and fast-twitch fibers, the top 10 most abundant proteins constituted 50% of the proteome, in concordance with our previous muscle proteomics studies[3,9]. We examined the reproducibility of the biological quintuplicates and found high correlations within the groups (slow PRE, fast PRE, slow POST, and fast POST) (median Pearson's correlation = 0.91; Supplementary Data 4). Principal component analysis (PCA) revealed an excellent separation between slow- and fast-twitch fibers (Fig. 2a). Component 1 accounted for 31.8% of total variance and various isoforms of contractile proteins mainly drove the separation of the fiber pools (Fig. 2b). Segregation of slow- from fast-twitch fibers was mainly driven by MHC (MYH7, MYH6), Myosin light chain (MYL2, MYL3, MYL5), Tropomyosin (TPM4), Troponin (TNNI1), and Myozenin (MYOZ2) (Fig. 2b). Segregation of fast- from slow-twitch fibers was not only due to contractile proteins but to various other proteins, including NGFI-A-binding protein 2 (NAB2), Angiomotin (AMOT), and Family with sequence similarity 129 member A or protein Niban (FAM129A)(Fig. 2b, c). When comparing the proteome between slow and the fast muscle fibers, differentially expressed proteins were identified using an a posteriori information fusion scheme combining the biological relevance (fold change) and the statistical significance (P value) as described previously[28]. A $\pi$-value significance score cut-off of 0.05 was selected. Statistical analysis returned 471 proteins that were significantly different between slow- and fast-twitch fibers (significance score ≤0.05; Fig. 2d and Supplementary Data 5). In accordance with previous findings[4], we observed slow- and fast-twitch fiber-specific expression of various isoforms of sarcomeric proteins (Fig. 2c, d). Furthermore, we observed expected differences in glycolytic and oxidative enzymes between slow- and fast-twitch muscle fibers (Supplementary Data 5). Functional enrichment (Fischer Exact Test FDR < 0.02) on significantly different proteins highlighted enrichment of several protein categories, including muscle contraction, $Ca^{2+}$ channel, and glycolysis/gluconeogenesis, confirming fiber-type-specific contractile and metabolic differences (Fig. 2d and Supplementary Data 5). In addition, the enrichment of the protein categories associated with secreted proteins (extracellular region, extracellular vesicle exosome, and vesicle-mediated transport) indicates that slow- and fast-twitch muscle fibers have different secretion profiles (Fig. 2d).

Several proteins exhibited fiber-type-specific expression profiles. Among these, fiber-type-specific differences in FAM129A, PDZ, and LIM domain protein 1 (PDLIM1), and isopentenyl-diphosphate delta-isomerase 2 (IDI2) have, to our knowledge, never been reported before (Fig. 2c, d and Supplementary Data 5). The fast-twitch fiber dominant protein FAM129A regulates phosphorylation of proteins involved in translation[29]. PDLIM1 was exclusively quantified in slow-twitch fibers (Supplementary Data 5 and Fig. 2c). PDLIM1 interacts with four and a half LIM domain protein 1 (FHL1) and potentially regulates its localization within cells[30]. FHL1 is important for sarcomere assembly and muscle fiber maturation and it is associated with several types of myopathies. Although FHL1 expression was similar between fiber types, its interaction with PDLIM1 might be accountable for fiber-type-specific muscular dystrophies and acquired disorders[31,32]. IDI2, another rarely studied protein, was ~5-fold higher in slow- vs. fast-twitch fibers. IDI2 regulates isoprenoid metabolism and has been linked to statin-mediated myopathy[33]. Whether IDI2 spares/affects slow- and fast-twitch fibers during myopathy is unknown. Our discovery of novel fiber-type-specific markers holds promise for future research of muscle function and diseases.

Muscle contraction is driven by excitation–contraction coupling processes orchestrated by the highly organized sarcoplasmic reticulum (SR), terminal cisternae, transverse tubules (T-tubules), plasma membrane, and $Ca^{2+}$ channel complex (Fig. 2e, f and Supplementary Data 6)[34]. Gene Ontology Cellular Component (GOCC) annotations revealed that fast-twitch muscle fibers had a greater abundance of proteins related to the $Ca^{2+}$ channel complex, T-tubular system, and SR+ terminal cisternae (Fig. 2f and Supplementary Data 6). This observation coincides with the greater SR volume and convoluted structure in fast-twitch fibers, explaining their faster SR $Ca^{2+}$ kinetics[35].

We assessed organelle composition by calculating percent protein abundance of different organelles using subcellular localization annotations (GOCC) (Fig. 2g and Supplementary Data 6). As expected, mitochondrial and peroxisomal proteins were significantly more abundant in slow-twitch fibers, in line with their superior capacity for oxidative phosphorylation, fatty acid oxidation, and redox control. We observed significant fiber-type differences in proteins associated with the nucleus, plasma membrane, and cytoskeleton (Fig. 2g). These organelle-specific differences between the fiber types could be central to their differences in metabolic and contractile properties.

In summary, our comprehensive proteome coverage confirmed known and reveals new differences between slow- and fast-twitch fibers providing a starting point for exploring their role in muscle biology.

**Proteomic response to exercise training in slow- and fast-twitch muscle fibers.** Skeletal muscle has a remarkable ability to remodel its protein signature in response to physical activity and disuse[36], but this has never been analyzed in a fiber-type-specific manner. In response to 12 weeks of endurance exercise training, we found 237 and 172 proteins that were significantly regulated in slow- and fast-twitch muscle fibers, respectively (significance score ≤0.05, Fig. 3a, b and Supplementary Data 7), signifying marked remodeling of the muscle fiber proteome. Of these training-regulated proteins, ~40% were annotated to mitochondrion (Keywords, UniProt), while >65% were annotated to metabolic processes (Gene Ontology Biological Process) in both fiber types, demonstrating major metabolic rearrangement in response to exercise training. Exercise training also altered the abundance of 14 and 8 contractile fiber proteins in slow- and fast-twitch muscle fibers, respectively, which did not include the most abundant and fiber-type-defining MYH7 and MYH2. This was confirmed in whole-muscle lysate by western blotting (Fig. 3c) and proteomic analysis (Fig. 3a and Supplementary Data 7). MYH1 (type IIx) was downregulated by exercise training in whole-muscle lysate measured by proteomic analysis (Fig. 3a and Supplementary Data 7), while western blot analysis on whole-muscle lysate displayed a small, but non-significant, reduction (Fig. 3c). Collectively, this suggests that changes in MYH expression were not a major contributor to the fiber-type-specific adaptations to training.

We further investigated the effects of exercise training in slow- and fast-twitch muscle fibers individually. We observed greater adaptation to exercise training in slow-twitch (237 proteins) as opposed to fast-twitch fibers (172 proteins). Expectedly, training increased expression of proteins involved in oxidative phosphorylation (NADH dehydrogenase (NDUFB4, NDUFB10), ATP synthase (ATP5E), Mitochondrial DMQ-hydroxylase (COQ7), Citrate synthase (CS), and Cytochrome $c$ oxidase (COX7C)), mitochondrial import (translocase of the outer mitochondrial membrane (TOM) complex (TOMM40) and inner mitochondrial membrane (TIM) complex (TIMM21), and Transmembrane protein 70 (TMEM70), fatty acid metabolism (long-chain-fatty-acid-CoA ligase 1 (ACSL1), and Enoyl-CoA hydratase (ECHS1)),

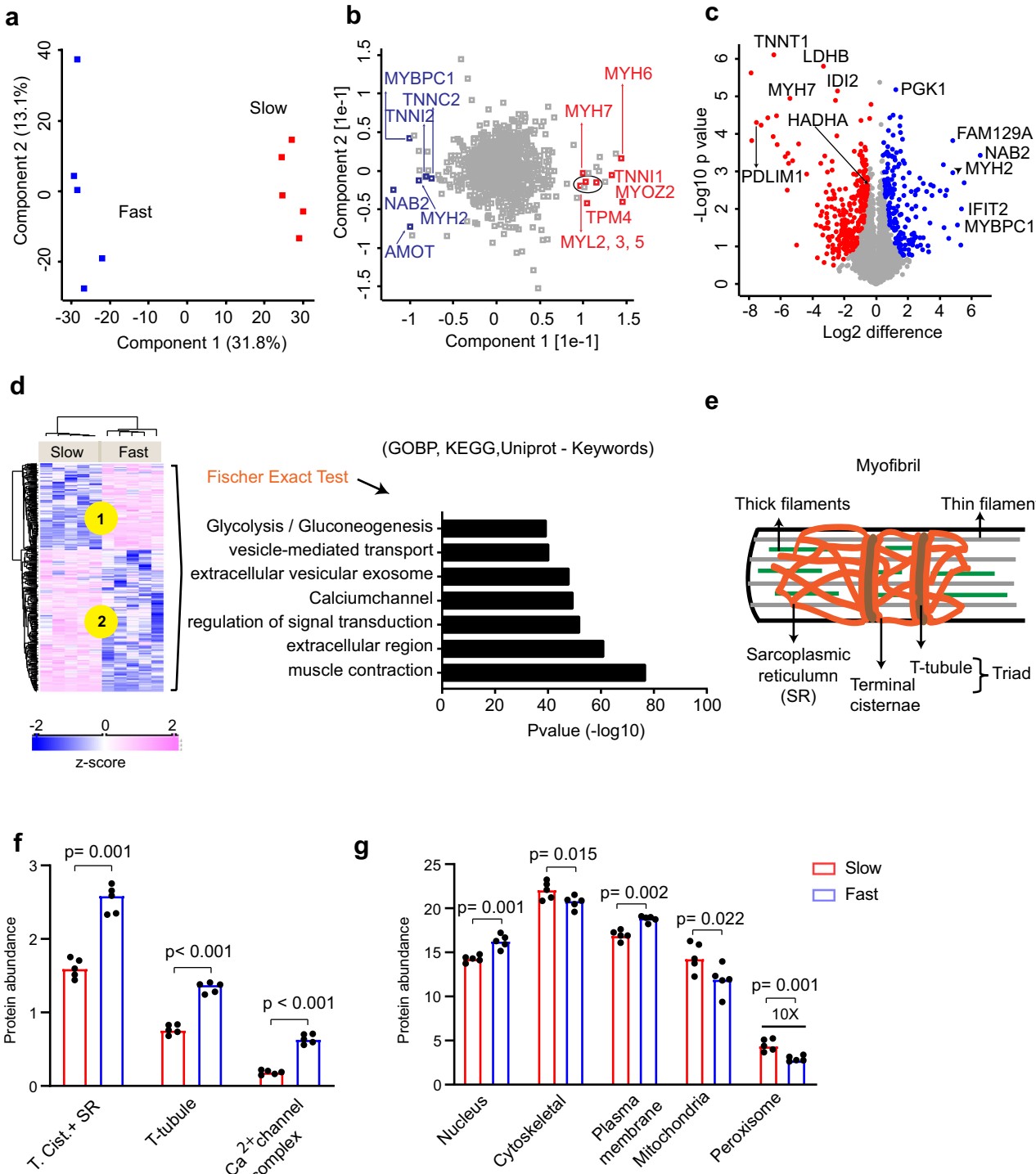

**Fig. 2 Quantitative proteome differences between slow- and fast-twitch muscle fibers. a** Principal component analysis (PCA) separating slow- and fast-twitch fibers in two groups. **b** PCA loading displaying selected examples of the most abundant proteins. **c** Volcano plot comparing protein abundance in slow- and fast-twitch muscle fibers. Differently regulated proteins (significance score ≤0.05; paired linear model with Xiao correction[72]) are marked in blue (high in fast-twitch fiber) and red (high in slow-twitch fiber). **d** Hierarchical clustering of significantly different proteins (left) and Fischer exact test for enrichment analysis of significantly different proteins (right). **e** Graphical representation of a myofibril. **f** Comparative protein abundance of Terminal cisternae, sarcoplasmic reticulum (SR), T-tubule, and $Ca^{2+}$ channel complex in slow- and fast-twitch fibers. **g** Major organelle protein composition in slow- and fast-twitch muscle fibers. Data in **f**, **g** are shown as median. *P* values by two-tailed paired *t*-test are indicated in **f**, **g**.

and creatinine metabolism (creatine kinase B, CKB). However, often the various isoforms of the proteins involved in these processes were upregulated by training in slow- but not in fast-twitch fibers or vice versa (Fig. 3a, b and Supplementary Data 7). Hexokinase II (HKII), a classical marker for training adaptions in

skeletal muscle, was significantly increased in slow-twitch fibers. It was not quantified in fast-twitch fibers likely due to low abundance, which was confirmed by western blotting (Supplementary Fig. 1c) and in accordance with previous observations on fiber-type-specific HKII expression[37]. Physical activity affects the

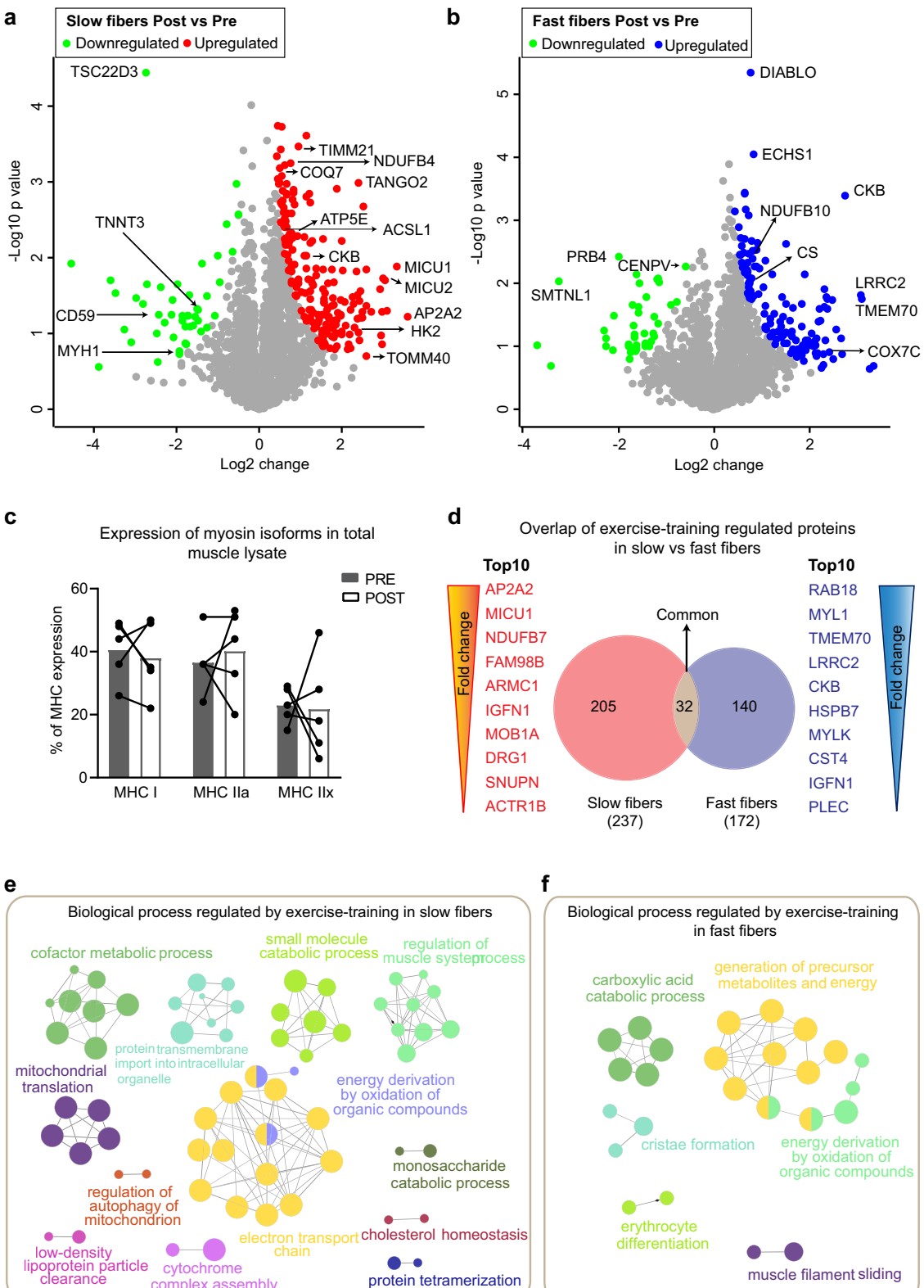

**Fig. 3 Proteomic responses of slow- and fast-twitch muscle fibers to exercise training. a** Volcano plot comparing protein abundance in slow- muscle fibers after exercise training. Differently regulated proteins (significance score ≤0.05; paired linear model with Xiao correction[72]) are marked in red (high after exercise training) and green (low after exercise training). **b** Volcano plot comparing protein abundance in fast-twitch muscle fibers after exercise training. Differently regulated proteins (significance score ≤0.05; paired linear model with Xiao correction[72]) are marked in blue (high after exercise training) and green (low after exercise training). **c** MHC composition in whole-muscle lysate PRE and POST training ($n = 5$). **d** Overlap of significantly different proteins between slow- and fast-twitch fibers. ClueGO-enriched network of exercise training-regulated proteins in slow- (**e**) and fast-twitch (**f**) fibers. The network represents clusters of related Gene Ontology terms enriched as nodes and connected if they annotate common proteins. For visualization purposes, clusters are labeled with the most informative term; node size represents the term enrichment significance and nodes.

intracellular signaling pathways in part via regulation of intracellular levels of $Ca^{2+}$ [38]. The mitochondrial calcium uniporter (MCU) is a selective channel for $Ca^{2+}$ entry into mitochondria[39]. MCU expression remained unchanged with training, whereas the two regulators of its activity, mitochondrial calcium uptake protein 1 and 2 (MICU1, 2), were increased by exercise training only in slow-twitch fibers (Fig. 3a, b and Supplementary Data 7). Mitochondrial $Ca^{2+}$ uptake has pleiotroic roles, including regulation of oxidative metabolism, $Ca^{2+}$-dependent functions in the cytosol, and apoptosis[40]. Thus, the training-induced increase of MICU1 and 2 expression in slow-twitch fibers might be linked to increased oxidative metabolism in these fibers. Finally, in both slow- and fast-twitch muscle fibers, exercise training upregulated proteins that were previously not associated with exercise training (e.g. slow-twitch muscle: Adapter related protein complex 2 (AP2A2) and Transport and Golgi organization 2 homolog (TANGO2), fast-twitch muscle: Leucine rich repeat containing 2 (LRRC2) and Diablo homolog (DIABLO)) opening new avenues for exercise research.

We next investigated the overlap between training-regulated proteins in slow- and fast-twitch fibers (Fig. 3d). Surprisingly, only 32 proteins were regulated in both slow- and fast-twitch fibers, of which 19 were mitochondrial (all upregulated in response to exercise training). Examples of these include proteins involved in oxidative phosphorylation (NDUFB7, NDUFAF2, ATP5E, ATP5F1, and electron transport flavoprotein (ETFB)), mitochondrial import (TIMM44), mitochondrial translation (mitochondrial ribosomal proteins (MRPL46 and MRPS35)), carnitine and creatinine metabolism (Carnitine O-acyltransferase (CRAT) and Creatine kinase S-type (CKMT2)) and fatty acid metabolism (ACSL1), suggesting that mitochondria are common effectors to exercise training for both fiber types. Moreover, when we ranked the top 10 exercise-regulated proteins based on fold change, only one protein, immunoglobulin-like and fibronectin-type III domain-containing protein 1 (IGFN1), was common in slow- and fast-twitch fibers. These results show that slow- and fast-twitch muscle fibers adapt differentially to exercise training.

We further performed a comprehensive analysis of biological processes enriched in the exercise training-regulated proteins in slow- and fast-twitch fibers using ClueGO[41]. Exercise training-regulated numerous biological processes pertaining to energy turnover, particularly metabolic pathways, and the mitochondria (Fig. 3e, f and Supplementary Data 8). In contrast to fast-twitch fibers, slow-twitch fibers displayed enrichments for more protein categories, partly because slow-twitch fibers had 27% more proteins regulated by exercise training. Enrichment of protein categories in slow-twitch fibers was primarily dominated by mitochondrial metabolism (electron transport chain, mitochondrial translation, cytochrome complex assembly). In summary, our unbiased proteomic analysis presents a detailed overview of effects of exercise training in slow- and fast-twitch muscle fibers. It confirms both previous observations and reveals novel proteins and biological processes regulated by exercise training.

**Slow- and fast-twitch muscle fibers adapt differentially to exercise training**. While many studies have identified mechanisms regulating exercise training adaptations in skeletal muscle[42], less attention has been given to elucidate those mechanisms in a fiber-type-dependent manner[43]. We analyzed the interaction of exercise training and fiber type, i.e. the fiber-type-specific adaptations to exercise training (Fig. 4a, b). Analysis returned 131 proteins that were regulated in a fiber-type-specific manner (significance score ≤0.05) (Fig. 4b and Supplementary Data 9). The number of proteins regulated in a fiber-type-specific manner is likely larger, as we only included proteins that were quantified

(three valid values in at least one group pre or post) in both fiber types for this analysis. The proteins, which were quantified only in one fiber type, was termed "exclusive" to that fiber type. The exclusive proteins could either be uniquely expressed in that fiber type or low abundant in the other fiber type and therefore not picked up by the mass spectrometer for sequencing (due to the nature of the technology). Of the 284 proteins, which were exclusively quantified in slow-twitch fibers, 61 were regulated by exercise training (Supplementary Data 9), this included HKII, MICU1, MICU2, and TANGO2 (described previously). These proteins are likely specifically regulated in slow-twitch fibers after exercise training. Similarly, among the 124 fast-twitch fiber-exclusive proteins, 13 were significantly changed by exercise training. For instance, TP53-induced glycolysis and apoptosis regulator (TIGAR), a glycolytic inhibitor and a protector against reactive oxygen species (ROS)-associated apoptosis[44], was increased after exercise training only in fast-twitch fibers (Supplementary Data 9).

We next investigated the functional relevance of the 131 proteins regulated in a fiber-type-specific manner. The enrichment analysis (Fischer exact test) did not return any significant protein categories partially because these proteins were involved in a wide range of biological processes (Supplementary Data 9). Notably, we found 18 known secreted proteins (e.g. Cystatin-S (CST4), CD59 glycoprotein (CD59), and Mucin-5B (MUC5B)), which were regulated in a fiber-type-dependent manner. For instance, the poorly characterized protein CST4 was ~9-fold upregulated in fast-twitch fibers while it remained unchanged in slow-twitch fibers upon exercise training (Fig. 4b, c and Supplementary Data 9). Proteins secreted from muscle (myokines) are implicated in the autocrine regulation of muscle metabolism as well as the para/endocrine regulation of other tissues[45]. Secreted protein CD59, which was downregulated in slow-twitch fibers, is a primary inhibitor of the membrane attack complex (MAC) of the complement system and plays an important role in the immune system[46]. Defects in CD59 are associated with MAC-induced lesions in juvenile dematomyositis[47], which suggests that reduced expression of CD59 in slow-twitch fibers could be a consequence of muscle damage in response to exercise training. The majority of these myokines are still not sufficiently characterized. As such, detailed functional validation of these may help understanding muscle metabolism at the fiber-type level.

Slow- and fast-twitch muscle fibers are different in composition of contractile and mitochondrial proteins. We observed fiber-type-specific changes in 13 contractile and 19 mitochondrial proteins in response to exercise training. Examples of these are highlighted in Fig. 4b, c. Interestingly, the exercise training-induced fiber-type-specific regulation of these proteins was independent of their PRE-training expression. For example, while expression of CKB was approximately 23-fold higher in slow- than in fast-twitch fibers PRE training, it exhibited a greater response to exercise training in fast-twitch fibers (Fig. 4b, c and Supplementary Data 9). Fiber-type-specific regulation by exercise training was also evident for proteins involved in vesicle trafficking (AP2A2), transcription (TATA-binding protein-associated factor 2N (TAF15)), and proteins with unknown functions (Ras-related protein Rab-18 (RAB18)) (Fig. 4b, c and Supplementary Data 9).

Our data show that human slow- and fast-twitch muscle fibers display remarkable differences in adaptations to exercise training. Interestingly, when we compared the training-responsive proteins in whole-muscle lysate with those seen in slow- and fast-twitch muscle fibers, we observed little overlap (Fig. 4d, e). For instance, only 21% and 12% of the upregulated proteins in whole-muscle lysate were also upregulated in slow- and fast-twitch muscle

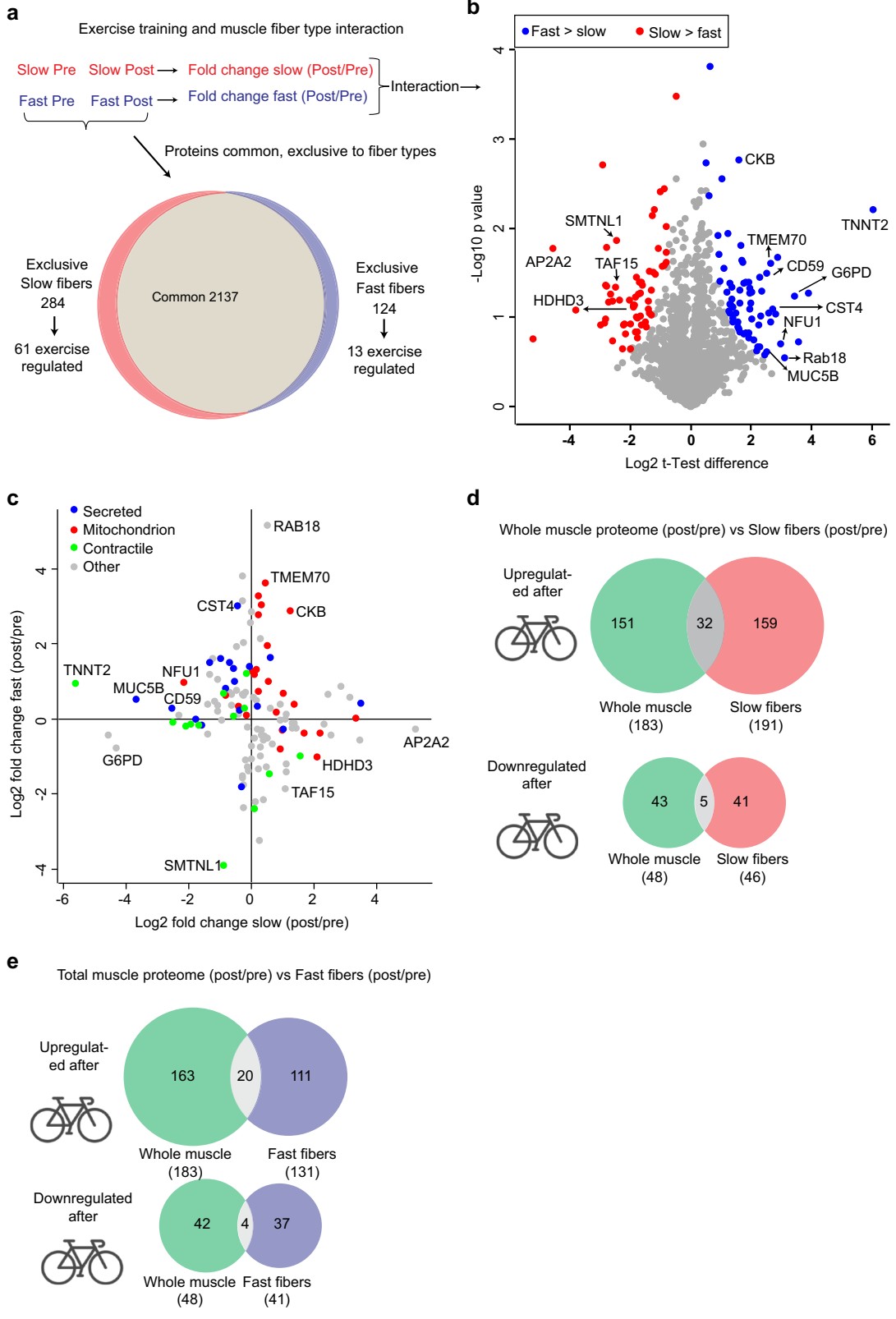

**Fig. 4 Slow- and fast-twitch muscle fibers adapt differentially to exercise training. a** Diagram displaying proteins that are commonly and exclusively expressed in slow- and fast-twitch muscle fibers + the number of exercise training-regulated proteins in the exclusive population. **b** Volcano plot comparing fold change (POST vs. PRE) between slow- and fast-twitch muscle fibers. Differently regulated proteins (significance score ≤0.05; paired linear model with Xiao correction[72]) are marked in blue (change in fast > slow-twitch fibers) and red (change in slow > fast-twitch fibers). **c** Scatter plot comparing differences in the $\log_2$-changes (POST vs. PRE) between slow- and fast-twitch fibers. The graph includes 131 fiber-type-specific (interaction) significant proteins with examples of few protein categories. **d** Overlap of exercise training-regulated proteins between whole-muscle lysate and slow-twitch muscle fibers. **e** Overlap of exercise training-regulated proteins between whole-muscle lysate and fast-twitch muscle fibers.

fibers, respectively. Overlapping for downregulated proteins was close to none. While this observation may seem surprising, human studies have demonstrated dissociations between exercise training adaptations observed in whole-muscle lysates with those observed at the single fiber level when assessed by immunoblotting[48–50]. This may be related to the extensive fiber-type-specific training response shown in this and other studies[49–52]. For example, while we observed no apparent changes in Haloacid dehalogenase-like hydrolase domain-containing protein 3 (HDHD3) expression in whole-muscle lysate (+0.18 log-fold change, significance score: 0.95), it exhibited a divergent regulatory pattern in fiber segments (Fig. 4). As such, in whole-muscle lysate, an effect in slow-twitch fibers may be opposed by a concomitant divergent change in fast-twitch fibers as also shown by others[43]. The heterogeneity between fibers and between individuals possibly also add to the relatively small overlap between fiber-type-specific and whole-muscle lysate adaptations. Indeed, proteome changes in whole-muscle lysate may be driven by changes in slow-twitch fibers in an individual with a high distribution of slow-twitch fibers and vice versa. In addition, whole-muscle lysates are likely contaminated by non-muscle cells to a greater degree than single fiber segments[43]. Last, technical variability, e.g. batch-to-batch variation from LC-MS measurements, may also contribute to discrepancies between single fiber and whole-muscle analysis. Notwithstanding, fiber-type-specific proteomics may provide a more comprehensive picture of the adaptations induced by exercise training than analysis on whole-muscle lysate.

**Fiber-type-specific effects of exercise training on proteins regulating mitochondrial content**. Expansion of mitochondria and enhancement of mitochondrial function is important for exercise training-induced improvements in muscle oxidative capacity as well as in whole-body metabolism and health[53]. Despite great interest in mitochondrial physiology, studies have so far only succeeded in determining fiber-type-specific training adaptations in individual proteins by immunoblotting[54]. Here we demonstrate extensive mitochondrial remodeling in slow- and fast-twitch fibers in response to exercise training, and interestingly, we observe substantial fiber-type-specific adaptations in proteins important for regulating mitochondrial content.

Mitochondrial proteins are encoded by nuclear DNA (nDNA) except for a few encoded by mitochondrial DNA (mtDNA). During an acute exercise bout, rapid increases in $Ca^{2+}$-cycling, ATP turnover, ROS production, and oxygen consumption activate intracellular signaling pathways leading to increased mitochondrial content[55]. This is initiated by activation of transcription factors increasing transcription of both nDNA and mtDNA. mRNA for mitochondrial nuclear-encoded proteins are translated in the cytosol to mitochondrial precursor proteins, which contain a positively charged targeting sequence at the N-terminal (transit peptide). These precursor proteins are translocated into mitochondria via the protein import machinery (the translocase of the outer mitochondrial membrane (TOM) complex and inner mitochondrial membrane (TIM) complex) (Fig. 5a)[56]. The above-mentioned mitochondrial transcription factor A (TFAM) is among the nuclear-encoded proteins (Fig. 5b). TFAM regulates transcription of mtDNA and is upregulated by exercise training[57]. Consistent with this, we observed that exercise training increased TFAM expression in both slow- and fast-twitch muscle fibers (Fig. 5b). mtDNA contains only 37 genes coding for tRNAs, rRNAs, and 13 polypeptide subunits of the oxidative phosphorylation (OXPHOS) complexes I, III, IV, and V[58]. In this study, exercise training increased expression of proteins involved in mitochondrial protein translation, as well as the expression of mtDNA-encoded proteins irrespective of fiber type (Fig. 5c, d and

Supplementary Data 10). Interestingly, the abundance of the 13 mtDNA-encoded proteins was similar between fiber types (Fig. 5d and Supplementary Data 10), suggesting that fiber-type-specific differences in expression of mitochondrial proteins are due to mitochondrial proteins encoded by nDNA and not mtDNA.

Proteins involved in regulation of cytoplasmic translation did not respond to exercise training and were similarly expressed between fiber types (Fig. 5e and Supplementary Data 10). In contrast, expression of transit peptide bearing proteins increased in response to training in both fiber types (Fig. 5f and Supplementary Data 10). Interestingly, only in slow-twitch fibers did the expression of TIM and TOM complexes increase in response to training (Fig. 5g and Supplementary Data 10). This suggests that this part of the import machinery of mitochondrial proteins is particularly important for slow-twitch fibers and their capacity for aerobic energy production.

Collectively, the capacity to increase mitochondrial content seemed to be higher in slow- than in fast-twitch fibers, but exercise training increased capacity in both fiber types although fiber-type-specific adaptations to exercise training were also evident.

**Fiber-type-specific regulation of glucose metabolism in response to exercise training**. During exercise, glucose metabolism contributes to ATP turnover in skeletal muscle[59]. In muscle, HKII rapidly phosphorylates glucose to glucose-6-phosphate (G6P), which is utilized in different metabolic pathways (Fig. 6a). As expected, total abundance of all glycolytic enzymes was expressed to a higher extent in fast- vs. slow-twitch fibers (Fig. 6b)[10]. In response to training, total abundance of glycolytic enzymes was increased in fast-twitch fibers only (Fig. 6b). When comparing the abundance of individual glycolytic proteins, only Enolase 3 was significantly upregulated with exercise training in fast-twitch fibers (Supplementary Data 11). In slow-twitch fibers, HKII was the only glycolytic protein significantly upregulated by exercise training, while it was below the quantification level in fast-twitch fibers (Supplementary Data 11).

Aside from the glycolytic pathway, G6P can enter the pentose phosphate pathway (PPP), which is controlled by the rate-limiting enzyme G6P dehydrogenase (G6PD)[60], or it can be incorporated into glycogen regulated by Glycogen Synthase (GYS1). Total abundance of enzymes of the PPP was similar between fiber types and was unaffected by exercise training (Supplementary Fig. 1f). However, we observed a higher expression of G6PD in slow- vs. fast-twitch fibers and a marked reduction (~95%) by exercise training only in slow-twitch fibers (Fig. 6c). G6PD was recently reported to be a negative regulator of muscle insulin sensitivity and increased expression in human skeletal muscle was associated with muscle insulin resistance[61]. Thus, we speculate that the dramatic decrease in G6PD expression in slow-twitch fibers may contribute to increased muscle insulin sensitivity and improved glucose uptake in this fiber type upon exercise training.

Exercise training increased GYS1 expression in both fiber types (Fig. 6d), likely contributing to the higher glycogen content after training (Fig. 1c). Conversely, expression of glycogenin (GYG1), the core protein of glycogen particles[62], was not altered by training suggesting that GYG1 is not a key factor responsible for changes in glycogen content induced by exercise training. This is in line with recent findings in GYG1 KO mice showing that GYG1 is not necessary for glycogen synthesis[63]. Training did not alter expression of glycogen phosphorylase (PYGM), the rate-limiting enzyme in glycogen degradation, yet marked fiber-type differences were evident, underlining the importance of a rapid glycogenolytic rate in fast-twitch fibers during exercise.

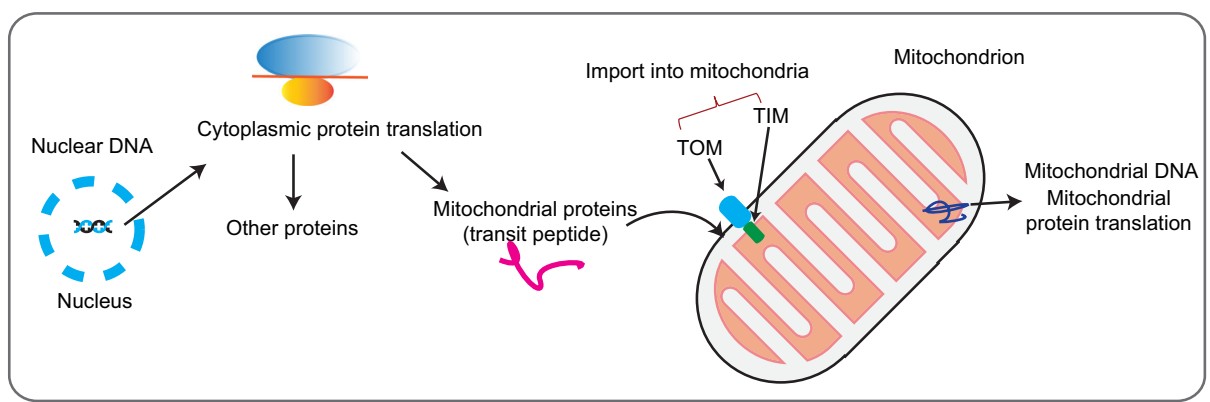

**Fig. 5 Fiber-type-specific effects of exercise training on capacity to increase mitochondrial content. a** Schematic representation of the steps involved in increasing mitochondrial content. **b** Percentage protein abundance of TFAM, **c** proteins involved in mitochondrial protein translation, **d** mitochondrial DNA-encoded proteins, **e** proteins involved in cytoplasmic protein translation, **f** transit peptide-associated proteins, **g** TIM TOM subunits. Data are shown as median ($n = 5$). $P$ values by two-way RM ANOVA followed by Tukey's post hoc test are indicated in Fig. 5b–g. "*" denotes a training effect and "†" denotes a difference between fiber types. Lines indicate main effects.

To maintain glycolytic flux, NADH needs to be reoxidized to $NAD^+$. Anaerobically, this is ensured by conversion of pyruvate to lactate by the enzyme lactate dehydrogenase (LDH) and aerobically in mitochondria facilitated by the glycerol 3-phosphate (G3P) and malate-aspartate (MA) shuttle. Exercise training increased expression of LDHB, but not LDHA (Fig. 6e) or lactate transporters (Supplementary Fig. 1e),

indicating that capacity for lactate production and release was not improved by the present exercise training program. Observations in human whole-muscle samples suggest that trained and untrained subjects have similar expressions of proteins in the G3P shuttle[64]. Our findings support this as exercise training did not affect glyceraldehyde-3-phosphate dehydrogenase 1 or 2 (GDP1, GDP2) expression in either fiber

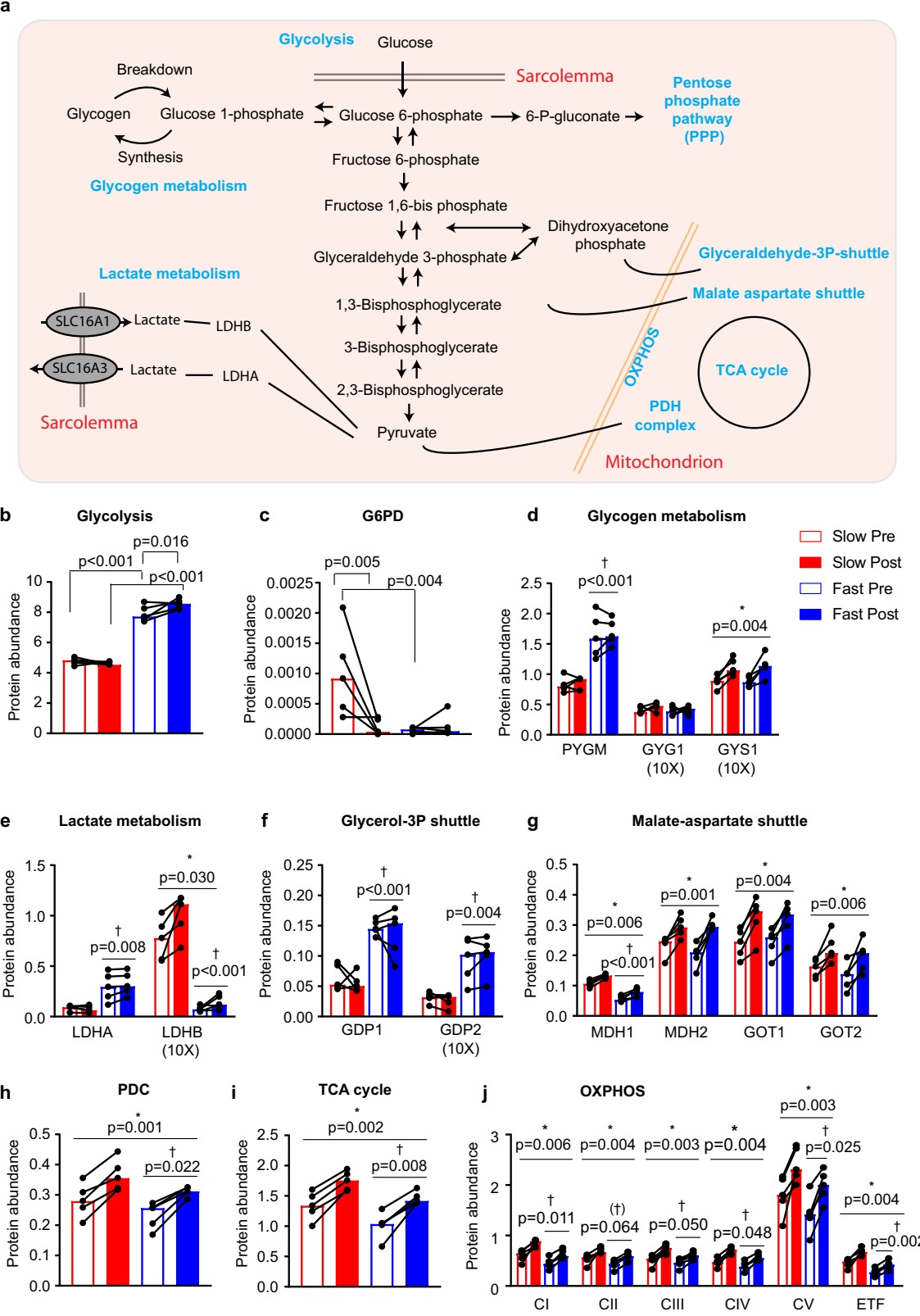

**Fig. 6 Fiber-type-specific regulation of glucose metabolism in response to exercise training. a** Schematic representation of glucose metabolism in skeletal muscle. **b** Percentage protein abundance of glycolytic enzymes, **c** glucose-6-phosphate dehydrogenase, **d** enzymes involved in glycogen metabolism, **e** lactic acid metabolism, **f** glycerol-3P shuttle, **g** malate-aspartate shuttle, **h** pyruvate dehydrogenase complex, **i** TCA cycle, and **j** OXPHOS ($n = 5$). $P$ values by two-way RM ANOVA followed by Tukey's post hoc test are indicated in Fig. 6b–j. "*" denotes a training effect and "†" denotes a difference between fiber types. Lines indicate main effects. 5× and 10× = multiplied by 5 and 10, respectively.

type (Fig. 6f). On the contrary, expression of all proteins in the MA shuttle was increased by training in both fiber types supporting previous observations in human whole-muscle samples[12,64] (Fig. 6g).

The pyruvate dehydrogenase complex (PDC) decarboxylates pyruvate to Acetyl CoA, which enters the tricarboxylic acid (TCA) cycle and OXPHOS during aerobic conditions. Studies have shown that training increases expression of proteins in the PDC, TCA cycle, and OXPHOS in whole-muscle samples[13,65]. Here we demonstrate that the total abundance of proteins in these pathways was increased in both fiber types in response to exercise training (Fig. 6h–j) although not all individual proteins were significantly upregulated (Supplementary Data 11). This suggests that the capacity for aerobic energy production adjusts to the altered demand imposed by exercise training in both fiber types.

In conclusion, we used a sequential digestion protocol, high-resolution quantitative MS, and sophisticated algorithms to generate deep muscle fiber proteomes from freeze-dried human muscle biopsies. Our methodology is easily applicable to many laboratories and to the many non-embedded snap-frozen muscle specimens already available from biobanks. From low sample amounts, we identified over 4000 proteins, representing in-depth proteomic analysis of slow- and fast-twitch muscle fibers before and after exercise training. Our fiber-type-specific proteomic analysis revealed previously unknown proteins specific to fast- and slow-twitch muscle fibers. Furthermore, we provide many examples of fiber-type-specific adaptations to training within several metabolic processes. Collectively, our results provide a deeper understanding of the adaptations to exercise training in a fiber-type-specific manner. Although recognizing that this is influenced by the type of exercise training and initial training status of the subjects, these data are important for elucidating molecular mechanisms underpinning the health- and performance-related benefits of exercise training.

## Methods

**Subjects**. Nine men gave their written, informed consent to participate in the study approved by the Regional Ethics Committee for Copenhagen (H-6-2014-038) and complied with the guidelines of the 2013 Declaration of Helsinki. Data on the effect of training on insulin sensitivity in response to acute exercise have been described elsewhere[18]. In the current study, we performed analyses on five subjects who were randomly selected from the whole cohort. Informed consent and ethical approval were also in place for current analysis. Given restraints related to instrumentation time, resources, and time to undertake single fiber isolation and subsequent MS as well as limited biological material, five subjects from the original study were selected for the present analysis. Subjects were young (24 ± 1 years, mean ± SEM), non-obese (BMI 24.2 ± 0.4, mean ± SEM), untrained (43.5 ± 2.5 ml kg$^{-1}$ min$^{-1}$, mean ± SEM), and healthy males.

**Myotubes and myoblasts**. Human muscle precursor cells were isolated from vastus lateralis muscle biopsies and differentiated in vitro as previously described[66]. Primary cells were collected on day 0 (myoblast) and day 5 (myotubes) of differentiation. The cell lysis was performed in buffer containing 4% SDS, 100 mM Tris-HCl pH 7.6, 100 mM DTT, and ddH$_2$O. Lysates were heated for 5 min at 96 °C, centrifuged at 16,000 $g$ for 10 min, and the supernatant was stored until further use. A pool of primary cells from three subjects was used for proteomics analysis. All subjects gave oral and written informed consent before inclusion and the study was performed in accordance with the Declaration of Helsinki and approved by The Regional Committee on Biomedical Research Ethics in Denmark (KF01-141/04)

**Experimental protocol**. The experimental protocol consisted of two experimental days separated by 12 weeks of endurance cycling training.

*Measurement of VO$_{2peak}$, body composition, and peak workload*. Minimum 1 week before training, peak oxygen uptake (VO$_{2peak}$) was determined by breath by breath measurements of VO$_2$ (Masterscreen CPX, IntraMedic, Denmark) during an incremental test to exhaustion on a bike ergometer (Monark, Ergomedic 839E, Sweden). Body composition was measured by dual X-ray absorptiometry (DPX-IQ Lunar, Lunar Corporation, USA). After familiarization with the one-legged knee-extensor ergometer, peak workload (PWL) of the knee-extensors was determined for both legs during a single-leg incremental test to exhaustion.

*Training regime*. The 12-week training program consisted of 4 × 1-h of indoor cycling exercise per week (Body bike supreme classic, Pedan, Denmark). The intensity of the training sessions ranged from 75–90% of maximal heart rate measured by Polar heart rate monitors (Polar CS400, Polar, Finland). Three of the four weekly training sessions were performed at the subjects' home residence, while the fourth training session was performed in the Department of Nutrition, Exercise and Sports, University of Copenhagen. Throughout the 12 weeks of training, subjects were instructed to continue their habitual diet, remain weight stable, and measure resting heart rate in the morning (3 days/week).

*Experimental days*. Subjects were instructed to record food intake for three days and to abstain from alcohol, caffeine, and strenuous physical activity 48 h before the PRE training experimental day. Subjects repeated their 3-day diet regime and abstained from alcohol, caffeine, and strenuous physical activity 48 h before the POST training experimental day.

On the morning of the experimental days, subjects arrived at the laboratory 1 h after having ingested a small breakfast (oatmeal, skimmed milk, sugar, 5% of daily energy intake). Then they performed 1 h of dynamic knee-extensor exercise, with a randomized leg, at 80% of PWL interspersed with 3 × 5 min at 100% of PWL. After exercise, subjects rested in the supine position and catheters (Pediatric Jugular Catherization set, Arrow Int., USA) was inserted into the femoral vein of both legs and in a dorsal hand vein (Venflon Pro Safety, Mediq, Denmark) for sampling of arterialized venous blood (heated hand vein). After 4 h of rest, a euglycemic hyperinsulinemic clamp (EHC) was initiated with a bolus of insulin (9 mU kg$^{-1}$, Actrapid, Novo Nordisk, Denmark) followed by 120 min of constant insulin infusion (1.4 mU min$^{-1}$ kg$^{-1}$). Blood samples were drawn simultaneously from all three catheters before (−60, −30, and 0 min) and during the EHC (15, 30, 45, 60, 80, 100, and 120 min). Before each blood sampling, femoral arterial blood flow was measured using the ultrasound Doppler technique (Philips iU22, ViCare Medical A/S, Denmark). Muscle biopsies of the vastus lateralis were obtained in both legs immediately before and after the clamp under local anesthesia (~3 ml xylocaine 2%; AstraZeneca, Denmark) using the Bergström needle technique with suction.

**Analysis of plasma samples**. Plasma glucose concentration was measured by a blood-gas analyzer (ABL800 FLEX, Radiometer, Denmark). Plasma insulin concentration was measured using an insulin ELISA kit (ALPCO, USA).

**Human skeletal muscle analyses**. In the current study, we analyzed muscle biopsies obtained in the rested leg both before and after the training period.

*Preparation of single fiber pools and whole-muscle samples for proteomics analysis*. Skeletal muscle fibers were isolated as previously described[27] with a few modifications. In total, 60–150 mg of muscle tissue was freeze-dried for 48 h before dissecting individual muscle fibers using a stereomicroscope and a fine forceps in a temperature- and humidity-controlled room (humidity of ~25%). Each fiber (length: 1.5 ± 0.4 mm, mean ± SD) was dissolved in 10 µl lysis buffer (4% SDS, 100 mM Tris-HCl pH 7.6, 100 mM DTT, and ddH$_2$O) and heated for 5 min at 96 °C. A small fraction of each solubilized fiber sample was used to identify MHC expression using dot-blotting and specific antibodies against MHC I or MHC II (anti-MHC human slow-twitch fibers (RRID:AB_528384) diluted 1:10,000 and anti-MHC human fast-twitch fibers (PRID:AB_528383) diluted 1:10,000, Developmental Studies Hybridoma Bank, Iowa, USA). One microliter (1/10 of a fiber) was spotted onto two identical PVDF membranes. One membrane was incubated overnight at 4 °C in primary antibody against MHC I and the other in primary antibody against MHC II. Membranes were incubated for 1 h at RT in secondary antibody conjugated with biotin (anti-mouse AB_2338566, Agilent Dako, CA, USA and anti-mouse AB_2687571, Jackson ImmunoResearch, PA, USA) followed by incubation in HRP-conjugated streptavidin (AB_2337238, Jackson ImmunoResearch, PA, USA) for 45 min at RT. Protein dots were visualized using enhanced chemiluminescence (Millipore) in a ChemiDoc MP imaging system (Bio-Rad, CA, USA). The remnant sample of each fiber was pooled according to MHC expression to form a sample of pooled type I and type II fibers from each biopsy (31–35 fibers/pool). Hybrid fibers (<1%) expressing two MHC isoforms were excluded. Freeze-dried whole-muscle samples (~2.5 mg dry weight) were dissolved in 300 µl of the same lysis buffer as the single fibers.

*Western blotting in fiber pools*. Additional fibers were dissected from four of the five subjects, fiber typed and pooled as described previously (section of single fiber isolation and fiber type determination) except that each fiber was solubilized in 5 µl ice-cooled Laemmli sample buffer (125 mM Tris-HCl, pH 6.8, 10% glycerol, 125 mM SDS, 200 mM DTT, 0.004% bromophenol blue). Protein concentration of the fiber pools was estimated using 4–20% Mini-PROTEAN TGX stain-free gels (Bio-Rad, CA, USA), which allowed for gel-protein imaging following UV-activation on a ChemiDoc MP imaging system (Bio-Rad, CA, USA). The intensity of the protein bands (40–260 kDa) was compared to a standard curve of mixed muscle homogenates with a known protein concentration.

To measure protein expression in the fiber pools, samples were separated on self-cast gels using SDS-PAGE followed by semidry transfer of proteins on PVDF membranes. Membranes were blocked for 5 min in 2% skimmed milk in TBS

containing 0.05% Tween-20 (TBST buffer) followed by overnight incubation at 4 °C in primary antibodies: Anti-HKII (AB_2232946, Cell Signaling Technology, MA, USA) diluted 1:500, anti-MDH2 diluted 1:700 and anti OXPHOS total cocktail (human) (AB_2756818, Abcam, Cambridge, UK) diluted 1:10,000). The following day, membranes were incubated with HRP-conjugated secondary antibodies (goat-anti-mouse-HRP (AB_2338504) and goat-anti-rabbit-HRP (AB_2337938) Jackson ImmunoResearch, USA) for 1 h at room temperature before visualizing protein bands with chemiluminescence (Millipore) and ChemiDoc MP imaging system (Bio-Rad). Some membranes were stripped and re-probed with a new primary antibody against another protein after removal of the first antibody by incubation in stripping buffer (62.3 mM Tris-HCl, 69.4 mM SDS, ddH$_2$O and 0.08% β-mercaptoethanol pH 6.7). The western blotting on fiber pools was performed on $n = 4$.

*Glycogen in fiber pools*. Glycogen content in the additional fiber pools (described above for WB) was measured by dot-blotting using a specific antibody against glycogen (diluted 1:1000) as previously described[27] (monoclonal mouse anti-glycogen recognizing the α-1.6 linkages in the glycogen particles, kindly provided by Otto Baba, Tokyo Medical and Dental University, Tokyo, Japan). Briefly, 150 ng of protein was spotted onto a PVDF-membrane. After air-drying, the membrane was re-activated in ethanol before blocking, incubation in primary and secondary antibody, and visualization as described in the section on Western blotting. The glycogen measurement in fiber pools was performed on $n = 4$.

*Whole-muscle glycogen*. Muscle biopsies were freeze-dried for 48 h and dissected free of visible blood, fat, and connective tissue. Muscle homogenates were generated as previously described[27]. Lysates were recovered by centrifuging the homogenates (20 min, 18,320 $g$, 4 °C). Homogenate and lysate protein content were determined by the bicinchoninic acid method (Pierce Biotechnology, Inc., USA). Muscle glycogen content was measured in homogenates (150 μg protein) as glycosyl units after acid hydrolysis determined by a fluorometric method as previously described[27].

*Whole-muscle MHC expression*. For MHC determination in muscle biopsies, whole-muscle lysates were diluted 1:3 with 100% glycerol/Laemmli sample buffer (50/50) and run on 8% self-cast stain-free gels containing 0.5% 2,2,2-trichloroethanol[67]. A total of 3 μg of lysate protein was separated for 16 h at 140 V as previously described[68]. Protein bands were visualized by ultraviolet activation of the stain-free gel on a ChemiDoc MP Imaging System (Bio-Rad) and quantified as described in the section on Western blotting.

*Proteomics analysis*. From human primary muscle cells, slow- and fast-twitch muscle fiber pools and whole-muscle lysates, proteins were precipitated using cold acetone (four times the volume of the sample) incubation overnight (−20 °C). Samples were centrifuged (3 × 10 min, 4 °C, 16,000 $g$) and washed in 80% cold acetone. The precipitates were dissolved in 200 μl UREA buffer (8 M UREA in 0.1 M Tris-HCl pH 8.5) and processed according to the MED-FASP (multiple enzyme digestion with filter-aided sample preparation) protocol using the endoproteinases LysC and trypsin[21]. Briefly, on a FASP filter, protein precipitates from the human muscle cells and muscle fiber pools were alkylated using iodoacetamide (IAA) and digested using LysC for 18 h at room temperature. LysC peptides were filtered and collected. Undigested proteins on the filter were washed with water and further digested using trypsin for 6 h at 37 °C. Tryptic peptides were filtered and collected in a separate tube. Both LysC and tryptic peptides were purified on C18 StageTips[69]. Digestion of whole-muscle lysate was performed with similar MED-FASP protocol. Additionally, whole-muscle lysate was prepared with conventional FASP in which samples were digested with LysC and Trypsin together.

Peptides were measured using LC-MS instrumentation consisting of an Easy nanoflow HPLC system (Thermo Fisher Scientific, Bremen, Germany) coupled via a nanoelectrospray ion source (Thermo Fischer Scientific, Bremen, Germany) to a Q Exactive HF mass spectrometer[22] or Q Exactive HF-X mass spectrometer. Purified peptides were separated on 50 cm C18 column (inner diameter 75 μm, 1.8 μm beads, Dr. Maisch GmbH, Germany). Peptides were loaded onto the column with buffer A (0.5% formic acid) and eluted with a 180 min linear gradient from 2–60% buffer B (80% acetonitrile, 0.5% formic acid). After the gradient, the column was washed with 90% buffer B and reequilibrated with buffer A. Mass spectra were acquired in a data-dependent manner with automatic switching between MS and MS/MS using a top 15 method. MS spectra were acquired in the Orbitrap analyzer with a mass range of 300–1750 $m/z$ and 60,000 resolutions at $m/z$ 200. HCD peptide fragment acquired at 27 normalized collision energy were analyzed at high resolution in the Orbitrap analyzer.

Raw MS files were analyzed using MaxQuant version 1.5.2.8 (ref. [20]) (http://www.maxquant.org). MS/MS spectra were searched by the Andromeda search engine (integrated into MaxQuant) against the decoy UniProt human database with forward and reverse sequences. In the main Andromeda search precursor, mass and fragment mass were matched with an initial mass tolerance of 6 and 20 ppm, respectively. The search included variable modifications of methionine oxidation and N-terminal acetylation and fixed modification of carbamidomethyl cysteine. Minimal peptide length was set to seven amino acids, and a maximum of

two miscleavages was allowed. The FDR was 0.01 for peptide and protein identifications. MS runs from skeletal muscle fibers and human muscle cells were analyzed with the "match between runs" option. This feature uses accurate LC retention time alignments to transfer identification from a peptide library obtained from human primary muscle cells, enabling successful fragmentation and identification of low-abundant muscle peptide and proteins more likely. For matching, a retention time window of 30 s was selected. When all identified peptides were shared between two proteins, results were combined and reported as one protein group. Matches to the reverse database were excluded. Protein quantification was based on the MaxLFQ algorithm integrated into the MaxQuant software[19].

**Quantification and statistical analysis**. Data files generated by MaxQuant were analyzed in Perseus software[70]. A protein was considered identified only if it was detected at least three times in at least one group (slow- or fast-twitch fibers). The percentage abundance of AMPK subunits and MHCs (Fig. 1f, g) was calculated using the unique peptide protein quantification. Categorical annotation was supplied in the form of Keywords (UniProt), Gene Ontology (GO) biological processes (BP), molecular functions (MF) and cellular components (CC), as well as participation in the KEGG pathway. All annotations were extracted from the UniProt database. For comparative proteome analysis, we used normalized intensities (LFQ). The normalized intensities are obtained using MaxLFQ algorithm in MaxQuant software[19]. MaxLFQ algorithm performs on par with labeling approaches[19] but eliminates a lot of identified proteins during the process. Therefore, in our dataset, we had ~2200 proteins/groups for quantitative comparisons. For the comparative analysis, we included proteins that were quantified at least three times in at least one group (Groups: slow PRE and fast PRE, slow PRE and slow POST, fast PRE and fast POST). The data were imputed to fill missing abundance values by drawing random numbers from a Gaussian distribution with a standard deviation of 30% in comparison to the standard deviation of measured protein abundances, and one standard deviation downshift from the mean. These parameters have been tuned to best simulate the distribution of low-abundant proteins. These values are universally applied to nearly all datasets that were generated with the LFQ algorithm. In Fig. 2, the PCA was performed on imputed values. To investigate fiber-type-specific adaptations to exercise training, fold change (POST vs. PRE) between slow- and fast-twitch fibers was compared. Proteins, which were exclusively quantified in either slow- or fast-twitch fibers, were excluded while analyzing fiber-type-specific adaptations.

To estimate protein-specific differences between muscle fiber types and fiber-type-specific adaptations to training, a series of linear models were performed on the proteome data in a paired design. The essence of the analysis was to estimate (1) fiber-type differences in the proteome signature at baseline (PRE), (2) within-fiber adaptations to training, and (3) between-fiber adaptations to training (i.e. fiber-type-specific adaptations). To identify and rank differentially expressed proteins, we chose a novel posteriori information fusion scheme, combining the biological relevance (fold change) and the statistical significance ($p$ value) as recently described by Xiao et al.[28]. The fusion scheme yields a protein significance score ($\pi$ value) and is robust in selecting differentially expressed genes/proteins while protecting against false discoveries. For this study, a significance score cut-off of 0.05 was selected.

Data presented in Figs. 2, 5 and 6 include analysis of summed protein abundance for various protein categories or abundance for individual protein abundance. For data in Fig. 2f, g, fiber-type differences were evaluated with a linear model. For data in Figs. 5 and 6, fiber-type-specific effects of exercise training were evaluated by a two factor repeated measures ANOVA with "Fiber type" (slow and fast) and "Exercise" (PRE and POST) as main effects for a full factorial design. $P$ values ≤0.05 were considered significantly regulated. Significant interactions were evaluated by Tukey's post hoc test (Figs. 5 and 6). Data are presented as median and $N = 5$, unless otherwise stated. Hierarchical clustering of significantly different proteins was performed after $z$-score normalization. Clustering was performed on median $z$-score values for each group. Fisher's exact test (FDR < 0.02) was performed on particular clusters, testing for enrichment or depletion of any annotation term in the cluster compared to the whole matrix.

Due to difficulties in isolating and typifying single muscle fibers, limited availability of the biopsies material, and limitations on the LC-MS instrumentation time, all experiments were performed only once for five subjects under PRE and POST conditions.

**Protein abundance**. Protein abundance was calculated based on LFQ values and expressed as percentage protein abundance. Briefly, the mass spectrometric signals of individual proteins (LFQ intensities) were divided by the summed LFQ intensities of all proteins and expressed as percentage protein abundance. The percentage protein abundance for protein categories (e.g. GOCC mitochondria) was calculated by the sum of LFQ intensities of proteins annotated to the respective category. Similarly, the percentage abundance of glycolysis, PPPs, PDC, TCA cycle, OXPHOS subunits, TIM TOM complexes, mitochondrial DNA-encoded proteins, cytoplasmic and mitochondrial protein translation was calculated by the sum of LFQ intensities of proteins annotated to the respective processes/complexes.

**Cluego network analysis**. To identify enriched biological processes within the set of proteins regulated by exercise training (237 proteins in slow-twitch fibers and 172 proteins in fast-twitch fibers), we used the Cytoscape[71] plugin ClueGO[41], which simplifies the biological interpretation of large lists of proteins. The enrichment was performed using annotations from all evidence levels of the Gene Ontology (GO) biological processes, calculated using the hypergeometric test and FDR corrected for multiple testing (Benjamini–Hochberg; $p \leq 0.05$). We set the list of all identified proteins in the experiment as background to reduce noise in the resulting set of enriched processes. The enriched terms are presented as a simplified network of functionally clustered biological processes following the GO structure. We excluded the categories containing enrichment of single biological process. Details of ClueGo enrichment network analysis with $p$ values are included in Supplementary Data 8.

**Calculations**. Leg glucose uptake was calculated using Fick's principle by multiplying blood glucose concentration A–V difference with femoral arterial blood flow (measured by using the ultrasound Doppler technique (Philips iU22, ViCare Medical A/S, Denmark)) and divided by lean leg mass measured by dual X-ray absorptiometry (DPX-IQ Lunar, Lunar Corporation, USA).

**Reporting summary**. Further information on research design is available in the Nature Research Reporting Summary linked to this article.

## Data availability
Data are processed and analyzed with freely available MaxQuant and Perseus software. Tandem mass spectra were searched against the Uniprot human databases (Version June 2015). The mass spectrometry proteomics data have been deposited at the ProteomeXchange Consortium (http://proteomecentral.proteomexchange.org) via the PRIDE partner repository with the dataset identifier PXD012824. All relevant proteomics data are available from the authors.

Source data are provided with this paper.

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

# ARTICLE

43. Murphy, R. M. & Lamb, G. D. Important considerations for protein analyses using antibody based techniques: down-sizing western blotting up-sizes outcomes. *J. Physiol.* **591**, 5823–5831 (2013).

44. Bensaad, K. et al. TIGAR, a p53-inducible regulator of glycolysis and apoptosis. *Cell* **126**, 107–120 (2006).

45. Pedersen, B. K. & Febbraio, M. A. Muscles, exercise and obesity: skeletal muscle as a secretory organ. *Nat. Rev. Endocrinol.* **8**, 457–465 (2012).

46. Meri, S. et al. Human protectin (CD59), an 18,000-20,000 MW complement lysis restricting factor, inhibits C5b-8 catalysed insertion of C9 into lipid bilayers. *Immunology* **71**, 1–9 (1990).

47. Gonçalves, F. G. P. et al. Immunohistological analysis of CD59 and membrane attack complex of complement in muscle in juvenile dermatomyositis. *J. Rheumatol.* **29**, 1301–1307 (2002).

48. Perry, B. D. et al. Dissociation between short-term unloading and resistance training effects on skeletal muscle Na+,K+-ATPase, muscle function, and fatigue in humans. *J. Appl. Physiol.* **121**, 1074–1086 (2016).

49. Skovgaard, C. et al. Effect of speed endurance training and reduced training volume on running economy and single muscle fiber adaptations in trained runners. *Physiol. Rep.* **6**, e13601 (2018).

50. Wyckelsma, V. L. et al. Intense interval training in healthy older adults increases skeletal muscle [3H]ouabain-binding site content and elevates Na+, K+-ATPase α2 isoform abundance in Type II fibers. *Physiol. Rep.* **5**, e13219 (2017).

51. Christiansen, D. et al. Cycling with blood flow restriction improves performance and muscle K + regulation and alters the effect of anti-oxidant infusion in humans. *J. Physiol.* **597**, 2421–2444 (2019).

52. MacInnis, M. J. et al. Superior mitochondrial adaptations in human skeletal muscle after interval compared to continuous single-leg cycling matched for total work. *J. Physiol.* **595**, 2955–2968 (2017).

53. Lanza, I. R. & Nair, K. S. Muscle mitochondrial changes with aging and exercise. *Am. J. Clin. Nutr.* **89**, 467S–471S (2009).

54. Christensen, P. M. et al. Unchanged content of oxidative enzymes in fast-twitch muscle fibers and VO2 kinetics after intensified training in trained cyclists. *Physiol. Rep.* **3**, e12428 (2015).

55. Hood, D. A., Tryon, L. D., Carter, H. N., Kim, Y. & Chen, C. C. W. Unravelling the mechanisms regulating muscle mitochondrial biogenesis. *Biochem. J.* **473**, 2295–2314 (2016).

56. Pfanner, N. & Meijer, M. Mitochondrial biogenesis: the Tom and Tim machine. *Curr. Biol.* **7**, R100–R103 (1997).

57. Bengtsson, J., Gustafsson, T., Widegren, U., Jansson, E. & Sundberg, C. J. Mitochondrial transcription factor A and respiratory complex IV increase in response to exercise training in humans. *Pflug. Arch. Eur. J. Physiol.* **443**, 61–66 (2001).

58. Anderson, S. et al. Sequence and organization of the human mitochondrial genome. *Nature* **290**, 457–465 (1981).

59. Richter, E. A. & Hargreaves, M. Exercise, GLUT4, and skeletal muscle glucose uptake. *Physiol. Rev.* **93**, 993–1017 (2013).

60. Kletzien, R. F., Harris, P. K. & Foellmi, L. A. Glucose-6-phosphate dehydrogenase: a 'housekeeping' enzyme subject to tissue-specific regulation by hormones, nutrients, and oxidant stress. *FASEB J.* **8**, 174–181 (1994).

61. Lee-Young, R. S. et al. Glucose-6-phosphate dehydrogenase contributes to the regulation of glucose uptake in skeletal muscle. *Mol. Metab.* **5**, 1083–1091 (2016).

62. Lomako, J., Lomako, W. M. & Whelan, W. J. Glycogenin: the primer for mammalian and yeast glycogen synthesis. *Biochim. Biophys. Acta* **1673**, 45–55 (2004).

63. Testoni, G. et al. Lack of glycogenin causes glycogen accumulation and muscle function impairment. *Cell Metab.* **26**, 256–266.e4 (2017).

64. Schantz, P. G., Sjoberg, B. & Svedenhag, J. Malate-aspartate and alpha-glycerophosphate shuttle enzyme levels in human skeletal muscle: methodological considerations and effect of endurance training. *Acta Physiol. Scand.* **128**, 397–407 (1986).

65. LeBlanc, P. J., Peters, S. J., Tunstall, R. J., Cameron-Smith, D. & Heigenhauser, G. J. F. Effects of aerobic training on pyruvate dehydrogenase and pyruvate dehydrogenase kinase in human skeletal muscle. *J. Physiol.* **557**, 559–570 (2004).

66. Henriksen, T. I. et al. Dysregulation of a novel miR-23b/27b-p53 axis impairs muscle stem cell differentiation of humans with type 2 diabetes. *Mol. Metab.* **6**, 770–779 (2017).

67. Ladner, C. L., Yang, J., Turner, R. J. & Edwards, R. A. Visible fluorescent detection of proteins in polyacrylamide gels without staining. *Anal. Biochem.* **326**, 13–20 (2004).

68. Kohn, T. A. & Myburgh, K. H. Electrophoretic separation of human skeletal muscle myosin heavy chain isoforms: the importance of reducing agents. *J. Physiol. Sci.* **56**, 355–360 (2006).

69. Rappsilber, J., Ishihama, Y. & Mann, M. Stop And Go Extraction tips for matrix-assisted laser desorption/ionization, nanoelectrospray, and LC/MS sample pretreatment in proteomics. *Anal. Chem.* **75**, 663–670 (2003).

70. Tyanova, S. et al. The Perseus computational platform for comprehensive analysis of (prote)omics data. *Nat. Methods* **13**, 731–740 (2016).

71. Shannon, P. et al. Cytoscape: a software environment for integrated models of biomolecular interaction networks. *Genome Res.* **13**, 2498–2504 (2003).

72. Clark, M. G. et al. Blood flow and muscle metabolism: a focus on insulin action. *Am. J. Physiol. Endocrinol. Metab.* **284**, E241–E258 (2003).

## Acknowledgements

We acknowledge the skilled technical assistance of Betina Bolmgren and Irene B. Nielsen (University of Copenhagen). We thank Ben Stocks for the critical reading and fruitful discussions.The study was supported by grants from the Danish Council for Independent Research Medical Sciences (FSS 6110-00498B/FNU 8021-00072B), The Novo Nordisk Foundation (NNF16OC0023046, NNF17OC0027224, and NNF15CC0001), The Lundbeck Foundation (R266-2017-4358), The research program (2016) Physical Activity and Nutrition for Improvement of Health funded by the University of Copenhagen Excellence Program for Interdisciplinary Research. S.D.E was supported by a research grant from the Danish Diabetes Academy funded by the Novo Nordisk Foundation. The Centre for Physical Activity Research (CFAS) is supported by a grant from TrygFonden. During the study period, the Centre for Inflammation and Metabolism (CIM) was supported by a grant from the Danish National Research Foundation (DNRF55). "Novo Nordisk Foundation Center for Basic Metabolic Research is an independent Research Center, based at the University of Copenhagen, Denmark and partially funded by an unconditional donation from the Novo Nordisk Foundation (NNF18CC0034900).

## Author contributions

Conception and design of research: A.S.D., D.E.S. and J.F.P.W. Performed experiments: D.E.S., C.C.S., B.K., E.A.R. and J.F.P.W. Performed analyses: A.S.D., D.E.S., M.H., J.B.B. and J.K.L. Interpreted results: all. Drafted the manuscript: A.S.D., D.E.S., J.B.B. and J.F.P.W. Edited and revised manuscript: all. Read and approved final version: all.

## Competing interests

The authors declare no competing interests.
