## [Peer Review File · Nature Communications]

Reviewers' Comments:

Reviewer #1:

Remarks to the Author:

Deep muscle-proteomic analysis of freeze-dried human muscle biopsies 1 reveals extensive fiber 2 type-specific adaptations to exercise training

Authors performed an extensive study on the effects of exercise training on the proteome of different skeletal muscle fiber types. Different fiber types exhibit type-specific adaptive responses to training and especially metabolic pathways and mitochondria seemed to be affected.

Overall the manuscript is very well written and understandable and the topic of this study is absolutely interesting for the scientific community. Some and major issues need to be clarified and improved within the manuscript before publication in Nat Comm. (more details see below).

Study design:

In general the number of samples/replicates is low making the identification of statistically relevant proteins very difficult. Was any analysis done assessing the interspecific variability of the muscle samples? Why were only 5 persons analysed in this study? Why weren't all subjects used for proteomic analysis? Moreover, verification of observed differential proteins (by e.g. quantitative western blotting or immunohistochemistry) is missing which is absolutely essential in order to assess the significance of the data. This data should be generated and included for at least a limited number of proteins described and discussed in detail.

Why was cardio training chosen and not weight lifting? Weight lifting might result in stronger effects on fiber type level.

Why T1 and T2a fibers were specifically assessed? It would have been beneficial to additionally investigate T2x fibers as they are the fastest fiber type in human skeletal muscles and are expected to be mainly affected by training.

If I understand correctly, biopsies were taken from the rested leg which was not used for the dynamic knee extensor exercise. Why? It would have made more sense to take biopsy from the non-rested leg? Results are not comparable if biopsies for the single fiber analysis were taken from the rested leg and measurements of whole muscle lysate MHC expression was performed on all biopsies.

General points:

In former studies the authors published contradictory results and should please comment on that/ take this into account in this manuscript:

e.g. in DOI 10.15252/embr.201439757: "OXPHOS protein levels were especially abundant in 2A fibers, in accordance with the greatest content of mitochondria among the subtypes (Supplementary Fig S6)". In <http://dx.doi.org/10.1016/j.celrep.2017.05.054> "...and clearly shows that respiratory chain proteins are more abundant in type 1 than in type 2A fibers (Figure S3A), a known feature of human muscle".

Several sports studies in combination with proteomics have been carried out and published in the past before and none is mentioned/referenced e.g. in the introduction. More information about former relevant studies should be referenced (e.g. page 5, line 13-16). Moreover proteomic analysis of freeze dried muscle tissue is not new and has been done before also on the fiber type level (e.g. doi: 10.1007/s00401-016-1592-7).

Specific points:

Page 7, line 9-12: How many protein groups can be detected without using the match between run option? This option is useful for overall analysis but it does not really reflect the true number of identified proteins.

Page 9, line 15-18: Which protein groups belong to the category "Signal"? Information is really vague and not is not sufficiently informative. A Supplemental table should be provided including protein group lists of the clusters and the respective GO term and pathway enrichment analysis.

Page 10, line 1: Fold change is even higher, because the data provided is log2 transformed. Non log transformed fold changes should be added, because they are better understandable for the reader.

Page 10, line 2: Was FHL1 also significantly regulated? If not was it at least identified in the data set? Information should be added here.

Page 11, line 1: how was this calculation done in detail?

Page 11, line 3: In previous publications (see above) the effect was observed exactly the other way around. Please comment on that.

Page 11, line 14-16: How many proteins were found to be differential after p-value correction?

Page 12, line 1-3: Pearson correlation of data (which was also carried out for the FT) would provide a more comprehensive information if no changes were found due to measuring whole muscle lysate or due to a high variability among the replicates.

Page 13, line 1-5: Are there more proteins of the glycogen pathways identified as being differential?

Page 13, line 9: SRP14 and SRP9 are forming a complex. Was SRP9 also identified or other interactors? How many peptides were identified? If no other interactors could be identified this argument is really vague.

Page 15, line 10-13: Not conclusive. Well-known markers of muscle damage are the proteins XIRP-1 and XIRP-2. Were they also found to be enriched in the samples?

Page 15, line 13-15: Only cardio training was carried out as described in the material and methods part. These conclusions have been made several times before in sports studies. References are missing.

Figure 5D/G: This figures are not useful at all because several proteins were compiled without getting any information how many: how many of the 37 mitochondrial encoded proteins could be identified in the data set and how many of them were compiled for this figure?

Page 19, line 4-6: Reference is from 1976. Aren't there any actual studies?

Page 19, line 9: Reference is missing

Page 20, line 15-17: Aren't there any actual studies?

Page 21, lines 6-7: Similar approaches had been done before on the fiber type level (see above).

Page 22, line 7-8: This number only refers to "match between run" option and does not reflect the number of truly identified proteins.

In summary, despite the important question and the beautifully readable form of the work, extensive changes and additional analysis are necessary before the quality of the manuscript is sufficient to allow publication in Nat Comm.

Katrin Marcus

Reviewer #2:

Remarks to the Author:

The manuscript by Deshmukh et al reports comprehensive proteomic analysis of 2 main classifications (type I and type II) of fibres in human skeletal muscle. The extraction of myofibres from freeze-dried muscle has been reported previously, as has proteomic analysis of individual myofibre types. Therefore the novel aspect of the current work is the analysis of samples taken prior to and after a 12 wk period of endurance training. Muscle samples were used from a previously published study (Steenberg et al 2019, J Physiol vol 597) that investigated the effects of chronic exercise training on muscle insulin sensitivity. The current manuscript reports new data on differences in protein abundance between Type I and Type II fibres at baseline, and also fibre-type specific responses to exercise training. The manuscript presents useful information on the muscle response to exercise and the collection of data is technically excellent.

Given the focus on exercise, it is surprising that the work is entirely devoid of citations to past proteomic studies of human muscle responses to training (this field began 10 years ago). The current work represents a significant advance in the field but it is highly unusual that no attempt is made to link the data to past proteomic data.

The manuscript strongly emphasises the importance of new fibre-type-specific data but this is overstated. The majority (558) of changes in protein abundance in response to exercise were not specific to a particular fibre type, and < 20 % (121) of proteins were regulated in a fibre type-specific manner. This does not seem to match the title "...extensive fibre type-specific adaptation..." Moreover, the authors do not present a comparison of proteomic analysis of whole muscle homogenates (i.e. typical approach) versus their new fibre type-specific workflow. Therefore it is not possible to measure the worth of the additional fibre-specific information. It might also be debated whether a fibre specific approach to muscle analysis is a worthwhile endeavour when performance of the whole muscle is of interest in exercise physiology. The interpretation of fibre type-specific data could easily be confounded by cooccurring changes in myofibre profile or selective myofibre hypertrophy depending on the training regimen. Similarly, it is difficult to know the consequences of motor unit activation pattern (i.e. relating to exercise intensity) on the interpretation of the proteomic data. Sub-dividing muscle in to 2 of the 3 fibre types and reporting the data in isolation also seems somewhat counter to the holistic philosophy of proteomics. None of these issues are considered in the authors discussion and in my opinion the presentation of the work lacks balance. The authors mention that by-design their work does not consider changes in fibre phenotype but they do not discuss the consequences of this limitation. For example, there is no mention of the known multiplicity of myofibre types (i.e. fibres that co-express 2 or more myosin heavy chain isoforms) and they do not discuss why they have not also analysed type IIx fibres, which in their cohort make up almost 20 % of the muscle. N.B. It is unusual that there were no changes in myosin heavy chain profile given the age and starting baseline of the participants, and the duration of the training intervention. One of the aims in performing proteomic analysis of isolated myofibre types was to investigate muscle fibre responses in the absence of contaminating fibroblast proteins. The manuscript does not report data on whether this aim was achieved. Again, a comparison of the current data against typical whole muscle analysis could give a measurable outcome to this aim. Lastly, the majority of proteins that were responsive to exercise were mitochondrial. Muscle mitochondria can be distinguished in to 2 sub-cellular populations (subsarcolemmal and intramyofibrillar) that exhibit different responses to exercise. It is a limitation of the current work that mitochondrial sub-populations were not also analysed, or that this aspect of the muscle response is not acknowledged.

Reviewer #3:

Remarks to the Author:

Major:

Imputation was applied for missing values. This leads to effects as described in the following example of a significant hit in table S6 and S7: MICU2 in the 2-Pre group has 4 missing values (table s2). After imputation these 4 missing points are higher than the one that was determined. In the 2-Post group 3 values are missing. This hit was reported as significant in both tables. The authors should consider to use a different strategy to replace missing values. It is crucial for a dataset with only 5 samples to mention how many missing values are present or to mark the replaced missing values in the dataset. The authors use paired tests. Please indicate the corresponding pairs in the tables, otherwise it is impossible to check the statistics. In the tables, please label the headers in a way consistent with the text.

Which calculations/normalization were done between suppl.table2 and suppl.table6. Can this explain why for the same sample number some values of one protein increases but for another it

decreases?

The authors used an alignment function to increase the identification rate. This is a common function in software for label-free quantification. The MS instrument is standard equipment in many facilities. Since decades it is also known that combination of proteases is beneficial for the identification, but each additional protease requires additional analysis time. What is the novelty of the presented workflow? How does it compare with other standard workflows that use the same time for analysis on comparable instruments. Any comparison with existing methods is missing. When big numbers of identifications are mentioned, it would be very beneficial to the readers to also mention how many proteins were identified in at least half of the samples or in above 90% of all samples to get an idea on the quality of the dataset. How many of the identified proteins can be quantified with this data analysis strategy, with a standard strategy or other standard methods. What is the identification rate without the addition of the myoblast and myotube samples?

The data is based on 5 subjects before and after training. To judge paired statistics it is important to mention which results are the pairs. It is not possible for the reader to check the statistics as this labeling is missing in the supplementary tables. Multiple testing correction is completely missing and would benefit the data analysis. The authors do not mention, at which fold change their method is valid. Fold change differences for the exercise dataset would be beneficial to judge if a significant hit has an impact. A volcano plot with fold change versus p-value for the exercise results gives direct information on the major changes. Please add such a plot for both fiber types to the main figures.

The reporting of the results is often misleading. When changes are reported it is expected that these changes are significant. The manuscript describes a lot of proteins that are increased or decreased upon training. Some of them are not present in the dataset containing the significant changes (suppl. Tables) or could not be found by the reviewer. The authors need to check all proteins mentioned in the manuscript for their significance and if none significant proteins are described to change they need to mention that these proteins are not significantly changing and need to mention the reason why they still believe that these changes are possibly true. Especially for the chapter about glucose metabolism.

For instance on page 18 the importance of changing HK2 is discussed, but HK2 is not in the suppl table 6 or 7 summarizing the significant changes and in the raw dataset (suppl. Table 2) 5 out of the 20 values are missing. Can the described differences be explained by missing values due to low abundance of the protein? Is it significantly changing? What is the significance level?

In addition, why are the authors not mentioning the by far more abundant HK1 (see their raw data) and also do not mention it in the graphs? Does the claim p19 line5 "we observed that training increased expression of all glycolytic enzymes ..." still hold when HK1 is included? What about other identified glycolytic enzymes? ALDOC was identified but also not mentioned. Does the claim still hold? The authors need to check all glycolytic enzymes when they claim all glycolytic enzymes and not just pick one isoform, and if they do they need to explain why they exclude the others. In the related figure S1a a lot of very small fold changes probably between 0.8-1.2 are reported. Are these changes all significant? Please indicate which are significant and which not.

As reported by Hood et al 2016, exercise leads to changes in mitochondrial biogenesis. Such a change would probably result in abundance changes of the majority of mitochondria related proteins and metabolite requirements. This knowledge in respect of the findings is crucial. The authors should discuss if such a remodeling of the cell explains all detected protein changes and which changes are independent of mitochondrial biogenesis. Is this dataset describing a known effect of training, but is a detailed description of mitochondrial biogenesis on the protein level?

In general for all figures containing box or bar plots. As the sample number is 5 or less please add the individual points instead or in addition to boxes or bars.

Minor

In the summary the authors mention in line13-14"... it altered expression levels of proteins involved in transcription, post-translational modifications, ..." but post-translational modifications were not mentioned to be regulated in the text. Please remove PTMs or describe in more detail, which PTMs were analyzed.

To judge paired statistics it is important to mention which results are the pairs. It is not possible for the reader to check the statistics as this labeling is missing.

The dot plots about the results from the glycogen measurements are missing, please add as panel in the main figure or as supplementary figure. What was the target of the glycogen antibody; glycogenin or the sugar? Please describe the antibody in more detail in the methods section. There are no statistical measurements for the glycogen analysis of the 4 measurements. Please do the calculations and mention that n=4 in the text. Please also add the results of the controls or standards that were used.

Page7 line5 "...we used a stringent FDR..." . A FDR of 1% is standard in proteomics analysis.

The purity of the fibers are tested by antibodies and the proteomics results. When comparing these results it would be beneficial to use a common nomenclature, like the gene name. Please further define the antibodies used in the materials section.

On page 8 line 11 "Thus, our fiber isolation procedure and proteomic workflow yielded detailed coverage of metabolic..." Up to this point no detailed coverage of metabolic features resulting from the proteomic workflow has been mentioned to come to this conclusion. Please clarify which features did the authors mean.

Page10 line2 The authors find an enrichment of PDLIM1 and discuss the potential interaction with FHL1. FHL1 is not changing or very little. Can the authors implement this finding in the discussion and discuss why one of the interaction partner is eight fold enriched but the other not.

Fig2F and G Please mention in detail in the methods section how the results were calculated and which statistics were used. The number of proteins per group n=... would be beneficial in the figure and an indication how the abundance was calculated in the legend; median or average or sum. If the sum was used, how was it normalized if the number of proteins in the different groups was different.

P12 line16 What is the fold change of CARM1. The z-score (fig 3) removes the variance, please add the fold difference, p-values and add the multiple testing corrected p-value to the figure3 c-g. Box plots out of only max.5 data points are often misleading. The individual data points should be added to the box plots or replace the box plots by the individual points.

P34 line 18 Did the authors really use a C10 column?

Point-by-point response to Reviewers' comments:

Reviewer #1

Deep muscle-proteomic analysis of freeze-dried human muscle biopsies 1 reveals extensive fiber 2 type-specific adaptations to exercise training

Authors performed an extensive study on the effects of exercise training on the proteome of different skeletal muscle fiber types. Different fiber types exhibit type-specific adaptive responses to training and especially metabolic pathways and mitochondria seemed to be affected.

Overall the manuscript is very well written and understandable and the topic of this study is absolutely interesting for the scientific community. Some and major issues need to be clarified and improved within the manuscript before publication in Nat Comm. (more details see below).

Study design:

In general the number of samples/replicates is low making the identification of statistically relevant proteins very difficult. Was any analysis done assessing the interspecific variability of the muscle samples? Why were only 5 persons analysed in this study? Why weren't all subjects used for proteomic analysis?

We thank the reviewer for this valid comment. Given the instrumentation time and resources to undertake single fiber isolation and subsequent mass-spectroscopy, we chose to select 5 random subjects from the original study. The sample size chosen is in accordance with that normally used in muscle proteomics (Murgia *et al.*, 2017) (n=4 in each group), (Holloway *et al.*, 2009) (n=5), (Hoffman *et al.*, 2015) (n=4), (Rakus *et al.*, 2015) (n=4). Since these are human samples, inter-subject variability is bound to be there but this was not analyzed prior to this study.

Moreover, verification of observed differential proteins (by e.g. quantitative western blotting or immunohistochemistry) is missing which is absolutely essential in order to assess the significance of the data. This data should be generated and included for at least a limited number of proteins described and discussed in detail.

We have now performed Western blotting on some of the proteins discussed in the manuscript (Figure S1C+D). Due to limited material, Western blotting was performed for a selected number of proteins.

Why was cardio training chosen and not weight lifting? Weight lifting might result in stronger effects on fiber type level.

Because the study was performed using biopsies from a former study, it was not a matter of a choice between cardiorespiratory training and strength training. We agree with the reviewer that the outcome would likely have been different if another training regime was performed; however, we already recognize this in our conclusion (page 21).

Why T1 and T2a fibers were specifically assessed? It would have been beneficial to additionally investigate T2x fibers as they are the fastest fiber type in human skeletal muscles and are expected to be mainly affected by training.

We agree that it would have been interesting to investigate the response in a pool of type IIx fibers. The reason for not doing so was solely due to practical reasons. Given the heterogeneity of human skeletal muscle, we had to dissect and isolate around 160 fibers just to obtain a pool of minimum 30-35 type I and type II fibers. Because human vastus lateralis muscle only contains a few percent of type IIx fibers, it would have taken upwards of 500 fibers to obtain a pool of around 30 type IIx fibers – for some individuals even upwards of 1000 fibers, hence requiring a substantial amount of work and biopsy material.

If I understand correctly, biopsies were taken from the rested leg which was not used for the dynamic knee extensor exercise. Why? It would have made more sense to take biopsy from the non-rested leg? Results are not comparable if biopsies for the single fiber analysis were taken from the rested leg and measurements of whole muscle lysate MHC expression was performed on all biopsies.

We think this may related to a misunderstanding. In the original study (Steenberg *et al.*, 2019), subjects performed an acute bout of single-leg knee-extensor exercise before and after 12 weeks of endurance exercise training. The exercise-training regime consisted of indoor cycling with both legs. Thus, we ended up with biopsies from a rested untrained leg, an acutely exercised untrained leg, a rested trained leg, and an acutely exercised trained leg.

For the present study, we chose to measure the response to training in the rested leg that was not confounded by the acute bout of exercise to get the pure effect of training.

As the reviewer suggest, we have included the data on the MHC expression of the rested leg only (figure 3C). In the revised manuscript, we have also performed proteome analysis of whole muscle samples from the same subjects. The conclusion is somewhat similar with no change in MHC I and IIa expression but with a minor significant decrease in IIx fibers.

General points:

In former studies the authors published contradictory results and should please comment on that/ take this into account in this manuscript: e.g. in DOI 10.15252/embr.201439757: "OXPHOS protein levels were especially abundant in 2A fibers, in accordance with the greatest content of mitochondria among the subtypes (Supplementary Fig S6)". In <http://dx.doi.org/10.1016/j.celrep.2017.05.054> "...and clearly shows that respiratory chain proteins are more abundant in type 1 than in type 2A fibers (Figure S3A), a known feature of human muscle".

The studies the reviewer cites were performed in non-human species. DOI 10.15252/embr.201439757 was performed in mice, in which the mitochondrial content is higher in type IIa fibers. In contrast, mitochondrial content is higher in type I fibers in human skeletal muscle. Please refer to the paper by Gouspillou et al 2014,

Plos One (doi: [10.1371/journal.pone.0103044](https://doi.org/10.1371/journal.pone.0103044)), Schiaffino *Acta Physiologica* 2010 (DOI:[10.1111/j.1748-1716.2010.02130.x](https://doi.org/10.1111/j.1748-1716.2010.02130.x)) for the species differences in mitochondrial content of the muscle fiber types. Rodent muscles have four fiber types (1, 2A, 2X and 2B), while human muscles only have three (1, 2A, 2X), as MyHC2B-MYH4 is essentially not expressed in human muscle (Andersen JL et al, *Acta Physiologica* 1994, DOI:[10.1111/j.1748-1716.1994.tb09730.x](https://doi.org/10.1111/j.1748-1716.1994.tb09730.x)). In addition, the same fiber type has different physiological properties, including amount of mitochondria, in rodents and humans. In mouse, our previous study in single muscle fibers shows that 2A fibers contain more mitochondria than type I fibers, in agreement with previous reports from mouse and rat muscles (Bloemberg & Quadrilatero, *PLoSone* 2012, doi: [10.1371/journal.pone.0035273](https://doi.org/10.1371/journal.pone.0035273)). In humans, on the other hand, we measure a higher expression of mitochondrial proteins in type 1 fibers compared to type 2A fibers, again reflecting a known feature of human fiber types (Howald et al., 1985, DOI:[10.1007/bf00589248](https://doi.org/10.1007/bf00589248)). For these reasons, we do not think this relevant for the discussion in this study.

Several sports studies in combination with proteomics have been carried out and published in the past before and none is mentioned/referenced e.g. in the introduction. More information about former relevant studies should be referenced (e.g. page 5, line 13-16).

We agree that several studies have measured the muscle proteome in response to exercise training. However, to our knowledge, only the whole muscle proteome was measured in these studies. We have included previous studies in the introduction of the manuscript (p. 5, line 10-11).

Moreover proteomic analysis of freeze dried muscle tissue is not new and has been done before also on the fiber type level (e.g. doi: [10.1007/s00401-016-1592-7](https://doi.org/10.1007/s00401-016-1592-7)).

We have had a closer look at the paper the reviewer refers to but cannot find anywhere in the method section that muscle tissue had been freeze-dried? It is correct that proteomics was performed at the fiber type level in that paper, but the fibers were not dissected from freeze-dried muscle tissue but instead laser microdissection was performed on cryosections of embedded muscles. In addition, only type I mouse muscle fibers were measured and as such does not provide information on fiber type-specific differences as in the present study. Proteomics analysis of freeze-dried tissue is not uncommon but to our knowledge, there is no published literature describing proteomic analysis on isolated muscle fibers from freeze-dried biopsies.

Our methodology is likely to be adapted by many others also because our samples are snap frozen immediately (~20 sec) after the biopsy procedure. This means that many different post-translational modifications are preserved to a higher degree than when fiber sampling is obtained from fresh muscle tissue (pre-treatment and harvesting takes several minutes) or when fresh muscle tissue is embedded for immunohistochemistry – minutes elapse before the sample is frozen. Thus, we foresee that our method will be used for illuminating the fiber type-specific post-translational modifications following various acute interventions, like hormone stimulation, single exercise bout etc.

Specific points:

Page 7, line 9-12: How many protein groups can be detected without using the match between run option? This option is useful for overall analysis but it does not really reflect the true number of identified proteins.

We agree and we have now provided this information in the manuscript (p. 7, line 6-7 and Supplementary Table 2). Without the 'match between run' option, only 2830 proteins in the fiber pools were identified.

Page 9, line 15-18: Which protein groups belong to the category "Signal"? Information is really vague and not is not sufficiently informative. A Supplemental table should be provided including protein group lists of the clusters and the respective GO term and pathway enrichment analysis.

According to reviewer's suggestions, we have reanalyzed the data using linear models in which p-values are corrected. The new statistics applies to comparisons displayed in figure 2, 3 and 4. When we repeated the statistics for figure 2, our observations did not change much but the Fisher Exact Test for enrichment gave slightly different results (likely due to increased number of significantly different proteins). The new analysis did not display 'Signal' as one of the enriched categories but it returned other categories (extracellular region, extracellular vesicle exosome, and vesicle-mediated transport) associated with protein secretion. In the revised manuscript, we have included a table (Table S6) where significantly different proteins representing two clusters (up and down regulated) are marked and lists of enriched categories are included. Since we have 98 enriched categories, it is difficult to provide the list of individual proteins from these categories. Instead, we have included corresponding annotations for all proteins, from where members of individual enriched categories can be fetched by simply using filter option in Excel. Additionally, in Supplementary Table 7, we have included members of protein categories listed in figure 2F and 2G.

Page 10, line 1: Fold change is even higher, because the data provided is log2 transformed. Non log transformed fold changes should be added, because they are better understandable for the reader.

In the revised manuscript, we have converted all log2 fold changes to non-log transformed values.

Page 10, line 2: Was FHL1 also significantly regulated? If not was it at least identified in the data set? Information should be added here.

Yes, FHL1 was quantified but did not adapt in response to training. We have included this information in the section of PDLIM1 and FHL1 (p. 10, line 3-8)

Page 11, line 1: how was this calculation done in detail?

For details about the calculation, please refer to page 31 and the section "protein abundance".

Page 11, line 3: In previous publications (see above) the effect was observed exactly the other way around. Please comment on that.

Please refer to previous comment on this matter.

Page 11, line 14-16: How many proteins were found to be differential after p-value correction?

This is a valid point. In our initial submission, p-values obtained from the two-way ANOVA were unadjusted. In the revised version, we have adopted a novel fusion scheme that adjust the raw p-values by taking into consideration both the biological significance (i.e. the log₂-change) and the raw p-value as described by Xiao (see method section p. 30, line 3-12, DOI: 10.1093/bioinformatics/btr671). This method has been shown to be effective in providing robust inferences and to protect against false discoveries.

Page 12, line 1-3: Pearson correlation of data (which was also carried out for the FT) would provide a more comprehensive information if no changes were found due to measuring whole muscle lysate or due to a high variability among the replicates.

We are unsure how reviewer #1 wants us to correlate MHC Western blot data from whole muscle lysate? The variability among samples was likely not that big as indicated by the rather small error bars in figure 3C. Furthermore, they are in line with whole muscle proteomic data (table S8).

Page 13, line 1-5: Are there more proteins of the glycogen pathways identified as being differential?

In figure 6D other proteins in glycogen metabolism are presented. In the previous version of the manuscript, in the figure 3, we included proteins involved in glycogen metabolism. These proteins were shortlisted based on non-corrected p-values. As suggested by the reviewers, the current version of the manuscript is based on corrected p-values. Previously described proteins from glycogen pathways in figure 3 are no longer a part of the manuscript.

Page 13, line 9: SRP14 and SRP9 are forming a complex. Was SRP9 also identified or other interactors? How many peptides were identified? If no other interactors could be identified this argument is really vague.

The previous data were based on non-corrected p-values. As suggested by the reviewers, the current version of the manuscript is based on corrected p-values. This means that the section referred to here is no longer a part of the manuscript.

Page 15, line 10-13: Not conclusive. Well-known markers of muscle damage are the proteins XIRP-1 and XIRP-2. Were they also found to be enriched in the samples?

XIRP1 and XIRP2 were quantified in the pools but were not regulated by exercise training.

Page 15, line 13-15: Only cardio training was carried out as described in the material and methods part. These conclusions have been made several times before in sports studies. References are missing.

The section referred to here is no longer a part of the manuscript.

Figure 5D/G: This figures are not useful at all because several proteins were compiled without getting any information how many: how many of the 37 mitochondrial encoded proteins could be identified in the data set and how many of them were compiled for this figure?

We thank reviewer for this suggestion. In the revised manuscript, we have included members of individual protein categories listed in figure 5 (Table S11). Of the 37 genes in mtDNA only 13 codes for OXPHOS proteins (the rest codes for tRNA and rRNAs). Of the 13 mtDNA-encoded OXPHOS proteins we detect 8. Please refer to table S11.

Page 19, line 4-6: Reference is from 1976. Aren't there any actual studies?

We have added a few recent studies, although we believe that the study from 1976 is still valid.

Page 19, line 9: Reference is missing

The purpose of this sentence was more speculative/hypothesizing. Thus we have modified the sentence instead.

Page 20, line 15-17: Aren't there any actual studies?

We have added a more recent study.

Page 21, lines 6-7: Similar approaches had been done before on the fiber type level (see above).

We do not fully agree that similar approaches has been done before due to the following reasons 1) the study mentioned in the previous comment was performed in mice, 2) muscles were not freeze-dried, 3) only type I muscle fibers were investigated.

Page 22, line 7-8: This number only refers to "match between run" option and does not reflect the number of truly identified proteins.

We have now provided the numbers for identified proteins with and without matching (p. 7, line 6-7 and Supplementary Table 2).

In summary, despite the important question and the beautifully readable form of the work, extensive changes and additional analysis are necessary before the quality of the manuscript is sufficient to allow publication in Nat Comm.

Reviewer #2

The manuscript by Deshmukh et al reports comprehensive proteomic analysis of 2 main classifications (type I and type II) of fibres in human skeletal muscle. The extraction of myofibres from freeze-dried muscle has been reported previously, as has proteomic analysis of individual myofibre types. Therefore the novel aspect of the current work is the analysis of samples taken prior to and after a 12 wk period of endurance training. Muscle samples were used from a previously published study (Steenberg et al 2019, J Physiol vol 597) that investigated the effects of chronic exercise training on muscle insulin sensitivity. The current manuscript reports new data on differences in protein abundance between Type I and Type II fibres at baseline, and also fibre-type specific responses to exercise training. The manuscript presents useful information on the muscle response to exercise

and the collection of data is technically excellent.

Given the focus on exercise, it is surprising that the work is entirely devoid of citations to past proteomic studies of human muscle responses to training (this field began 10 years ago). The current work represents a significant advance in the field but it is highly unusual that no attempt is made to link the data to past proteomic data.

The point is well taken and we have now cited previous proteomic studies investigating the response to training in humans both in the introduction and discussion.

The manuscript strongly emphasises the importance of new fibre type-specific data but this is overstated. The majority (558) of changes in protein abundance in response to exercise were not specific to a particular fibre type, and < 20 % (121) of proteins were regulated in a fibre type-specific manner. This does not seem to match the title “...extensive fibre type-specific adaptation...”

This is also a valid point and we have changed the title accordingly: “Deep muscle-proteomic analysis of freeze-dried human muscle biopsies reveals fiber type-specific adaptations to exercise training”

Moreover, the authors do not present a comparison of proteomic analysis of whole muscle homogenates (i.e. typical approach) versus their new fibre type-specific workflow. Therefore it is not possible to measure the worth of the additional fibre-specific information. It might also be debated whether a fibre specific approach to muscle analysis is a worthwhile endeavour when performance of the whole muscle is of interest in exercise physiology.

Thank you for these comments. Accordingly, we have analyzed the whole muscle lysates from the same subjects with exactly similar proteomics workflow. This analysis clearly shows the worth of studying pure muscle fibers. For details, see figure 4E+D, Table S3+S8 and text on p16. We agree that the analysis of the whole muscle is of interest in exercise physiology but our results can help scientific community to understand the fiber type-specific effects of exercise training. Furthermore, using skeletal muscle as a drug target has been found challenging for various reason. One of these is the risk of targeting the heart muscle as well – the heart muscle fiber phenotype is more alike the endurance type I skeletal muscle fiber and thus any differentiation between type I and II skeletal muscle fiber type might provide clues to a path targeting type II but not type I fibers (and perhaps heart muscle). In human skeletal muscle (contrast to rodents) the two fiber types are on average ~equally present and thus both fiber types contribute a significant amount of muscle tissue – making both highly relevant as targets.

The interpretation of fibre type-specific data could easily be confounded by cooccurring changes in myofibre profile or selective myofibre hypertrophy depending on the training regimen.

We are not completely sure how to interpret this question? We did try to measure size of the fibers before and after training. However, we only had embedded muscle samples prepared for cryosections in two of the five subjects. As the results between the two subjects were not consistent, it was hard to determine if hypertrophy

had occurred or not. Only one out of two subjects displayed “hypertrophy” in slow MHC I fibers. The size of fast IIX fibers was increased in one subject while decreased in the other, however no change in size of type IIA fibers (which make up the majority of the fast fiber pools) was observed in any of the two subjects. Thus, we do not think that the results obtained in the present study is merely due to hypertrophy, all though we cannot rule out some influence.

Similarly, it is difficult to know the consequences of motor unit activation pattern (i.e. relating to exercise intensity) on the interpretation of the proteomic data.

As we state in the conclusion our data is of course limited to this type of exercise training regime and it is likely that other training regimes e.g. strength exercise training would have provided other results. However, we did try to ensure recruitment of all fiber types by combining continuous endurance exercise training with short bouts (1 min) of high intensity exercise (>90% of maximal heart rate).

Sub-dividing muscle in to 2 of the 3 fibre types and reporting the data in isolation also seems somewhat counter to the holistic philosophy of proteomics. None of these issues are considered in the authors discussion and in my opinion the presentation of the work lacks balance.

With regards to the division of muscle into two and not three pools, please refer to previous comment to reviewer 1.

The authors mention that by-design their work does not consider changes in fibre phenotype but they do not discuss the consequences of this limitation. For example, there is no mention of the known multiplicity of myofibre types (i.e. fibres that co-express 2 or more myosin heavy chain isoforms) and they do not discuss why they have not also analysed type IIX fibers, which in their cohort make up almost 20 % of the muscle. N.B. It is unusual that there were no changes in myosin heavy chain profile given the age and starting baseline of the participants, and the duration of the training intervention.

It is correct that we mention that by design, our proteomic analysis does not consider a change in fiber phenotype based on the MHCs. However, since we do not find a change in fiber type composition when it is measured in whole muscle samples, we do not consider it as a limitation to our results. In the fiber type determination process hybrid fibers expressing both type I and IIA fibers were discarded as it could just as well be a type I fiber that was contaminated with a piece of a type IIA fiber or vice versa. Thus, our analysis is more or less devoid of at least I/IIA fibers, which can also be seen on the purity of the fiber pools presented in figure 1F. Regarding type IIA, IIX or hybrid IIA/IIX fibers they would be present in the fast fiber pool either way. It would of course have been interesting to measure the proteome of hybrid fibers as well. Nevertheless, for the same practical reasons given previously regarding measurements of type IIX fibers, this was not possible.

We also measured the MHCs in the entire cohort (n=9) from Steenberg et al 2019 and did not see a change in fiber type composition. In addition, whole muscle proteome analysis showed more or less similar adaptations to exercise training as in the analysis performed by Western blotting (p. 12, line 2-6)

One of the aims in performing proteomic analysis of isolated myofibre types was to investigate muscle fibre responses in the absence of contaminating fibroblast proteins. The manuscript does not report data on whether this aim was achieved.

Again, a comparison of the current data against typical whole muscle analysis could give a measurable outcome to this aim.

In the revised manuscript, we have performed proteomic analysis of whole muscle lysate. Based on our data we cannot directly confirm that the fiber pools are completely devoid of non-muscle cells. Both in the fiber pools and in the whole muscle proteome we find proteins associated, but perhaps not limited, to other cell types such as endothelium cells, adipocytes and smooth muscle cells. However, some of those proteins are also found in a previous single fiber proteome using another procedure to isolate fibers DOI: [10.1016/j.celrep.2017.05.054](https://doi.org/10.1016/j.celrep.2017.05.054). We cannot be completely sure that some of the suggested non-muscle proteins are in fact not present in skeletal muscle. Thus we cannot state whether the presence of some of these proteins is due to contamination by other cell types or whether these proteins are in fact present in skeletal muscle cells.

Lastly, the majority of proteins that were responsive to exercise were mitochondrial. Muscle mitochondria can be distinguished in to 2 sub-cellular populations (subsarcolemmal and intramyofibrillar) that exhibit different responses to exercise. It is a limitation of the current work that mitochondrial sub-populations were not also analysed, or that this aspect of the muscle response is not acknowledged.

We agree that the proteome of susarcolemmal and intramyofibrillar fraction could have been more informative because these mitochondrial subpopulations differ functionally in several ways. In our project, we are limited by the material therefore we did not consider proteomic analysis on these populations of mitochondria.

Reviewer #3

Major:

Imputation was applied for missing values. This leads to effects as described in the following example of a significant hit in table S6 and S7: MICU2 in the 2-Pre group has 4 missing values (table s2). After imputation these 4 missing points are higher than the one that was determined. In the 2-Post group 3 values are missing. This hit was reported as significant in both tables. The authors should consider to use a different strategy to replace missing values. It is crucial for a dataset with only 5 samples to mention how many missing values are present or to mark the replaced missing values in the dataset.

We completely agree with the reviewer. Imputation methods are heavily debated in our field. All imputation methods have their pro and cons. We imputed data to fill missing abundance values by drawing random numbers from a Gaussian distribution with a standard deviation of 30% in comparison to the standard deviation of measured protein abundances, and one standard deviation downshift of the mean. These parameters have been tuned in order to best simulate the distribution of low abundant proteins. In our set up,

we believe that the data is not missing by chance. Often when the data is missing, it is due to low abundance of proteins which are not detected by mass spectrometer (due to nature of the technology). Therefore, this method have been our method of choice for the last 10 years (please refer to papers from Matthias Mann's lab from the last few years).

In the revised manuscript, we have made an attempt to alleviate the concerns related to imputation issues. In the revised manuscript, before imputation, we apply slightly stringent criteria for quantification (min 3 valid values instead of 2, in at least one group). Also we included both imputed and non-imputed matrices as supplementary table (S8 and S6). Additionally, in the revised manuscript, we are providing supplemental figures (S2-S4) where we show the histogram for quantified and missing values for each group.

Regarding the reviewer's concern on MICU2 quantification, reviewer is comparing abundance in table S2 (raw normalized intensities) with other tables (Normalized (LFQ) intensities). All statistical comparisons are performed on normalized (LFQ) intensities.

We believe that our new approach of data analysis have improved overall data analysis and we hope it will be sufficient to answer reviewers questions.

The authors use paired tests. Please indicate the corresponding pairs in the tables, otherwise it is impossible to check the statistics. In the tables, please label the headers in a way consistent with the text.

It is a valid point, and we have now annotated the pairs in all revised supplementary tables.

Which calculations/normalization were done between suppl.table2 and suppl.table6. Can this explain why for the same sample number some values of one protein increases but for another it decreases?

Table S2 contains the list of the total proteins identified. The list represents just the raw (not normalized) intensities for individual proteins. The purpose of this table is to show the depth of protein coverage by MEDFASP protocol. In sup table S6, we have LFQ values, which are normalized using MaxLFQ algorithm implemented in MaxQuant software (<https://doi.org/10.1074/mcp.M113.031591>). Therefore, the normalized intensities are slightly different than the raw intensities in the Table S2. Throughout the manuscript, we used normalized LFQ intensities for quantitative comparison between the groups.

The authors used an alignment function to increase the identification rate. This is a common function in software for label-free quantification. The MS instrument is standard equipment in many facilities. Since decades it is also known that combination of proteases is beneficial for the identification, but each additional protease requires additional analysis time. What is the novelty of the presented workflow? How does it compare with other standard workflows that use the same time for analysis on comparable instruments.

Yes, the alignment function is becoming a common function in software for label-free quantification. We agree that the beneficial effects of combining proteases has been known for decades but it is not that often that the peptides from individual protease digestion are analyzed separately (LysC and trypsin in our case). Our goal here was to apply these methods on very valuable (and rare) samples like slow and fast muscle fibers and show

the deep proteome coverage at fiber type level. Although it increases the measurement time, it gives valuable information on protein quantification in fiber type-specific manner. Our study is the first of its kind presenting exercise training-induced adaptations in fiber type-specific manner.

In the revised manuscript, we compared our MED-FASP workflow with standard workflow where peptides from lysC and trypsin digestion are measured in single run (single shot). This analysis was performed on whole muscle lysates from the same subjects. Expectedly comparison of singleshot and MED-FASP (sequencing digestion with lysC and trypsin) revealed that MED-FASP workflow gives ~34% and ~37% higher identifications in protein and a unique peptides, respectively (Figure S1A,B).

Any comparison with existing methods is missing. When big numbers of identifications are mentioned, it would be very beneficial to the readers to also mention how many proteins were identified in at least half of the samples or in above 90% of all samples to get an idea on the quality of the dataset. How many of the identified proteins can be quantified with this data analysis strategy, with a standard strategy or other standard methods. What is the identification rate without the addition of the myoblast and myotube samples?

We agree. To provide this information to reader, we have now mentioned number of identifications with 50% valid values (page 7, line 5-6). In method section and in supplementary tables, we have also included information on how many proteins were used for quantitative comparison.

Other standard methods involve proteome analysis on whole muscle lysate using single run (single shot). In the revised manuscript, we have compared how the MED-FASP method is better than the single shot analysis.

We have also processed our data without match between runs option (myotubes). Without the 'match between run' option, only 2830 proteins in the fiber pools were identified.

The data is based on 5 subjects before and after training. To judge paired statistics it is important to mention which results are the pairs. It is not possible for the reader to check the statistics as this labeling is missing in the supplementary tables.

It is a valid point, we have now annotated the pairs in all revised supplementary tables.

Multiple testing correction is completely missing and would benefit the data analysis. The authors do not mention, at which fold change their method is valid. Fold change differences for the exercise dataset would be beneficial to judge if a significant hit has an impact.

This is valid concern and it is in line with issues raised by other reviewers. In the original dataset, all tests were performed in Perseus software. This software is user friendly and very popular in proteomics community but it has certain limitation. For example, this software does not have the ability to perform paired two way ANOVA and multiple testing correction. In the revised manuscript, we have re-analyzed data presented in figure 2, 3 and 4 where samples were compared in paired setting and p-values were adjusted applying a novel fusion scheme that takes both the biological significance (\log_2 -change) and the raw p-value into consideration as

described by Xiao et al. 2014. This method has been shown to be robust and to protect against false discoveries. We have also provided fold change information for all comparisons in the supplementary tables.

A volcano plot with fold change versus p-value for the exercise results gives direct information on the major changes. Please add such a plot for both fiber types to the main figures.

This is a very valuable suggestion. This has now been added in figure 3A+B.

The reporting of the results is often misleading. When changes are reported it is expected that these changes are significant. The manuscript describes a lot of proteins that are increased or decreased upon training. Some of them are not present in the dataset containing the significant changes (suppl. Tables) or could not be found by the reviewer. The authors need to check all proteins mentioned in the manuscript for their significance and if none significant proteins are described to change they need to mention that these proteins are not significantly changing and need to mention the reason why they still believe that these changes are possibly true. Especially for the chapter about glucose metabolism. For instance on page 18 the importance of changing HK2 is discussed, but HK2 is not in the suppl table 6 or 7 summarizing the significant changes and in the raw dataset (suppl. Table 2) 5 out of the 20 values are missing. Can the described differences be explained by missing values due to low abundance of the protein? Is it significantly changing? What is the significance level?

In the original manuscript, this confusion might have occurred because the statistical tests used in the previous version (analyses were performed using an unpaired ANOVA model and the p-values were unadjusted). We hope that the reviewer will find that these concerns are now adequately addressed in the revised manuscript.

Additionally, in figure 5 and 6 while describing specific protein category (from examples TIMTOM complex, glycolysis etc), we combined the abundance of proteins from those specific category and compared their total abundance within and across the groups (slow pre, slow post, fast pre, fast post). At summed abundance level, some of these categories appear to be regulated by exercise in fiber type-specific manner. However, this does not mean that individual members of these categories are also significantly different. We presented our data this way to keep in mind the muscle biologist readers.

In the revised manuscript, we have discussed the abundances on HK2 in details and hopefully our explanation is now clear. Regarding the concern comparing Table S2 and Figure 6 – the values do not match because Table S2 contains raw, non normalized intensities and Figure 6 contains sum of LFQ intensities. In the revised manuscript, we have explained the details about the statistical methods and significance level.

In addition, why are the authors not mentioning the by far more abundant HK1 (see their raw data) and also do not mention it in the graphs? Does the claim p19 line5 “we observed that training increased expression of all glycolytic enzymes ...” still hold when HK1 is included? What about other identified glycolytic enzymes? ALDOC was identified but also not mentioned. Does the claim still hold? The authors need to check all glycolytic enzymes when they claim all glycolytic enzymes and not just pick one isoform, and if they do they need to explain why they exclude the others.

We have clarified this issue by modifying the text where we describe the effects of exercise training on glycolysis. Since we use slightly stringent criteria for quantification, we concluded that HKII is quantified in slow fibers only. HKI was not regulated by exercise therefore it is not included in the text but in Table S12, we have included its fold-changes in response to exercise (both in slow and fast fibers). We did not include ALDOC because it was not quantified. In the revised manuscript, we present the list of glycolytic, PHD and TCA enzyme and their fold change and regulation (significance) upon exercise in both slow and fast fibers (Table S12).

In the related figure S1a a lot of very small fold changes probably between 0.8-1.2 are reported. Are these changes all significant? Please indicate which are significant and which not.

In the revised manuscript, we have removed the figure S1a (glycolytic enzyme), S1c (PHD enzyme) and S1D (TCA enzymes) and presented their abundance in supplementary table S12. We have also included whether these proteins are significantly changing with exercise training (with fold change and significance score) in slow and fast fibers.

As reported by Hood et al 2016, exercise leads to changes in mitochondrial biogenesis. Such a change would probably result in abundance changes of the majority of mitochondria related proteins and metabolite requirements. This knowledge in respect of the findings is crucial. The authors should discuss if such a remodeling of the cell explains all detected protein changes and which changes are independent of mitochondrial biogenesis. Is this dataset describing a known effect of training, but is a detailed description of mitochondrial biogenesis on the protein level?

We cannot be completely sure that the increase in mitochondria-related proteins is not merely due to increased mitochondrial content. However, we cannot directly measure mitochondrial content. One possibility is to use oxphos protein abundance as surrogate measures of mitochondrial content – but from a physiological point of view, any increase in mitochondrial related proteins likely results in increased capacity whether or not it is related to more organelles or more membrane area. We are open to suggestions on how to use our data to discuss this issue.

In general for all figures containing box or bar plots. As the sample number is 5 or less please add the individual points instead or in addition to boxes or bars.

Since we have included raw data and the error bars are rather small, we do not think that this is necessary. It is also not common procedure in the proteomics literature. However, if the reviewer think it is necessary, we can do it.

Minor

In the summary the authors mention in line13-14" ... it altered expression levels of proteins involved in transcription, post-translational modifications, ..." but post-translational modifications were not mentioned to be regulated in the text. Please remove PTMs or describe in more detail, which PTMs were analyzed.

The sentence has now been altered

To judge paired statistics it is important to mention which results are the pairs. It is not possible for the reader to check the statistics as this labeling is missing.

It is a valid point. We have now annotated the pairs in all revised supplementary tables.

The dot plots about the results from the glycogen measurements are missing, please add as panel in the main figure or as supplementary figure. What was the target of the glycogen antibody; glycogenin or the sugar? Please describe the antibody in more detail in the methods section. There are no statistical measurements for the glycogen analysis of the 4 measurements. Please do the calculations and mention that $n=4$ in the text. Please also add the results of the controls or standards that were used.

The representative dot blot of glycogen is already included in the main figure 1C. The entire blot containing standards and samples has been included in the Source Data File. The standards used were samples with a known concentration of glycogen measured biochemically. They show a dynamic range of samples with low and high glycogen levels, respectively DOI:10.1113/jphysiol.2014.283267. The antibody recognizes the α -1,6 linkages in the glycogen structure, this information has now been added to the method section. Statistical measurements have now been performed for glycogen measurements in fiber pools. The effect of training was not statistically significant but close (slow fibers $p=0.06$, fast fibers $p=0.1$). The low n and thus low power is likely the explanation. The individual values are included in the figure. We have added in the method section that the analysis was performed on $n=4$.

Page7 line5 "...we used a stringent FDR..." . A FDR of 1% is standard in proteomics analysis.

The sentence has been modified and stringent has been removed.

The purity of the fibers are tested by antibodies and the proteomics results. When comparing these results it would be beneficial to use a common nomenclature, like the gene name. Please further define the antibodies used in the materials section.

The heading in figure 1F is changed to the same nomenclature as figure 1B. The gene names of myosin heavy chains are kept to highlight specific isoforms.

On page 8 line 11 "Thus, our fiber isolation procedure and proteomic workflow yielded detailed coverage of metabolic..." Up to this point no detailed coverage of metabolic features resulting from the proteomic workflow has been mentioned to come to this conclusion. Please clarify which features did the authors mean.

The sentence has been modified.

Page10 line2 The authors find an enrichment of PDLIM1 and discuss the potential interaction with FHL1. FHL1 is not changing or very little. Can the authors implement this finding in the discussion and discuss why one of the interaction partner is eight fold enriched but the other not.

FHL1 did not change with exercise training; we have added this information to the discussion.

Fig2F and G Please mention in detail in the methods section how the results were calculated and which statistics were used. The number of proteins per group n=... would be beneficial in the figure and an indication how the abundance was calculated in the legend; median or average or sum. If the sum was used, how was it normalized if the number of proteins in the different groups was different.

The results were calculated as summed protein abundances for the proteins of the different categories. We have included a table, which lists the proteins in each of the categories of fig 2F and 2G (Table S7). The majority of proteins were quantified in both fiber types therefore there was no concern regarding number of proteins being different in different groups. We used paired linear models with Xiao correction to evaluate fiber type-specific differences in these figures (p 30, line 3-12)

P12 line16 What is the fold change of CARM1. The z-score (fig 3) removes the variance, please add the fold difference, p-values and add the multiple testing corrected p-value to the figure3 c-g. Box plots out of only max.5 data points are often misleading. The individual data points should be added to the box plots or replace the box plots by the individual points.

The previous data was based on non-corrected p-values. As suggested by the reviewers, the current version of the manuscript is based on corrected p-values. This means that the section referred to here is no longer a part of the manuscript.

P34 line 18 Did the authors really use a C10 column?

It was a typo and it has now been corrected to C18.

Reviewers' Comments:

Reviewer #1:

Remarks to the Author:

Deep muscle-proteomic analysis of freeze-dried human muscle biopsies 1 reveals extensive fiber 2 type-specific adaptations to exercise training

Authors performed an extensive study on the effects of exercise training on the proteome of different skeletal muscle fiber types. Different fiber types exhibit type-specific adaptive responses to training and especially metabolic pathways and mitochondria seemed to be affected.

In the revised version of the manuscript authors addressed most of my raised questions satisfactorily. The results are comprehensible and statistically valid now and were verified using an independent method. Results are very relevant for the scientific community.

There are still two points I want to point out:

Authors insist on the fact they are the first to perform proteomics on individual muscle fibers which is definitely not true and was shown in other publications already. Species and procedure of muscle preparation were different that is true but this is not my point. My point is that proteomics of single muscle fibers was already performed successfully and this fact should be addressed properly.

The second point addresses the calculation and description of numbers of identified proteins. The calculation is misleading. Since the authors state that they do single muscle fiber proteomics, the numbers of proteins that can be identified in each fiber should be given here instead of counting proteins from measured pools and high numbers of muscle cells (see page 7). This number does not at all reflect the true number of identified proteins in single fibers. On page 21 authors state to identify over 4000 proteins from low sample amounts. Again – how many proteins are identified from e.g. a single fibre? Or even more informative what is the amount of sample resulting in 4000 identified proteins?

Those points should be objectively addressed prior to publication.

Reviewer #2:

None

Reviewer #3

The comments of the second revision are colored in blue.

Major:

Imputation was applied for missing values. This leads to effects as described in the following example of a significant hit in table S6 and S7: MICU2 in the 2-Pre group has 4 missing values (table s2). After imputation these 4 missing points are higher than the one that was determined. In the 2-Post group 3 values are missing. This hit was reported as significant in both tables. The authors should consider to use a different strategy to replace missing values. It is crucial for a dataset with only 5 samples to mention how many missing values are present or to mark the replaced missing values in the dataset.

We completely agree with the reviewer. Imputation methods are heavily debated in our field. All imputation methods have their pro and cons. We imputed data to fill missing abundance values by drawing random numbers from a Gaussian distribution with a standard deviation of 30% in comparison to the standard deviation of measured protein abundances, and one standard deviation downshift of the mean. These parameters have been tuned in order to best simulate the distribution of low abundant proteins. In our set up, we believe that the data is not missing by chance. Often when the data is missing, it is due to low abundance of proteins which are not detected by mass spectrometer (due to nature of the technology). Therefore, this method have been our method of choice for the last 10 years (please refer to papers from Matthias Mann's lab from the last few years).

In the revised manuscript, we have made an attempt to alleviate the concerns related to imputation issues. In the revised manuscript, before imputation, we apply slightly stringent criteria for quantification (min 3 valid values instead of 2, in at least one group). Also we included both imputed and non-imputed matrices as supplementary table (S8 and S6). Additionally, in the revised manuscript, we are providing supplemental figures (S2-S4) where we show the histogram for quantified and missing values for each group.

Regarding the reviewer's concern on MICU2 quantification, reviewer is comparing abundance in table S2 (raw normalized intensities) with other tables (Normalized (LFQ) intensities). All statistical comparisons are performed on normalized (LFQ) intensities.

We believe that our new approach of data analysis have improved overall data analysis and we hope it will be sufficient to answer reviewers questions.

Increasing the criteria from in the worst case 2 out of 10 to 3 out of 10 is not very stringent. Along the line of problems with imputations, the authors now changed on Page 10, line 3 the text to make the following claim: "PDLIM1 was the most significantly different protein and had a 181-fold higher expression in slow vs. fast fibers."

If the authors make such a statement, they need to check the data thoroughly. PDLIM1 was not detected in one group (Suppl. Table 6). An exact expression change between detected in one group and not detected in another can not be calculated. This is basically infinite and the resulting 181- fold change is thus completely arbitrary as it completely depends on the imputation parameters. In addition, the authors should know from their long year experience with label free analysis that such high values like 181-fold are by far not accurate.

Please, check all numbers in the manuscript again thoroughly and mention in the text if and in which way the conclusion is influenced by imputations. I still think that for such a small number of donors imputation is not the best option, but as long as the authors work with imputation in a transparent way, the reader has a chance to understand which claims are influenced and which are not.

The authors use paired tests. Please indicate the corresponding pairs in the tables, otherwise it is impossible to check the statistics. In the tables, please label the headers in a way consistent with the text.

It is a valid point, and we have now annotated the pairs in all revised supplementary tables.

Which calculations/normalization were done between suppl.table2 and suppl.table6. Can this explain why for the same sample number some values of one protein increases but for another it decreases?

Table S2 contains the list of the total proteins identified. The list represents just the raw (not normalized) intensities for individual proteins. The purpose of this table is to show the depth of protein coverage by MEDFASP protocol. In sup table S6, we have LFQ values, which are normalized using MaxLFQ algorithm implemented in MaxQuant software (<https://doi.org/10.1074/mcp.M113.031591>). Therefore, the normalized intensities are slightly different than the raw intensities in the Table S2. Throughout the manuscript, we used normalized LFQ intensities for quantitative comparison between the groups.

The authors used an alignment function to increase the identification rate. This is a common function in software for label-free quantification. The MS instrument is standard equipment in many facilities. Since decades it is also known that combination of proteases is beneficial for the identification, but each additional protease requires additional analysis time. What is the novelty of the presented workflow? How does it compare with other standard workflows that use the same time for analysis on comparable instruments.

Yes, the alignment function is becoming a common function in software for label-free quantification. We agree that the beneficial effects of combining proteases has been known for decades but it is not that often that the peptides from individual protease digestion are analyzed separately (LysC and trypsin in our case). Our goal here was to apply these methods on very valuable (and rare) samples like slow and fast muscle fibers and show the deep proteome coverage at fiber type level. Although it increases the measurement time, it gives valuable information on protein quantification in fiber type-specific manner. Our study is the first of its kind presenting exercise training-induced adaptations in fiber type-specific manner.

In the revised manuscript, we compared our MED-FASP workflow with standard workflow where peptides from lysC and trypsin digestion are measured in single run (single shot). This analysis was performed on whole muscle lysates from the same subjects. Expectedly comparison of singleshot and MED-FASP (sequencing digestion with lysC and trypsin) revealed that MED-FASP workflow gives ~34% and ~37% higher identifications in protein and a unique peptides, respectively (Figure S1A,B).

The manuscript reads as you have compared a single measurement (single shot) versus two measurements one for LysC and one for trypsin (MED-FASP one for each enzyme). This is not a fair comparison as a measurement that is twice as long (MED-FASP) should always lead to more identifications. A good comparison would be to

spend the same time on the MS for the single shot and the MED-FASP and guarantee the most effective analysis when the single shot is measured twice as long. In addition, twice the material is needed for the MED-FASP protocol. A valid comparison also includes a fair discussion about the drawbacks of the method, like the longer time in preparation, LC-MS time, sample amount... These points need to be mentioned and discussed.

Any comparison with existing methods is missing. When big numbers of identifications are mentioned, it would be very beneficial to the readers to also mention how many proteins were identified in at least half of the samples or in above 90% of all samples to get an idea on the quality of the dataset. How many of the identified proteins can be quantified with this data analysis strategy, with a standard strategy or other standard methods. What is the identification rate without the addition of the myoblast and myotube samples?

We agree. To provide this information to reader, we have now mentioned number of identifications with 50% valid values (page 7, line 5-6). In method section and in supplementary tables, we have also included information on how many proteins were used for quantitative comparison.

The number of proteins that are identified in 50% of the samples was: "... identified protein groups in muscle fibers were 3,360" After switching off match between runs 2830 were left. Please also add how many proteins were identified in at least 50% and 90% of the samples after switching of match between runs.

Other standard methods involve proteome analysis on whole muscle lysate using single run (single shot). In the revised manuscript, we have compared how the MED-FASP method is better than the single shot analysis.

As mentioned above this is not a fair comparison. Another fair comparison would be to exchange LysC in your workflow by for instance AspN, GluC, elastase, or LysN and proof that the combination of LysC with trypsin is better or at least cite some of the many papers in which this has been done before.

We have also processed our data without match between runs option (myotubes). Without the 'match between run' option, only 2830 proteins in the fiber pools were identified.

The data is based on 5 subjects before and after training. To judge paired statistics it is important to mention which results are the pairs. It is not possible for the reader to check the statistics as this labeling is missing in the supplementary tables.

It is a valid point, we have now annotated the pairs in all revised supplementary tables.

Multiple testing correction is completely missing and would benefit the data analysis. The authors do not mention, at which fold change their method is valid. Fold change differences for the exercise dataset would be beneficial to judge if a significant hit has an impact.

This is valid concern and it is in line with issues raised by other reviewers. In the original dataset, all tests were performed in Perseus software. This software is user friendly and very popular in proteomics community but it has certain limitation. For example, this software does not have the ability to perform paired two way ANOVA and multiple testing correction. In the revised manuscript, we have re-analyzed data presented in figure 2, 3 and 4 where samples were compared in paired setting and p-values were adjusted applying a novel fusion scheme that takes both the biological significance (log2-change) and the raw p-value into consideration as

described by Xiao et al. 2014. This method has been shown to be robust and to protect against false discoveries. We have also provided fold change information for all comparisons in the supplementary tables.

A volcano plot with fold change versus p-value for the exercise results gives direct information on the major changes. Please add such a plot for both fiber types to the main figures.

This is a very valuable suggestion. This has now been added in figure 3A+B.

The reporting of the results is often misleading. When changes are reported it is expected that these changes are significant. The manuscript describes a lot of proteins that are increased or decreased upon training. Some of them are not present in the dataset containing the significant changes (suppl. Tables) or could not be found by the reviewer. The authors need to check all proteins mentioned in the manuscript for their significance and if none significant proteins are described to change they need to mention that these proteins are not significantly changing and need to mention the reason why they still believe that these changes are possibly true. Especially for the chapter about glucose metabolism. For instance on page 18 the importance of changing HK2 is discussed, but HK2 is not in the suppl table 6 or 7 summarizing the significant changes and in the raw dataset (suppl. Table 2) 5 out of the 20 values are missing. Can the described differences be explained by missing values due to low abundance of the protein? Is it significantly changing? What is the significance level?

In the original manuscript, this confusion might have occurred because the statistical tests used in the previous version (analyses were performed using an unpaired ANOVA model and the p-values were unadjusted). We hope that the reviewer will find that these concerns are now adequately addressed in the revised manuscript.

Additionally, in figure 5 and 6 while describing specific protein category (from examples TIMTOM complex, glycolysis etc), we combined the abundance of proteins from those specific category and compared their total abundance within and across the groups (slow pre, slow post, fast pre, fast post). At summed abundance level, some of these categories appear to be regulated by exercise in fiber type-specific manner. However, this does not mean that individual members of these categories are also significantly different. We presented our data this way to keep in mind the muscle biologist readers.

In summed abundances, a significant change of one member of the group can lead to a significant change of the whole group. In respect of moonlighting functions of proteins, this might falsify the conclusion as already recognized by the authors. In order to prevent false conclusions the following numbers are required to be added: how many proteins are in a protein category, how many of these were quantified (not identified!) in the study, how many showed a significant change in the same direction of the summed change and how many in the opposite direction.

Fig 5 E LDHA fast: Visually there is no difference but it is indicated as $p < 0.01$. Is this correct?

In the revised manuscript, we have discussed the abundances on HK2 in details and hopefully our explanation is now clear. Regarding the concern comparing Table S2 and Figure 6 – the values do not match because Table S2

contains raw, non normalized intensities and Figure 6 contains sum of LFQ intensities. In the revised manuscript, we have explained the details about the statistical methods and significance level.

In addition, why are the authors not mentioning the by far more abundant HK1 (see their raw data) and also do not mention it in the graphs? Does the claim p19 line5 “we observed that training increased expression of all glycolytic enzymes ...” still hold when HK1 is included? What about other identified glycolytic enzymes? ALDOC was identified but also not mentioned. Does the claim still hold? The authors need to check all glycolytic enzymes when they claim all glycolytic enzymes and not just pick one isoform, and if they do they need to explain why they exclude the others.

We have clarified this issue by modifying the text where we describe the effects of exercise training on glycolysis. Since we use slightly stringent criteria for quantification, we concluded that HKII is quantified in slow fibers only. HK1 was not regulated by exercise therefore it is not included in the text but in Table S12, we have included its fold-changes in response to exercise (both in slow and fast fibers). We did not include ALDOC because it was not quantified. In the revised manuscript, we present the list of glycolytic, PHD and TCA enzyme and their fold change and regulation (significance) upon exercise in both slow and fast fibers (Table S12).

On page 19 line 4-8 the authors write:

In addition, total abundance of glycolytic enzymes only increased with training in fast fibers (Fig. 6B), indicating training-induced enhanced capacity of glycolysis in fast muscle fibers. Interestingly, when comparing the abundance of individual glycolytic proteins, only HKII (in slow fibers) and Enolase 3 (in fast muscle fibers) were significantly upregulated with exercise training (Supplementary Table 12).

Here, the authors recognize that Enolase 3 is the only significant changing glycolytic protein in fast fibers. This leads to the total abundance of glycolytic enzymes to change. If the authors recognize that a conclusion is not correct as the pathway result is the result of a single enzyme, why do they now include it in the text. Later in the manuscript they mention glycolysis is changing. Again, the authors need to make sure that their claims about pathways/groups of proteins are not based on single proteins, but on the majority of the pathway.

In the related figure S1a a lot of very small fold changes probably between 0.8-1.2 are reported. Are these changes all significant? Please indicate which are significant and which not.

In the revised manuscript, we have removed the figure S1a (glycolytic enzyme), S1c (PHD enzyme) and S1D (TCA enzymes) and presented their abundance in supplementary table S12. We have also included whether these proteins are significantly changing with exercise training (with fold change and significance score) in slow and fast fibers.

As reported by Hood et al 2016, exercise leads to changes in mitochondrial biogenesis. Such a change would probably result in abundance changes of the majority of mitochondria related proteins and metabolite requirements. This knowledge in respect of the findings is crucial. The authors should discuss if such a remodeling of the cell explains all detected protein changes and which changes are independent of mitochondrial biogenesis. Is this dataset describing a known effect of training, but is a detailed description of mitochondrial biogenesis on the protein level?

We cannot be completely sure that the increase in mitochondria-related proteins is not merely due to increased mitochondrial content. However, we cannot directly measure mitochondrial content. One possibility is to use oxphos protein abundance as surrogate measures of mitochondrial content – but from a physiological point of view, any increase in mitochondrial related proteins likely results in increased capacity whether or not it is related to more organelles or more membrane area. We are open to suggestions on how to use our data to discuss this issue.

The most important point is that the authors write in the manuscript, that there is the possibility that the proteomic changes they see might be due to the changes in mitochondrial biogenesis as reported previously for instance by Hood et al. 2016. One possibility to differentiate between protein alterations related to or being unrelated to mitochondrial biogenesis would be a comparison with existing data on mitochondrial biogenesis. My suggestion would be to look up in the literature, if there are descriptions on which proteins are altered upon changes in mitochondrial biogenesis, preferentially more than a single dataset and compare them with the presented alterations.

In general for all figures containing box or bar plots. As the sample number is 5 or less please add the individual points instead or in addition to boxes or bars.

Since we have included raw data and the error bars are rather small, we do not think that this is necessary. It is also not common procedure in the proteomics literature. However, if the reviewer think it is necessary, we can do it.

If the reviewer would have thought it is not necessary, he would have not mentioned it as a major point. It is good practice to do so and most of the good publications in proteomics follow this good practice. This is also the reason why it is mentioned in the Nature Editorial Policy Checklist that you have signed. See the point: “Data representation; Point: Individual data points are shown when possible, and always for $n < 10$ ”

Minor

In the summary the authors mention in line13-14”... it altered expression levels of proteins involved in transcription, post-translational modifications, ...” but post-translational modifications were not mentioned to be regulated in the text. Please remove PTMs or describe in more detail, which PTMs were analyzed.

The sentence has now been altered

The authors removed PTM in the summary, but now added a comment in another part of the manuscript: *Page 5, line 15-16: This approach is advantageous in future measurements of fiber type-specific posttranslational modifications to ...*

Why do the authors introduce a new claim, that this method will be advantageous for PTM analysis. There is no relation to PTM mentioned before.

To judge paired statistics it is important to mention which results are the pairs. It is not possible for the reader to check the statistics as this labeling is missing.

It is a valid point. We have now annotated the pairs in all revised supplementary tables.

The dot plots about the results from the glycogen measurements are missing, please add as panel in the main figure or as supplementary figure. What was the target of the glycogen antibody; glycogenin or the sugar? Please describe the antibody in more detail in the methods section. There are no statistical measurements for the glycogen analysis of the 4 measurements. Please do the calculations and mention that n=4 in the text. Please also add the results of the controls or standards that were used.

The representative dot blot of glycogen is already included in the main figure 1C. The entire blot containing standards and samples has been included in the Source Data File. The standards used were samples with a known concentration of glycogen measured biochemically. They show a dynamic range of samples with low and high glycogen levels, respectively DOI:10.1113/jphysiol.2014.283267. The antibody recognizes the α -1,6 linkages in the glycogen structure, this information has now been added to the method section. Statistical measurements have now been performed for glycogen measurements in fiber pools. The effect of training was not statistically significant but close (slow fibers p=0.06, fast fibers p=0.1). The low n and thus low power is likely the explanation. The individual values are included in the figure. We have added in the method section that the analysis was performed on n=4.

The reviewer could not find the statistics in the manuscript. Why do the authors not mention the statistics in the manuscript after calculating and realizing that they are not significant (for p<0.05)? This is a very important finding that the claims on glycogen are not significant for p< 0.05 and based on n=4. Still, the glycogen measurements have a tendency and this finding should be kept in the manuscript, but please also mention the statistics in the main text of the manuscript and do not hide the n=4 in the method section; add it to the main text. Be fair to your readers.

Page7 line5 "...we used a stringent FDR..." . A FDR of 1% is standard in proteomics analysis.

The sentence has been modified and stringent has been removed.

The purity of the fibers are tested by antibodies and the proteomics results. When comparing these results it would be beneficial to use a common nomenclature, like the gene name. Please further define the antibodies used in the materials section.

The heading in figure 1F is changed to the same nomenclature as figure 1B. The gene names of myosin heavy chains are kept to highlight specific isoforms.

On page 8 line 11 "Thus, our fiber isolation procedure and proteomic workflow yielded detailed coverage of metabolic..." Up to this point no detailed coverage of metabolic features resulting from the proteomic workflow has been mentioned to come to this conclusion. Please clarify which features did the authors mean.

The sentence has been modified.

Page10 line2 The authors find an enrichment of PDLIM1 and discuss the potential interaction with FHL1. FHL1 is not changing or very little. Can the authors implement this finding in the discussion and discuss why one of the interaction partner is eight fold enriched but the other not.

FHL1 did not change with exercise training; we have added this information to the discussion.

Fig2F and G Please mention in detail in the methods section how the results were calculated and which statistics were used. The number of proteins per group $n=...$ would be beneficial in the figure and an indication how the abundance was calculated in the legend; median or average or sum. If the sum was used, how was it normalized if the number of proteins in the different groups was different.

The results were calculated as summed protein abundances for the proteins of the different categories. We have included a table, which lists the proteins in each of the categories of fig 2F and 2G (Table S7). The majority of proteins were quantified in both fiber types therefore there was no concern regarding number of proteins being different in different groups. We used paired linear models with Xiao correction to evaluate fiber typespecific differences in these figures (p 30, line 3-12)

P12 line16 What is the fold change of CARM1. The z-score (fig 3) removes the variance, please add the fold difference, p-values and add the multiple testing corrected p-value to the figure3 c-g. Box plots out of only max.5 data points are often misleading. The individual data points should be added to the box plots or replace the box plots by the individual points.

The previous data was based on non-corrected p-values. As suggested by the reviewers, the current version of the manuscript is based on corrected p-values. This means that the section referred to here is no longer a part of the manuscript.

P34 line 18 Did the authors really use a C10 column?

It was a typo and it has now been corrected to C18.

Additional comments:

Page 9, line 10: The authors mention 469 significant changing proteins between slow and fast fibers. "Statistical analysis returned 469 proteins that were significantly different 11 between slow and fast fibers (significance score<0.05, Fig. 2D, Supplementary Table 6)."

But in Suppl. Table. 6 there are 471 proteins marked as significant. Please make sure to report the correct numbers.

Page 11, line 16: ...we found 236 and 171 proteins that were significantly...

The number of proteins marked with "+" in the Suppl. Table 8 are 237 and 172 and after checking the p-value entries these were only 170 < 0.05 in the fast fibers. Please check the whole manuscript again and define if you are using $p<0.05$ or $p\leq 0.05$. Make sure you always work with the same set of proteins and report the same number.

Minor:

Typo in the affiliation of the first author.

Typo in Suppl. Table 2 is labeled as Suppl. Table 1 in excel.

Page 80, line 22: Typo PREvs.Post

In summary, the authors need to check their whole manuscript again to assure that the numbers are consistent throughout the manuscript. They should be transparent with their statistics and if a claim is based on missing data (meaning the proteins is quantified in $n < 5$ per group) it should be mentioned along with the claim in the main text of the manuscript.

Manuscript ID - NCOMMS-19-12954A

We would like to thank the reviewers for their comprehensive work and very appreciated comments. Below is the point-to-point response to reviewers. Comments and answers from the first revision are kept for clarity and are marked by black and red, respectively. Comments and answers concerning the current (2nd) revision are written in light blue and purple, respectively. To check the new text in the revised manuscript, please refer to manuscript file with track changes. The page numbers and the lines highlighted in yellow correspond to the manuscript file with track changes.

Reviewer #1 (Remarks to the Author):

Deep muscle-proteomic analysis of freeze-dried human muscle biopsies 1 reveals extensive fiber 2 type-specific adaptations to exercise training. Authors performed an extensive study on the effects of exercise training on the proteome of different skeletal muscle fiber types. Different fiber types exhibit type-specific adaptive responses to training and especially metabolic pathways and mitochondria seemed to be affected. In the revised version of the manuscript authors addressed most of my raised questions satisfactorily. The results are comprehensible and statistically valid now and were verified using an independent method. Results are very relevant for the scientific community. There are still two points I want to point out:

Authors insist on the fact they are the first to perform proteomics on individual muscle fibers which is definitely not true and was shown in other publications already. Species and procedure of muscle preparation were different that is true but this is not my point. My point is that proteomics of single muscle fibers was already performed successfully and this fact should be addressed properly.

We do not claim that we are the first to perform proteomics on muscle fibers. But to our knowledge, this is the first study that performed proteomics analysis of *muscle fibers isolated from freeze-dried samples in the context of physical exercise*. In the previous studies, the muscle fibers were isolated from either fresh muscle biopsies (Murgia et al <https://doi.org/10.15252/embr.201439757>) or by laser capture dissection procedures (Winter et al doi: [10.1007/s00401-016-1592-7](https://doi.org/10.1007/s00401-016-1592-7)). To provide more clarity on this topic, we have included these references in the introduction part (page 5, lines 17-19). Thus, we believe the section is now more clear with respect to the novelty of the current study. If we have missed any relevant literature besides the above-mentioned, please let us know so we can include them in the manuscript.

The second point addresses the calculation and description of numbers of identified proteins. The calculation is misleading. Since the authors state that they do single muscle fiber proteomics, the numbers of proteins that can be identified in each fiber should be given here instead of counting proteins from measured pools and high numbers of muscle cells (see page 7). This number does not at all reflect the true number of identified proteins in single fibers. On page 21 authors state to identify over 4000 proteins from low sample amounts. Again – how many proteins are identified from e.g. a single fibre? Or even more informative what is the amount of sample resulting in 4000 identified proteins?

We are sorry if the description related to the number of identified proteins was unclear. We do not claim that we are analyzing single muscle fiber proteome. In the first paragraph of the results section, we wrote ‘Pools of 31-35 typified slow and fast twitch fibers were prepared for subsequent proteomic analysis and glycogen measurements’. In the same section, we also mention corresponding protein amount from the pool of the fibers (30 microgram). So the number reported here is from the pool of typified fibers and not from individual single fibers. This also applied to the sentences on p22 – over 4000 proteins are identified from the pool of the fibers. We pooled the fibers because our goal was to measure deep proteome of slow and fast muscle fibers. In our pilot experiments we compared the single fiber proteome and pooled fiber proteome. As displayed in the figure below in single fibers, we identified approximately 1800 proteins and in the pooled fibers identified approx.. 3200 proteins. Since our goal was to measure deep proteome, we performed analysis on the pooled (typified) fibers.

Reviewer #3

Major:

Imputation was applied for missing values. This leads to effects as described in the following example of a significant hit in table S6 and S7: MICU2 in the 2-Pre group has 4 missing values (table s2). After imputation these 4 missing points are higher than the one that was determined. In the 2-Post group 3 values are missing. This hit was reported as significant in both tables. The authors should consider to use a different strategy to replace missing values. It is crucial for a dataset with only 5 samples to mention how many missing values are present or to mark the replaced missing values in the dataset.

We completely agree with the reviewer. Imputation methods are heavily debated in our field. All imputation methods have their pro and cons. We imputed data to fill missing abundance values by drawing random numbers from a Gaussian distribution with a standard deviation of 30% in comparison to the standard deviation of measured protein abundances, and one standard deviation downshift of the mean. These parameters have been tuned in order to best simulate the distribution of low abundant proteins. In our set up, we believe that the data is not missing by chance. Often when the data is missing, it is due to low abundance of

proteins which are not detected by mass spectrometer (due to nature of the technology). Therefore, this method have been our method of choice for the last 10 years (please refer to papers from Matthias Mann's lab from the last few years).

In the revised manuscript, we have made an attempt to alleviate the concerns related to imputation issues. In the revised manuscript, before imputation, we apply slightly stringent criteria for quantification (min 3 valid values instead of 2, in at least one group). Also we included both imputed and non-imputed matrices as supplementary table (S8 and S6). Additionally, in the revised manuscript, we are providing supplemental figures (S2-S4) where we show the histogram for quantified and missing values for each group. Regarding the reviewer's concern on MICU2 quantification, reviewer is comparing abundance in table S2 (raw normalized intensities) with other tables (Normalized (LFQ) intensities). All statistical comparisons are performed on normalized (LFQ) intensities.

We believe that our new approach of data analysis have improved overall data analysis and we hope it will be sufficient to answer reviewers questions.

Increasing the criteria from in the worst case 2 out of 10 to 3 out of 10 is not very stringent. Along the line of problems with imputations, the authors now changed on Page 10, line 3 the text to make the following claim: "PDLIM1 was the most significantly different protein and had a 181-fold higher expression in slow vs. fast fibers." If the authors make such a statement, they need to check the data thoroughly. PDLIM1 was not detected in one group (Suppl. Table 6). An exact expression change between detected in one group and not detected in another can not be calculated. This is basically infinite and the resulting 181- fold change is thus completely arbitrary as it completely depends on the imputation parameters. In addition, the authors should know from their long year experience with label free analysis that such high values like 181-fold are by far not accurate. Please, check all numbers in the manuscript again thoroughly and mention in the text if and in which way the conclusion is influenced by imputations. I still think that for such a small number of donors imputation is not the best option, but as long as the authors work with imputation in a transparent way, the reader has a chance to understand which claims are influenced and which are not.

Data filtering and data imputations are one of the most discussed issues in our field, as we know there are advantages and disadvantages of using imputation methods. Filtering the data is a critical step because we do not want to throw away important biological findings. Like all other studies, here when we compared two groups, we have filtered proteins that were quantified at least 3 times in at least one group (that's 3 out of 5 in at least one group). Before imputation, the majority of proteins in the dataset had >50% valid values (>5 out of 10). In our view, 'filtering 60-70% values in at least one group' is more appropriate than filtering 50-90% values on the total dataset. This is because often proteins are exclusively expressed/quantified in one of the groups (like in slow and fast muscle fibers). Therefore, we feel that the reviewer's understanding of filtering 3 out of 10 is partially incorrect.

The reviewer's concern related to imputations and fold-change is very valid. We appreciate this comment. If a protein is exclusively quantified in one group vs another group, after imputation, its absolute fold change is bound to be gigantic between the groups. In the revised manuscript, we have removed the fold change for PDLIM1 and mentioned that the protein was exclusively quantified in slow fiber. We prefer to write log2 fold change but since reviewer 2 suggested writing absolute fold change, we included this information in the previous version. Nevertheless, in the revised manuscript, we have gone through all text concerning fold change information. PDLIM1 was the only protein, which was exclusively quantified in one group. If the readers

are interested in any specific protein, they will have the opportunity to look into the supplementary tables, where we have provided information on protein quantifications before and after imputations.

The authors use paired tests. Please indicate the corresponding pairs in the tables, otherwise it is impossible to check the statistics. In the tables, please label the headers in a way consistent with the text.

It is a valid point, and we have now annotated the pairs in all revised supplementary tables.

Which calculations/normalization were done between suppl.table2 and suppl.table6. Can this explain why for the same sample number some values of one protein increases but for another it decreases?

Table S2 contains the list of the total proteins identified. The list represents just the raw (not normalized) intensities for individual proteins. The purpose of this table is to show the depth of protein coverage by MEDFASP protocol. In sup table S6, we have LFQ values, which are normalized using MaxLFQ algorithm implemented in MaxQuant software (<https://doi.org/10.1074/mcp.M113.031591>). Therefore, the normalized intensities are slightly different than the raw intensities in the Table S2. Throughout the manuscript, we used normalized LFQ intensities for quantitative comparison between the groups.

The authors used an alignment function to increase the identification rate. This is a common function in software for label-free quantification. The MS instrument is standard equipment in many facilities. Since decades it is also known that combination of proteases is beneficial for the identification, but each additional protease requires additional analysis time. What is the novelty of the presented workflow? How does it compare with other standard workflows that use the same time for analysis on comparable instruments.

Yes, the alignment function is becoming a common function in software for label-free quantification. We agree that the beneficial effects of combining proteases has been known for decades but it is not that often that the peptides from individual protease digestion are analyzed separately (LysC and trypsin in our case). Our goal here was to apply these methods on very valuable (and rare) samples like slow and fast muscle fibers and show the deep proteome coverage at fiber type level. Although it increases the measurement time, it gives valuable information on protein quantification in fiber type-specific manner. Our study is the first of its kind presenting exercise training-induced adaptations in fiber type-specific manner.

In the revised manuscript, we compared our MED-FASP workflow with standard workflow where peptides from lysC and trypsin digestion are measured in single run (single shot). This analysis was performed on whole muscle lysates from the same subjects. Expectedly comparison of singleshot and MED-FASP (sequencing digestion with lysC and trypsin) revealed that MED-FASP workflow gives ~34% and ~37% higher identifications in protein and a unique peptides, respectively (Figure S1A,B).

The manuscript reads as you have compared a single measurement (single shot) versus two measurements one for LysC and one for trypsin (MED-FASP one for each enzyme). This is not a fair comparison as a measurement that is twice as long (MED-FASP) should always lead to more identifications. A good comparison would be to spend the same time on the MS for the single shot and the MED-FASP and guarantee the most effective analysis when the single shot is measured twice as long. In addition, twice the material is needed for the MEDFASP protocol. A valid comparison also includes a fair discussion about the drawbacks of the method, like the longer time in preparation, LC-MS time, sample amount... These points need to be mentioned and discussed.

Thank you for your comments. This is a very valid point. Due to the nature of the technology, peptides are picked up randomly during each run. Therefore we should have compared 2x measurements of single shot proteome with trypsin digestion to the MED-FASP (LysC and Trypsin) runs. In the original MED-FASP paper by our colleagues (Wisniewski and Mann, Analytical chemistry 2012) such comparison was done. They reported that parallel measurements of tryptic digest from HeLa lysate had 70% peptides in common while consecutive digestion using LysC and trypsin had only 3.6% peptides in common (that is why MED-FASP results in higher protein identifications). In our study, while comparing single shot vs MED-FASP we should have measured tryptic peptides twice. We believe that we would have got similar results as described in the original MED-FASP paper and our conclusion would not change much. We would have liked to re-measure these samples but unfortunately, we do not have the material left to do so.

As the reviewer suggested, in the revised manuscript, we have added the following sentence to highlight limitations of the MED-FASP method 'Whilst higher identifications are apparent with multiple digestion steps compared to single-shot proteomes, this comes at the cost of larger starting material and longer measurement times.' (Page 8, lines 4-6)

Any comparison with existing methods is missing. When big numbers of identifications are mentioned, it would be very beneficial to the readers to also mention how many proteins were identified in at least half of the samples or in above 90% of all samples to get an idea on the quality of the dataset. How many of the identified proteins can be quantified with this data analysis strategy, with a standard strategy or other standard methods. What is the identification rate without the addition of the myoblast and myotube samples?

We agree. To provide this information to reader, we have now mentioned number of identifications with 50% valid values (page 7, line 5-6). In method section and in supplementary tables, we have also included information on how many proteins were used for quantitative comparison.

The number of proteins that are identified in 50% of the samples was: "... identified protein groups in muscle fibers were 3,360" After switching off match between runs 2830 were left. Please also add how many proteins were identified in at least 50% and 90% of the samples after switching of match between runs.

Thank you for your comment. When the matching was turned off, the total number of identification with 50% valid values was 2299 and with 90% valid values the identification was 1616. The marked reduction in the numbers with 90% valid values is not surprising because several proteins in the data set are uniquely quantified in either slow or fast fiber or only after exercise. Upon the reviewer's request, we have now included total number of identification with 50% and 90% valid values (page 7, lines 14-15).

Other standard methods involve proteome analysis on whole muscle lysate using single run (single shot). In the revised manuscript, we have compared how the MED-FASP method is better than the single shot analysis.

As mentioned above this is not a fair comparison. Another fair comparison would be to exchange LysC in your workflow by for instance AspN, GluC, elastase, or LysN and proof that the combination of LysC with trypsin is better or at least cite some of the many papers in which this has been done before.

We appreciate this comment. This is a very valid point. Several studies describe the advantages of using multiple enzymes to increase protein identification and proteome sequence coverage. For example, in the first

MED-FASP paper (Wisniewski and Mann, Analytical chemistry 2012) two-step and three-step digestion with various combinations of trypsin, LysC, Glu-c, Arg-C and Asp-N were tested. Since we had limited starting material, we decided to use two-step digestion. In the above mentioned paper, when they compared two-step digestion, consecutive digestion using LysC and trypsin gave the best results. Therefore we utilized this strategy for this study. Nevertheless, other combinations, particularly with three-step digestion, might give superior results. In the revised manuscript, a couple of sentences describing these aspects have been added and previous work has been cited (page 8, lines 2-4).

We have also processed our data without match between runs option (myotubes). Without the 'match between run' option, only 2830 proteins in the fiber pools were identified.

The data is based on 5 subjects before and after training. To judge paired statistics it is important to mention which results are the pairs. It is not possible for the reader to check the statistics as this labeling is missing in the supplementary tables.

It is a valid point, we have now annotated the pairs in all revised supplementary tables.

Multiple testing correction is completely missing and would benefit the data analysis. The authors do not mention, at which fold change their method is valid. Fold change differences for the exercise dataset would be beneficial to judge if a significant hit has an impact.

This is valid concern and it is in line with issues raised by other reviewers. In the original dataset, all tests were performed in Perseus software. This software is user friendly and very popular in proteomics community but it has certain limitation. For example, this software does not have the ability to perform paired two way ANOVA and multiple testing correction. In the revised manuscript, we have re-analyzed data presented in figure 2, 3 and 4 where samples were compared in paired setting and p-values were adjusted applying a novel fusion scheme that takes both the biological significance (log2-change) and the raw p-value into consideration as described by Xiao et al. 2014. This method has been shown to be robust and to protect against false discoveries. We have also provided fold change information for all comparisons in the supplementary tables.

A volcano plot with fold change versus p-value for the exercise results gives direct information on the major changes. Please add such a plot for both fiber types to the main figures.

This is a very valuable suggestion. This has now been added in figure 3A+B.

The reporting of the results is often misleading. When changes are reported it is expected that these changes are significant. The manuscript describes a lot of proteins that are increased or decreased upon training. Some of them are not present in the dataset containing the significant changes (suppl. Tables) or could not be found by the reviewer. The authors need to check all proteins mentioned in the manuscript for their significance and if none significant proteins are described to change they need to mention that these proteins are not significantly changing and need to mention the reason why they still believe that these changes are possibly true. Especially for the chapter about glucose metabolism. For instance on page 18 the importance of changing HK2 is discussed, but HK2 is not in the suppl table 6 or 7 summarizing the significant changes and in the raw dataset (suppl. Table 2) 5 out of the 20 values are missing. Can the described differences be explained by missing values due to low abundance of the protein? Is it significantly changing? What is the significance level?

In the original manuscript, this confusion might have occurred because the statistical tests used in the previous version (analyses were performed using an unpaired ANOVA model and the p-values were unadjusted). We hope that the reviewer will find that these concerns are now adequately addressed in the revised manuscript. Additionally, in figure 5 and 6 while describing specific protein category (from examples TIMTOM complex, glycolysis etc), we combined the abundance of proteins from those specific category and compared their total abundance within and across the groups (slow pre, slow post, fast pre, fast post). At summed abundance level, some of these categories appear to be regulated by exercise in fiber type-specific manner. However, this does not mean that individual members of these categories are also significantly different. We presented our data this way to keep in mind the muscle biologist readers.

In the revised manuscript, we have discussed the abundances on HK2 in details and hopefully our explanation is now clear. Regarding the concern comparing Table S2 and Figure 6 – the values do not match because Table S2 contains raw, non normalized intensities and Figure 6 contains sum of LFQ intensities. In the revised manuscript, we have explained the details about the statistical methods and significance level.

In summed abundances, a significant change of one member of the group can lead to a significant change of the whole group. In respect of moonlighting functions of proteins, this might falsify the conclusion as already recognized by the authors. In order to prevent false conclusions the following numbers are required to be added: how many proteins are in a protein category, how many of these were quantified (not identified!) in the study, how many showed a significant change in the same direction of the summed change and how many in the opposite direction.

Fig 5 E LDHA fast: Visually there is no difference but it is indicated as $p < 0.01$. Is this correct?

We agree with this comment. Our approach to sum the abundance of proteins from specific protein categories is not very different from using GO annotations. We and others have utilized summed protein abundance in several papers (PMID: 25597705, 26825538, 26245529, 29350465, 25616865). As the reviewer points out, the proteins from different GO categories can have moonlighting functions and this may falsify conclusions. When we sum the abundance for specific protein class/category, we achieve statistical significance between two conditions (for example PRE vs POST). However, this does not mean that every single protein in that class/category is significantly regulated by training. Often very few proteins in that protein class/category are statistically significant. Nevertheless, all or most proteins show a trend of either upregulation or downregulation. To provide more clarity on this aspect, in the revised manuscript, we have updated Table S11 where we provided a list of individual protein members for each protein class/category with their fold change and significance. Similar information was already included for figure 6 (please refer to Table S12).

For this purpose, we have used only quantified values (not identified). As described previously, we considered protein quantified when it was quantified in at least 3 subjects in at least one group.

Regarding Fig 6E on LDHA expression, the symbol (††) indicates that there is a fiber type difference. There is no effect of training on LDHA expression in either slow or fast fibers, which we also state on page 21, line 18-19

In addition, why are the authors not mentioning the by far more abundant HK1 (see their raw data) and also do not mention it in the graphs? Does the claim p19 line5 “we observed that training increased expression of all glycolytic enzymes ...” still hold when HK1 is included? What about other identified glycolytic enzymes? ALDOC was identified but also not mentioned. Does the claim still hold? The authors need to check all glycolytic enzymes when they claim all glycolytic enzymes and not just pick one isoform, and if they do they need to explain why they exclude the others.

We have clarified this issue by modifying the text where we describe the effects of exercise training on glycolysis. Since we use slightly stringent criteria for quantification, we concluded that HKII is quantified in slow fibers only. HK1 was not regulated by exercise therefore it is not included in the text but in Table S12, we have included its fold-changes in response to exercise (both in slow and fast fibers). We did not include ALDOC because it was not quantified. In the revised manuscript, we present the list of glycolytic, PHD and TCA enzyme and their fold change and regulation (significance) upon exercise in both slow and fast fibers (Table S12).

On page 19 line 4-8 the authors write:

In addition, total abundance of glycolytic enzymes only increased with training in fast fibers (Fig. 6B), indicating training-induced enhanced capacity of glycolysis in fast muscle fibers. Interestingly, when comparing the abundance of individual glycolytic proteins, only HKII (in slow fibers) and Enolase 3 (in fast muscle fibers) were significantly upregulated with exercise training (Supplementary Table 12).

Here, the authors recognize that Enolase 3 is the only significant changing glycolytic protein in fast fibers. This leads to the total abundance of glycolytic enzymes to change. If the authors recognize that a conclusion is not correct as the pathway result is the result of a single enzyme, why do they now include it in the text. Later in the manuscript they mention glycolysis is changing. Again, the authors need to make sure that their claims about pathways/groups of proteins are not based on single proteins, but on the majority of the pathway.

It is a valid point, and we have now rephrased the paragraph (page 20, lines 9-19).

In the related figure S1a a lot of very small fold changes probably between 0.8-1.2 are reported. Are these changes all significant? Please indicate which are significant and which not.

In the revised manuscript, we have removed the figure S1a (glycolytic enzyme), S1c (PHD enzyme) and S1D (TCA enzymes) and presented their abundance in supplementary table S12. We have also included whether these proteins are significantly changing with exercise training (with fold change and significance score) in slow and fast fibers.

As reported by Hood et al 2016, exercise leads to changes in mitochondrial biogenesis. Such a change would probably result in abundance changes of the majority of mitochondria related proteins and metabolite requirements. This knowledge in respect of the findings is crucial. The authors should discuss if such a remodeling of the cell explains all detected protein changes and which changes are independent of mitochondrial biogenesis. Is this dataset describing a known effect of training, but is a detailed description of mitochondrial biogenesis on the protein level?

We cannot be completely sure that the increase in mitochondria-related proteins is not merely due to

increased mitochondrial content. However, we cannot directly measure mitochondrial content. One possibility is to use oxphos protein abundance as surrogate measures of mitochondrial content – but from a physiological point of view, any increase in mitochondrial related proteins likely results in increased capacity whether or not it is related to more organelles or more membrane area. We are open to suggestions on how to use our data to discuss this issue.

The most important point is that the authors write in the manuscript, that there is the possibility that the proteomic changes they see might be due to the changes in mitochondrial biogenesis as reported previously for instance by Hood et al. 2016. One possibility to differentiate between protein alterations related to or being unrelated to mitochondrial biogenesis would be a comparison with existing data on mitochondrial biogenesis. My suggestion would be to look up in the literature, if there are descriptions on which proteins are altered upon changes in mitochondrial biogenesis, preferentially more than a single dataset and compare them with the presented alterations.

We have consulted people working in the area of mitochondria, and none of them have been able to pin-point mitochondria proteins as specific markers for mitochondria biogenesis versus markers reflecting just expansion of the existing mitochondria membrane network (mitochondria volume).

Referring to Hood et al 2016 a range of methodologies is mentioned as potential readouts of mitochondria volume/ function. Given the material at hand in the present study many of these cannot be applied to this study:

- We do not have material prepared for imaging and fluorescence. Thus, fluorescence or electron microscopy is not an option.
- mtDNA has not been isolated in the fiber pools. To do this, it would require a whole new set of fiber isolation which is very time consuming and difficult due to the limited material left. TFAM activates transcription of mtDNA and packages mtDNA and its expression is suggested to change in parallel with mtDNA amount, and could serve as a proxy for mitochondrial biogenesis (PMID: 15016765). However, the copy number of mtDNA in a single mitochondria is not always 1 to 1 and the role of other proteins cannot be ruled out (PMID: 21808029). Thus, even though we did observe increased TFAM expression with training in both fiber types, which could suggest increased mtDNA and hence (perhaps) mitochondrial biogenesis, this is not a perfect marker. Furthermore, any associations between TFAM and other proteins are difficult to do due to the low number of samples
- Functional studies using measuring mitochondria protein synthesis or respiration would have required isolation of fibers from fresh tissue, which has never been available for the present study.
- Estimates of mitophagy, requires more sophisticated analyses on mitochondrial subfractions (e.g. localization of autophagy adapter proteins, such as the lipidated form of microtubule-associated protein light chain 3 (LC3-II) or p62.
- Expression profiles or activities of nuclear encoded enzymes such as CS and SDH have been used as markers of mitochondria volume as have the holoenzyme COX (subunits of both mitochondria and nuclear encoded genes). The use of these marker enzymes or subunits therein, often parallels

morphometric estimates of mitochondrial volume derived using EM. Our observations of increased CS and some of the COX subunits could at best indicate an increase in mitochondria volume.

Due to these reasons we have chosen to avoid the term mitochondrial biogenesis in the current version of the manuscript.

In general for all figures containing box or bar plots. As the sample number is 5 or less please add the individual points instead or in addition to boxes or bars.

Since we have included raw data and the error bars are rather small, we do not think that this is necessary. It is also not common procedure in the proteomics literature. However, if the reviewer think it is necessary, we can do it.

If the reviewer would have thought it is not necessary, he would have not mentioned it as a major point. It is good practice to do so and most of the good publications in proteomics follow this good practice. This is also the reason why it is mentioned in the Nature Editorial Policy Checklist that you have signed. See the point: "Data representation; Point: Individual data points are shown when possible, and always for $n < 10$ "

We are sorry for our misinterpretation of this comment. Individual data points are now shown in the figures containing bar graphs.

Minor

In the summary the authors mention in line13-14"... it altered expression levels of proteins involved in transcription, post-translational modifications, ..." but post-translational modifications were not mentioned to be regulated in the text. Please remove PTMs or describe in more detail, which PTMs were analyzed.

The sentence has now been altered

The authors removed PTM in the summary, but now added a comment in another part of the manuscript: *Page 5, line 15-16: This approach is advantageous in future measurements of fiber type-specific posttranslational modifications to ...* Why do the authors introduce a new claim, that this method will be advantageous for PTM analysis. There is no relation to PTM mentioned before.

We agree with reviewer's comments. In the revised manuscript, we have removed this sentence.

To judge paired statistics it is important to mention which results are the pairs. It is not possible for the reader to check the statistics as this labeling is missing.

It is a valid point. We have now annotated the pairs in all revised supplementary tables.

The dot plots about the results from the glycogen measurements are missing, please add as panel in the main figure or as supplementary figure. What was the target of the glycogen antibody; glycogenin or the sugar? Please describe the antibody in more detail in the methods section. There are no statistical measurements for the glycogen analysis of the 4 measurements. Please do the calculations and mention that $n=4$ in the text. Please also add the results of the controls or standards that were used.

The representative dot blot of glycogen is already included in the main figure 1C. The entire blot containing standards and samples has been included in the Source Data File. The standards used were samples with a known concentration of glycogen measured biochemically. They show a dynamic range of samples with low and high glycogen levels, respectively DOI:10.1113/jphysiol.2014.283267. The antibody recognizes the α -1,6 linkages in the glycogen structure, this information has now been added to the method section. Statistical measurements have now been performed for glycogen measurements in fiber pools. The effect of training was not statistically significant but close (slow fibers $p=0.06$, fast fibers $p=0.1$). The low n and thus low power is likely the explanation. The individual values are included in the figure. We have added in the method section that the analysis was performed on $n=4$.

The reviewer could not find the statistics in the manuscript. Why do the authors not mention the statistics in the manuscript after calculating and realizing that they are not significant (for $p<0.05$)? This is a very important finding that the claims on glycogen are not significant for $p<0.05$ and based on $n=4$. Still, the glycogen measurements have a tendency and this finding should be kept in the manuscript, but please also mention the statistics in the main text of the manuscript and do not hide the $n=4$ in the method section; add it to the main text. Be fair to your readers.

We are sorry about this mistake. We completely agree that the statistics should be included in the manuscript. Figure 1C and the sentence regarding glycogen in whole muscle and the fiber pools have now been updated with this information (Page 6, lines 14-15)

Page7 line5 "...we used a stringent FDR..." . A FDR of 1% is standard in proteomics analysis.

The sentence has been modified and stringent has been removed.

The purity of the fibers are tested by antibodies and the proteomics results. When comparing these results it would be beneficial to use a common nomenclature, like the gene name. Please further define the antibodies used in the materials section.

The heading in figure 1F is changed to the same nomenclature as figure 1B. The gene names of myosin heavy chains are kept to highlight specific isoforms.

On page 8 line 11 "Thus, our fiber isolation procedure and proteomic workflow yielded detailed coverage of metabolic..." Up to this point no detailed coverage of metabolic features resulting from the proteomic workflow has been mentioned to come to this conclusion. Please clarify which features did the authors mean.

The sentence has been modified.

Page10 line2 The authors find an enrichment of PDLIM1 and discuss the potential interaction with FHL1. FHL1 is not changing or very little. Can the authors implement this finding in the discussion and discuss why one of the interaction partner is eight fold enriched but the other not.

FHL1 did not change with exercise training; we have added this information to the discussion.

Fig2F and G Please mention in detail in the methods section how the results were calculated and which statistics were used. The number of proteins per group $n=...$ would be beneficial in the figure and an indication

how the abundance was calculated in the legend; median or average or sum. If the sum was used, how was it normalized if the number of proteins in the different groups was different.

The results were calculated as summed protein abundances for the proteins of the different categories. We have included a table, which lists the proteins in each of the categories of fig 2F and 2G (Table S7). The majority of proteins were quantified in both fiber types therefore there was no concern regarding number of proteins being different in different groups. We used paired linear models with Xiao correction to evaluate fiber typespecific differences in these figures (p 30, line 3-12)

P12 line16 What is the fold change of CARM1. The z-score (fig 3) removes the variance, please add the fold difference, p-values and add the multiple testing corrected p-value to the figure3 c-g. Box plots out of only max.5 data points are often misleading. The individual data points should be added to the box plots or replace the box plots by the individual points.

The previous data was based on non-corrected p-values. As suggested by the reviewers, the current version of the manuscript is based on corrected p-values. This means that the section referred to here is no longer a part of the manuscript.

P34 line 18 Did the authors really use a C10 column?

It was a typo and it has now been corrected to C18.

Additional comments:

Page 9, line 10: The authors mention 469 significant changing proteins between slow and fast fibers. "Statistical analysis returned 469 proteins that were significantly different 11 between slow and fast fibers (significance score<0.05, Fig. 2D, Supplementary Table 6)." But in Suppl. Table. 6 there are 471 proteins marked as significant. Please make sure to report the correct numbers.

Page 11, line 16: ...we found 236 and 171 proteins that were significantly...
The number of proteins marked with "+" in the Suppl. Table 8 are 237 and 172 and after checking the p-value entries these were only 170 < 0.05 in the fast fibers. Please check the whole manuscript again and define if you are using $p < 0.05$ or $p \leq 0.05$. Make sure you always work with the same set of proteins and report the same number.

Thank you for pointing out this mistake. We were indeed not consistent with the information in the text and supplemental table. For all comparisons involving proteomics data, we have used $p \leq 0.05$. In the revised manuscript, we have double checked the numbers and corrected the text accordingly.

Minor:

Typo in the affiliation of the first author.

In the revised manuscript, the typo in the affiliation has been corrected.

Typo in Suppl. Table 2 is labeled as Suppl. Table 1 in excel.

In the revised manuscript, this has been corrected.

Page 80, line 22: Typo PREvs.Post

This typo under figure legends for Figure 5 has been corrected.

In summary, the authors need to check their whole manuscript again to assure that the numbers are consistent throughout the manuscript. They should be transparent with their statistics and if a claim is based on missing data (meaning the proteins is quantified in $n < 5$ per group) it should be mentioned along with the claim in the main text of the manuscript.

We really appreciate the thorough comments and comprehensive work on the entire manuscript, figures and tables by reviewer 3. We have considered all suggestions and revised accordingly and believe that this has improved the manuscript.

In the second revision of the manuscript, we became aware of some inconsistency in how we have written abbreviations of the many proteins we discuss in the manuscript. This has now been made consistent. Inevitably, this increased the number of words therefore we did many minor corrections to comply with word limit by nature communications.

Reviewers' Comments:

Reviewer #1:

Remarks to the Author:

Deep muscle-proteomic analysis of freeze-dried human muscle biopsies 1 reveals extensive fiber 2 type-specific adaptations to exercise training

Authors performed an extensive study on the effects of exercise training on the proteome of different skeletal muscle fiber types. Different fiber types exhibit type-specific adaptive responses to training and especially metabolic pathways and mitochondria seemed to be affected.

After further revision the manuscript significantly improved and most but not all comments were addressed adequately. Some points and questions remain to be clarified before publication:

Page 7, lines 3-5: "Stringent identification criteria" are not clear and authors should mention in the text whether 50% of valid values was applied on the total data set or on specified groups.

Page 9, lines 4-6: In the Figure 31.8% are reported for Component 1. The information should be reconciled.

Page 9, lines 12-13: The significance score mentioned here should be explained. To my understanding authors refer to p30 I 22, but clarification is needed here.

Page 12, lines 7-9: This is not surprising as participants only conducted cardio related exercise. Authors should include information on typical exercise related fiber type loads e.g. Type I needed for long endurance running Type II for fast movements.

Page 14, lines 19-21: A closer look to on/off proteins would be of high interest though. Authors should review their data on existing on/off proteins and provide information on them as well (at least give accessions/names in the tables). (see also comment Legend figure 4)

Page 22, subjects: As already pointed out in the first review, authors do not explain why they solely performed an analysis on five subjects from the cohort. Clarification is still missing here.

Page 24, Human skeletal muscle analyses: This was also a point raised in the first review which was not addressed so far. Authors should clarify why the rested leg was used for the proteomics analysis as one should conclude that greater changes should be observable in the trained leg.

Page 30, lines 20-22 & figure 2 legend: As the method described by Xiao et al. is quite novel and the majority of the readers will not be familiar with it, authors should provide a short explanation on the method here.

Legend figure 2: It is not clear whether the authors used uncorrected p-values here or the mentioned significant score. Please clarify this point (also for the following figure legends).

Legend figure 4: What does exclusive mean? Clarification is needed how on/off proteins are determined in this manuscript.

After clarification of those points the manuscript should be published in Nat Comm.

Reviewer #3:

None

Manuscript ID - NCOMMS-19-12954B

We would like to thank the reviewers for their comprehensive work and very appreciated comments. Below is the point-to-point response to reviewers. The response to reviewers answer is written in green color. While answering reviewer 3, we have kept the original comments and answers (blue and red) from the previous (2nd) revision. To check the new text in the revised manuscript, please refer to manuscript file with track changes. The page numbers and the lines highlighted in yellow correspond to the manuscript file with track changes.

Reviewer 1

Deep muscle-proteomic analysis of freeze-dried human muscle biopsies 1 reveals extensive fiber 2 type-specific adaptations to exercise training. Authors performed an extensive study on the effects of exercise training on the proteome of different skeletal muscle fiber types. Different fiber types exhibit type-specific adaptive responses to training and especially metabolic pathways and mitochondria seemed to be affected. After further revision the manuscript significantly improved and most but not all comments were addressed adequately. Some points and questions remain to be clarified before publication:

Page 7, lines 3-5: “Stringent identification criteria” are not clear and authors should mention in the text whether 50% of valid values was applied on the total data set or on specified groups.

Thank you for your comment. The 50% valid value criterion was applied to the total data set (not on specific groups). To provide more clarity on the identification criteria, we have followed your suggestion and specified that the 50% valid values were applied to the total data set (Page 7, line 4).

Page 9, lines 4-6: In the Figure 31.8% are reported for Component 1. The information should be reconciled.

Thank you for your careful reading. We have corrected this typo.

Page 9, lines 12-13: The significance score mentioned here should be explained. To my understanding authors refer to p30 | 22, but clarification is needed here.

According to the reviewer’s suggestion, we have explained the significance score on page 9 (line 17-21). The following sentences are added in the text-

When comparing the proteome between slow and the fast muscle fibers, differentially expressed proteins were identified using an a posteriori information fusion scheme combining the biological relevance (fold-change) and the statistical significance (p-value) as described previously. A π -value significance score cut-off of 0.05 was selected.

Page 12, lines 7-9: This is not surprising as participants only conducted cardio related exercise. Authors should include information on typical exercise related fiber type loads e.g. Type I needed for long endurance running Type II for fast movements.

This may be related to a misunderstanding of our study design. The 12 weeks training program was conducted using a stationary bicycle using both legs (normal biking). The training consisted of 4 sessions per week and included bouts of exercises with both low and high intense workloads. This ensures recruitment of the majority of muscle fibers (and thus both type I and type II fibers) in the knee extensors.

We believe the study protocol is both clearly and correctly described already (Page 24).

We also suggest reviewer to take a look at following answer (*in italic*) from our first revision

We think this may related to a misunderstanding. In the original study (Steenberg et al., 2019), subjects performed an acute bout of single-leg knee-extensor exercise before and after 12 weeks of endurance exercise training. The exercise-training regime consisted of indoor cycling with both legs. Thus, we ended up with biopsies from a rested untrained leg, an acutely exercised untrained leg, a rested trained leg, and an acutely exercised trained leg. For the present study, we chose to measure the response to training in the rested leg that was not confounded by the acute bout of exercise to get the pure effect of training.

Accordingly, for the current study, we only included the biopsies that were obtained from the rested leg before and after the training period, hence not being confounded by acute exercise.

Page 14, lines 19-21: A closer look to on/off proteins would be of high interest though. Authors should review their data on existing on/off proteins and provide information on them as well (at least give accessions/names in the tables). (see also comment Legend figure 4)

Yes, we completely agree. These on/off proteins or fiber-type specific proteins could be extremely interesting to study fiber-type specific effects. We have indeed included text describing these proteins on page 15 as well as the list in supplemental table S10. In addition to their accessions and names, we have included which of these proteins are regulated by exercise training. In the revised manuscript, we have also included text describing what exclusive means (page 15, lines 7-9)

Page 22, subjects: As already pointed out in the first review, authors do not explain why they solely performed an analysis on five subjects from the cohort. Clarification is still missing here.

During the first revision, we justified the inclusion of five subjects as below -

We thank the reviewer for this valid comment. Given the instrumentation time and resources to undertake single fiber isolation and subsequent mass-spectroscopy, we chose to select 5 random subjects from the original study. The sample size chosen is in accordance with that normally used in muscle proteomics (Murgia et al., 2017) (n=4 in each group), (Holloway et al., 2009) (n=5), (Hoffman et al., 2015) (n=4), (Rakus et al.,

2015) (n=4). *Since these are human samples, inter-subject variability is bound to be there but this was not analyzed prior to this study.*

In addition to these reasons, we had limited amount of material available for the analysis. We were not sure that the reviewer wanted us to include this information in the main text. In the revised version, we have added few lines justifying the reasoning behind inclusion of five subjects (Page 22, line 19-21).

Page 24, Human skeletal muscle analyses: This was also a point raised in the first review which was not addressed so far. Authors should clarify why the rested leg was used for the proteomics analysis as one should conclude that greater changes should be observable in the trained leg.

Please see response to comment #4 above

Page 30, lines 20-22 & figure 2 legend: As the method described by Xiao et al. is quite novel and the majority of the readers will not be familiar with it, authors should provide a short explanation on the method here.

To identify and rank differentially expressed proteins, we chose a novel posteriori information fusion scheme, combining the biological relevance (fold-change) and the statistical significance (p-value) as recently described by Xiao et al. The fusion scheme yields a protein significance score (π value) and is robust in selecting differentially expressed genes/proteins while protecting against false discoveries.

Legend figure 2: It is not clear whether the authors used uncorrected p-values here or the mentioned significant score. Please clarify this point (also for the following figure legends).

Perhaps the reviewer had missed the information in the figure legend. In all figure legends, we have mentioned that we have used the significance score.

Legend figure 4: What does exclusive mean? Clarification is needed how on/off proteins are determined in this manuscript.

We are sorry if this was not clear. In the revised manuscript, we have clarified this on page 15 lines 7-9.

After clarification of those points the manuscript should be published in Nat Comm.

Reviewer 3

Page 4: The manuscript reads as you have compared a single measurement (single shot) versus two measurements one for LysC and one for trypsin (MED-FASP one for each enzyme). This is not a fair

comparison as a measurement that is twice as long (MED-FASP) should always lead to more identifications. A good comparison would be to spend the same time on the MS for the single shot and the MED-FASP and guarantee the most effective analysis when the single shot is measured twice as long. In addition, twice the material is needed for the MEDFASP protocol. A valid comparison also includes a fair discussion about the drawbacks of the method, like the longer time in preparation, LC-MS time, sample amount... These points need to be mentioned and discussed.

Thank you for your comments. This is a very valid point. Due to the nature of the technology, peptides are picked up randomly during each run. Therefore we should have compared 2x measurements of single shot proteome with trypsin digestion to the MED-FASP (LysC and Trypsin) runs. In the original MED-FASP paper by our colleagues (Wisniewski and Mann, Analytical chemistry 2012) such comparison was done. They reported that parallel measurements of tryptic digest from HeLa lysate had 70% peptides in common while consecutive digestion using LysC and trypsin had only 3.6% peptides in common (that is why MED-FASP results in higher protein identifications). In our study, while comparing single shot vs MED-FASP we should have measured tryptic peptides twice. We believe that we would have got similar results as described in the original MED-FASP paper and our conclusion would not change much. We would have liked to re-measure these samples but unfortunately, we do not have the material left to do so.

As the reviewer suggested, in the revised manuscript, we have added the following sentence to highlight limitations of the MED-FASP method 'Whilst higher identifications are apparent with multiple digestion steps compared to single-shot proteomes, this comes at the cost of larger starting material and longer measurement times.' (Page 8, lines 4-6)

I do not agree that results from cell culture (HeLa cells) as used in the Analytical Chemistry study 2012 is comparable to results from human tissue in scarce amounts. This is at least what our own experience shows. Indeed, including the sentence in Page 8 is sufficient to make this point clear for the reader.

This is a very valid point. Thank you for this suggestion. Accordingly, we have included a couple of sentences to provide more clarity on this topic (page number 7, lines 17-20).

.....

Page 5: Thank you for your comment. When the matching was turned off, the total number of identification with 50% valid values was 2299 and with 90% valid values the identification was 1616. The marked reduction in the numbers with 90% valid values is not surprising because several proteins in the data set are uniquely quantified in either slow or fast fiber or only after exercise. Upon the reviewer's request, we have now included total number of identification with 50% and 90% valid values (page 7, lines 14-15).

The sentence marked in bold contradicts the statement on page 3 "PDLIM1 was the only protein, which was exclusively quantified in one group". Please clarify. A group-wise check/analysis might help to clarify this point.

Yes, exactly. Since the study involves 4 groups (pre slow, pre fast, post slow and post fast), the list of exclusive protein will vary depending on how comparisons are performed. When we wrote 'PDLIM1 was the only protein, which was exclusively quantified in one group' we referred to the Group-wise comparison (Figure 2C, Figure 3A, 3B). Since we have included all raw data and extensive list of supplemental table, we believe that it is possible to extract details on quantification of any proteins by any readers.

Reviewers' Comments:

Reviewer #4:

Remarks to the Author:

Deep muscle-proteomic analysis of freeze-dried human muscle biopsies 1 reveals extensive fiber 2 type-specific adaptations to exercise training

Authors performed an extensive study on the effects of exercise training on the proteome of different skeletal muscle fiber types. Different fiber types exhibit type-specific adaptive responses to training and especially metabolic pathways and mitochondria seemed to be affected.

In the current revised version of the manuscript authors addressed most of the raised questions satisfactorily. Results are very relevant for the scientific community.

There are still two points, which I want to point out:

p.5, l.8ff: In their introduction the authors only mention the isolation of fiber types by laser microdissection and their subsequent mass spectrometric analysis from paraformaldehyde sections. However, there are already a variety of publications that have successfully applied exactly this technique to frozen muscle biopsies of human and murine tissue that have been stored for several years, obtaining important information on the proteomic composition of fiber types, protein aggregates, as well as other cellular compartments (PMID: 23115302; PMID: 27393313; PMID: 32887649; PMID: 28009083). The authors should include this information in their introduction, since the analysis of frozen muscle tissue and the subsequent extraction of fiber types is not a novelty.

p.9 l.17ff. The citation given for the n-value is incorreced (it is reference 68 not 70)

Furthermore, I am still having trouble to understand, why this significance score was chosen. After reading the corresponding reference this score is actually used to achieve a consistent sorting of genes or proteins for enrichment analysis. This is achieved by multiplying log-FC and the $-\log_{10}$ -p-value. My concern is that there is no fixed threshold at which this combined value is significant. However, since the Students t-test and the fold change have to be calculated for this score, I assume that it will be no difficulty for the authors to add the Students t-test p-values and also a correction of it. Therefore, I would highly recommend and appreciate adding the Student's t-test p-value to the supplementary data provided (S6, S8), as this is the standard statistical value used in proteomics. Without well-established statistics, the scientific community cannot interpret the result's value.

Reviewer 4

Deep muscle-proteomic analysis of freeze-dried human muscle biopsies 1 reveals extensive fiber 2 type-specific adaptations to exercise training.

Authors performed an extensive study on the effects of exercise training on the proteome of different skeletal muscle fiber types. Different fiber types exhibit type-specific adaptive responses to training and especially metabolic pathways and mitochondria seemed to be affected. In the current revised version of the manuscript authors addressed most of the raised questions satisfactorily. Results are very relevant for the scientific community.

There are still two points, which I want to point out:

p.5, l.8ff: In their introduction the authors only mention the isolation of fiber types by laser microdissection and their subsequent mass spectrometric analysis from paraformaldehyde sections. However, there are already a variety of publications that have successfully applied exactly this technique to frozen muscle biopsies of human and murine tissue that have been stored for several years, obtaining important information on the proteomic composition of fiber types, protein aggregates, as well as other cellular compartments (PMID: 23115302; PMID: 27393313; PMID: 32887649; PMID: 28009083). The authors should include this information in their introduction, since the analysis of frozen muscle tissue and the subsequent extraction of fiber types is not a novelty.

Response:

In accordance with the reviewer suggestion, we have now reworded the MS in the abstract, introduction as well as summary and we hope you will find the text more balanced now. We now reference another of the suggested papers in our MS (PMID: 23115302 and PMID: 28009083) reflecting LCM –MS in human biopsy specimens.

p.9 l.17ff. The citation given for the π -value is incorreced (it is reference 68 not 70)

Furthermore, I am still having trouble to understand, why this significance score was chosen. After reading the corresponding reference this score is actually used to achieve a consistent sorting of genes or proteins for enrichment analysis. This is achieved by multiplying log-FC and the $-\log_{10}$ -p-value. My concern is that there is no fixed threshold at which this combined value is significant. However, since the Students t-test and the fold change have to be calculated for this score, I assume that it will be no difficulty for the authors to add the Students t-test p-values and also a correction of it. Therefore, I would highly recommend and appreciate adding the Student's t-test p-value to the supplementary data provided (S6, S8), as this is the standard statistical value used in proteomics. Without well-established statistics, the scientific community cannot interpret the result's value.

Response:

Thank you for pointing out the incorrect citation. In the revised manuscript, we have corrected this mistake.

In accordance with the reviewer's suggestion, we have added the p-values of each paired linear model analysis in the supplementary data tables. With regards to the concern of "no fixed threshold" of the significance score (Xiao et al. 2014), there is neither no such statistical consensus or particular rationale for the commonly applied fixed alpha level of 0.05, but for consistency we chose a cut-off of 0.05 as done by others who also applied the method described by Xiao et al. in other omics fields, including phosphoproteomics (Lin et al. 2020 - PNAS) and genomics (Liu et al. 2015 – Nature). Given the exploratory nature of the present study, we chose this approach to dichotomize regulated proteins. We performed a comprehensive validation of several target proteins by immunoblotting, which confirmed the mass-spec findings. Furthermore, several of the proteins that were significantly regulated by training in our study have been shown to be regulated by exercise training in previous studies. In contrast, the pitfall of undertaking a too conservative approach, such as multiple correction using the Bonferroni Method, is clearly illustrated for the present dataset, in which we would have erroneously concluded that exercise training only regulated a few of the more than 2000 proteins quantified. Hence, we argue that several of the commonly applied multiple correction procedures are ill suited for the purpose of exploratory analysis as in the present study.

References:

Shah P, Fritz JV, Glaab E, Desai MS, Greenhalgh K, Frchet A, Niegowska M, Estes M, Jäger C, Seguin-Devaux C, Zenhausern F, Wilmes P. A microfluidics-based in vitro model of the gastrointestinal human-microbe interface. *Nat Commun*. 2016 May 11;7:11535. doi: 10.1038/ncomms11535. PMID: 27168102; PMCID: PMC4865890.

Liu N, Dai Q, Zheng G, He C, Parisien M, Pan T. N(6)-methyladenosine-dependent RNA structural switches regulate RNA-protein interactions. *Nature*. 2015 Feb 26;518(7540):560-4. doi: 10.1038/nature14234. PMID: 25719671; PMCID: PMC4355918.

Lin Y, Wozniak JM, Grimsey NJ, Girada S, Patwardhan A, Molinar-Inglis O, Smith TH, Lapek JD, Gonzalez DJ, Trejo J. Phosphoproteomic analysis of protease-activated receptor-1 biased signaling reveals unique modulators of endothelial barrier function. *Proc Natl Acad Sci U S A*. 2020 Mar 3;117(9):5039-5048. doi: 10.1073/pnas.1917295117. Epub 2020 Feb 18. PMID: 32071217; PMCID: PMC7060683.